# INO-SGD: Addressing Utility Imbalance under Individualized Differential Privacy

**Xiao Tian**[1,2], **Jue Fan**[1,2], **Rachael Hwee Ling Sim**[1] **& Bryan Kian Hsiang Low**[1]
[1]Department of Computer Science, National University of Singapore
[2]Agency for Science, Technology and Research (A*STAR), Singapore
{xiao.tian, jue.fan, rachael.sim}@u.nus.edu
lowkh@comp.nus.edu.sg

## Abstract

*Differential privacy* (DP) is widely employed in machine learning to protect confidential or sensitive training data from being revealed. As data owners gain greater control over their data due to personal data ownership, they are more likely to set their own privacy requirements, necessitating *individualized DP* (IDP) to fulfil such requests. In particular, owners of data from more sensitive subsets, such as positive cases of stigmatized diseases, likely set stronger privacy requirements, as leakage of such data could incur more serious societal impact. However, existing IDP algorithms induce a critical *utility imbalance* problem: *Data from owners with stronger privacy requirements may be severely underrepresented in the trained model, resulting in poorer performance on similar data from subsequent users during deployment*. In this paper, we analyze this problem and propose the INO-SGD algorithm, which strategically down-weights data within each batch to improve performance on the more private data across all iterations. Notably, our algorithm is specially designed to satisfy IDP, while existing techniques addressing utility imbalance neither satisfy IDP nor can be easily adapted to do so. Lastly, we demonstrate the empirical feasibility of our approach.

## 1 Introduction

Machine learning (ML) focuses on building a *model* that draws patterns from existing *data* so that it can be applied to future prediction tasks. Since data is essential yet always scarce, *model owners* who possess and train ML models often have to collect data from individual *data owners*. For example, medical institutes acquire data from hospitals to train a model that predicts the risk of heart diseases (Panagopoulos et al., 2022). To build the best model, model owners should ideally be allowed to use all data without restrictions. However, this brings potential threat to data owners' privacy when some third-party users access the trained model. For example, adversarial techniques such as *membership inference attacks* (Shokri et al., 2017) can reveal identity or sensitive data. Thus, model owners must **ensure *data privacy* when training models** to achieve a long-term healthy collaboration.

Existing works have established advanced techniques to represent and preserve privacy. *Differential privacy* (DP) introduced by Dwork (2006) is a widely accepted measure of privacy. The core intuition behind DP is that personal information is protected if for any individual datum $d$, any dataset with $d$ results in a trained model that is almost indistinguishable from another model trained on the same dataset without $d$ with regards to a predetermined *privacy budget* set by data owners. Then a potential attacker cannot tell for certain from the trained model whether an individual datum is present in the dataset. Based on DP, many *differentially private ML* (DP-ML) techniques have been developed and deployed, including **differentially private stochastic gradient descent (DP-SGD)** (Abadi et al., 2016), where Gaussian noise is used to perturb the gradient at each SGD step.

With rising awareness of data protection and privacy, many laws have legally enforced *personal data ownership*, such as *General Data Protection Regulation* (GDPR) (Regulation, 2018) in the European Union and *California Consumer Privacy Act* (CCPA) (Pardau, 2018) in the United States. Such laws state that data are legal properties of their owners, which entitle data owners to decide the extent to which model owners can use their data and hence indicate their own privacy requirements.

In practice, **privacy requirements can vary across data owners** (Berendt et al., 2005; Jensen et al., 2005). For example, owners of data from more sensitive subsets (e.g., positive cases of stigmatized diseases, historically disadvantaged races) may require stronger privacy in classification tasks as data leakage could incur discrimination or denial of services (Best & Arseniev-Koehler, 2023; Lai & Tanner, 2022). At first glance, a model owner can fulfil such *individualized differential privacy* (IDP) requirements by maintaining the strongest level of privacy across all data owners. **However, as described by the *privacy-utility tradeoff* (Makhdoumi & Fawaz, 2013), using the strongest DP level undesirably reduces the utility (performance) of the trained ML model.** To exploit each owner's privacy budget as much as possible (to attain highest utility), Boenisch et al. (2024) proposes *individualized DP-SGD* (IDP-SGD) to achieve IDP, which we introduce in detail in Sec. 2.1.

However, **(I) does IDP-SGD have any limitation, such as when data owners holding different groups of data within the data space have different privacy requirements?** For example, in *classification* problems, each class might have different privacy requirements; privacy requirements may vary across *categories* of data sharing similar traits (e.g., different types of the same disease). In Sec. 3.1, we show that **IDP-SGD has the following issue**: when data owners with exclusive data have different IDP requirements, the trained ML model may have lower utilities for the more private owners. During deployment, the imbalanced utilities will negatively affect the model owner (e.g., medical institute with a disease model) and subsequent users (e.g., patients with the same diseases as the more private owners) affected by the model's decisions. This leads to the next question: **(II) Can we proactively correct this *IDP-induced utility imbalance* across data owners?** To address this, we propose the *individualized noisy ordered SGD* (INO-SGD) algorithm in Sec. 3.2. Our key idea is that **we should down-weight certain less important gradients, which in our case harm the model, as they worsen the imbalance and distract the model from learning the more difficult data, which would be from the more private owners or harder-to-predict data from less private owners**. Importantly, the algorithm has to be specially designed such that it still satisfies each data owner's IDP requirement, and we explain in Sec. 3.1 why existing techniques addressing data imbalance cannot be easily adapted to do so. We theoretically analyze the benefits of INO-SGD in Sec. 3.3 and demonstrate its empirical performance in Sec. 4.

## 2 BACKGROUND AND RELATED WORKS

### 2.1 (INDIVIDUALIZED) DIFFERENTIAL PRIVACY

Let $\mathcal{D}$ be the collection of datasets. Two datasets $D, D^d \in \mathcal{D}$ are *neighboring with regards to datum $d$* if they differ only by $d$ (i.e., $D \triangle D^d = \{d\}$, where $\triangle$ denotes symmetric difference). A randomized *algorithm* $\mathcal{A} : \mathcal{D} \to \boldsymbol{\Omega}$ takes in a (training) dataset $D \in \mathcal{D}$ and returns a (randomized) output vector $\boldsymbol{\omega} \in \boldsymbol{\Omega} \subseteq \mathbb{R}^r$. Let $p_{\mathcal{A}}^D$ denote the distribution of $\mathcal{A}(D)$. *Differential privacy* (DP) (Dwork, 2006) works by comparing the output distributions $p_{\mathcal{A}}^D$ and $p_{\mathcal{A}}^{D^d}$ when algorithm $\mathcal{A}$ takes in neighboring datasets $D$ and $D^d$. Intuitively, privacy is protected if any attacker cannot easily distinguish the output distributions produced by any pair of neighboring datasets. In App. B.1, we introduce different notions of DP, including $(\epsilon, \delta)$-*DP* (Dwork et al., 2006) and *Rényi DP* (Mironov, 2017), which differ in how the difference between $p_{\mathcal{A}}^D$ and $p_{\mathcal{A}}^{D^d}$ is measured. All these notions involve a parameter $\epsilon \geq 0$ called *privacy budget*: a smaller $\epsilon$ (close to 0) requires both distributions to be more indistinguishable, and hence corresponds to *stronger* privacy.

Suppose there are $N$ data owners $[N] := \{1, 2, \cdots, N\}$ with subsets $(D_n)_{n \in [N]}$ and privacy budgets $\boldsymbol{\epsilon} := (\epsilon_n)_{n \in [N]}$. *Individualized DP* (IDP) works by restricting the difference in output distributions $p_{\mathcal{A}}^D$ and $p_{\mathcal{A}}^{D^d}$ by the unique privacy budget of the differing datum $d$'s owner, $o(d)$. Formally,

**Definition 2.1** (IDP (Alaggan et al., 2016))**.** A randomized algorithm $\mathcal{A} : \mathcal{D} \to \boldsymbol{\Omega}$ satisfies $(\boldsymbol{\epsilon}, \delta)$-*individualized differential privacy* ($(\boldsymbol{\epsilon}, \delta)$-IDP) if for any datum $d$, any pair of neighboring datasets $D, D^d \in \mathcal{D}$ such that $D \triangle D^d = \{d\}$ and every possible output set $\mathbf{O} \subseteq \boldsymbol{\Omega}$,

$$\Pr[\mathcal{A}(D) \in \mathbf{O}] \leq e^{\epsilon_{o(d)}} \cdot \Pr[\mathcal{A}(D^d) \in \mathbf{O}] + \delta.$$

$\mathcal{A}$ satisfies $(\boldsymbol{\alpha}, \bar{\boldsymbol{\epsilon}})$-*individualized Rényi DP* ($(\boldsymbol{\alpha}, \bar{\boldsymbol{\epsilon}})$-IRDP) if for any datum $d$ and any pair of such $D$ and $D^d$, the *Rényi divergence* of order $\alpha_{o(d)}$ (see App. B.1.2) between $p_{\mathcal{A}}^D$ and $p_{\mathcal{A}}^{D^d}$ satisfies

$$\mathfrak{D}_{\alpha_{o(d)}} \left( p_{\mathcal{A}}^d \,\|\, p_{\mathcal{A}}^{D^d} \right) \leq \bar{\epsilon}_{o(d)}.$$

Through algorithm $\mathcal{A}$, owners with smaller privacy budgets $\epsilon_n$ attain stronger privacy. To fulfil IDP, Boenisch et al. (2024) proposes two IDP-SGD variants, SAMPLE and SCALE (refer to App. B.2). SAMPLE samples owner $n$'s data using an *individualized sampling rate* $q_n$; SCALE controls the algorithm's *modular sensitivity* $\Delta_{\mathcal{A}}^d := \sup_{D \triangle D^d = \{d\}} \|\mathcal{A}(D) - \mathcal{A}(D^d)\|$ with respect to each datum $d$, which ensures the maximum change a datum can bring is restricted by its owner's privacy budget. To achieve this, SCALE scales each individual gradient $\mathbf{g}_d$ such that its norm does not exceed its owner $o(d)$'s *individualized clipping threshold* $C_{o(d)}$. The processed gradients are then summed up and perturbed by a Gaussian noise of $\mathbf{0}_r$ ($r$-sized zero vector) mean and $\sigma^2 \mathbf{I}_{r \times r}$ ($r \times r$ identity matrix) variance where $\sigma$ is the *noise scale*. The privacy guarantee of IDP-SGD is as follows:

**Theorem 2.2** (Privacy of IDP-SGD (Boenisch et al., 2024)). IDP-SGD *running for $T$ iterations satisfies $(\boldsymbol{\alpha}, \bar{\boldsymbol{\epsilon}})$-IRDP, where $\bar{\epsilon}_n = 2T\alpha_n q_n^2 C_n^2 / \sigma^2$.*

## 2.2 DATA IMBALANCE AND RELEVANCE TO THIS WORK

In this section, we introduce the *data imbalance* issue in ML and its relevance to this work. Let $\boldsymbol{\theta}$ be model parameters and $\ell(\boldsymbol{\theta}; d)$ denote the loss on datum $d$ under $\boldsymbol{\theta}$. Let $\mathcal{L}(\boldsymbol{\theta}; D) := \sum_{d \in D} \ell(\boldsymbol{\theta}; d)$ be the optimization objective (empirical loss). Suppose the training set can be partitioned into disjoint groups of data $g_1, g_2, \cdots, g_N$ such that data within each group share similar traits and data across different groups have different traits. Thereby, each group has its own optimization objective $\mathcal{L}_n(\boldsymbol{\theta}) := \mathcal{L}(\boldsymbol{\theta}; g_n)$. Data imbalance refers to (†) **the imbalance in sizes of different groups in the training set** (hence in each sampled batch of gradient-based algorithms under uniform sampling), that is, $|g_1|, |g_2|, \cdots, |g_N|$ differ significantly. For example, *class imbalance* (Niaz et al., 2022) exists when the training set $D$ is partitioned based on class labels.

Data imbalance has two major negative impacts on gradient-based training algorithms such as SGD: *minority initial drop* (**MID**) (Ye et al., 2021) and *biased optimization objective* (**BOO**). We illustrate them by showing a typical learning curve under data imbalance in Fig. 1(a) considering 2 groups, minority group $g_1$ (RED) and majority group $g_2$ (BLUE)[1]. **MID** refers to an initial drop or fluctuation in model utility on minority group $g_1$, while utility on majority group $g_2$ improves as normal. As shown in Fig. 1(b), let $\mathbf{G}_{t,g_1}$ and $\mathbf{G}_{t,g_2}$ denote the summed gradient of $g_1$ and $g_2$ used to update model parameters from $\boldsymbol{\theta}_{t-1}$ to $\boldsymbol{\theta}_t$ at iteration $t$. Let $\angle_t$ denote the angle between $\mathbf{G}_{t,g_1}$ and $\mathbf{G}_{t,g_2}$. Francazi et al. (2023) shows that for full-batch gradient descent to ensure the losses for both groups strictly decrease (in expectation) at iteration $t$, $\angle_t$ has to satisfy the following condition (Cond. (1)):

$$1 + \cos\left(\angle_t\right) \frac{\|\mathbf{G}_{t,g_2}\|}{\|\mathbf{G}_{t,g_1}\|} > 0. \tag{1}$$

At earlier iterations, when $\boldsymbol{\theta}_{t-1}$ is far from the optimum for both groups, every individual gradient is large. Thus, the magnitude $\|\mathbf{G}_{t,g_2}\|$ for majority group $g_2$ is much larger than $\|\mathbf{G}_{t,g_1}\|$ since it contains more gradients. The angle $\angle_t$ is expected to be large as the directions of $\mathbf{G}_{t,g_1}$ and $\mathbf{G}_{t,g_2}$ differ when the two groups have different optimization objectives. As $\cos(\angle_t)$ and hence the left-hand side of Cond. (1) are negative, Cond. (1) is not satisfied. Thus, when the loss for majority group $g_2$ decreases, the loss for minority group $g_1$ may not. The loss for $g_1$ is only guaranteed to decrease when Cond. (1) is satisfied after more iterations: as the model parameters approach the optimum of $g_2$, the gradients in $g_2$ become small in magnitude, making the left-hand side of Cond. (1) positive. **BOO** applies to the final optimum found as shown in Fig. 1(c) when there are sufficient iterations for the model to converge. The training process will focus more on reducing the expected loss $\mathbb{E}_{d \in g_2}[\ell(\boldsymbol{\theta}; d)]$ for majority group $g_2$. This is because with more data from $g_2$, a decrease in $\mathbb{E}_{d \in g_2}[\ell(\boldsymbol{\theta}; d)]$ will bring a larger decrease in the objective $\mathcal{L}(\boldsymbol{\theta}; D) = \mathcal{L}_1(\boldsymbol{\theta}) + \mathcal{L}_2(\boldsymbol{\theta})$ than a decrease in $\mathbb{E}_{d \in g_1}[\ell(\boldsymbol{\theta}; d)]$. Therefore, BOO biasedly results in less optimal utility for $g_1$.

At first glance, readers may think that MID is less harmful than BOO because MID would disappear with more iterations. However, this is not true under (I)DP which only allows a limited number of iterations. It is possible that the privacy budgets are used up when MID still goes on, causing a significantly low utility for the minority groups, especially when the ML task is difficult (shown later in Sec. 4), **making MID more harmful**. Thus, under (I)DP we should **focus on the entire learning dynamics**, which refers to the model utility for each group at every stage of training.

In our work, however, we do not assume data imbalance (†) (i.e., each owner can have a similar number of data). Instead, we focus on how **IDP induces the same impacts MID and BOO** for

---

[1]This can be generalized to multiple groups (Francazi et al., 2023).

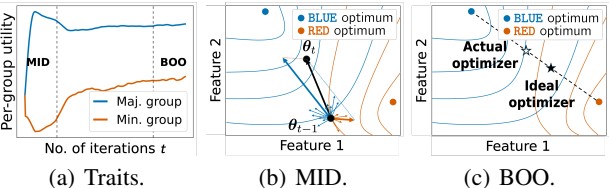
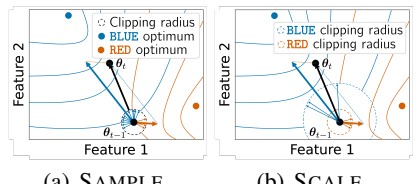

(a) Traits.    (b) MID.    (c) BOO.       (a) SAMPLE.    (b) SCALE.

Figure 1: **Graphical illustration of traits and causes of MID and BOO under data imbalance.** MID happens at earlier stages due to imbalanced gradients, while BOO describes the final stage where the trained model favors majority classes. Here **BLUE** and **RED** optimum refer to the optimizers within the group (i.e., $\arg\min_{\boldsymbol{\theta}} \mathcal{L}(\boldsymbol{\theta}; g_1 \text{ or } g_2)$).

Figure 2: **Illustration of IDP-induced utility imbalance with** 2 **data owners, BLUE (less private) and RED (more private).** Model parameters $\boldsymbol{\theta}$ are updated in a direction that reduces **BLUE**'s loss but increases **RED**.

gradient-based algorithms, which lead to **utility imbalance** across different groups. In Sec. 3.1 and App. B.3, we explain why this **cannot be solved in the same way as existing methods** tackling data imbalance. A separate branch of works on DP and biasedness of ML models is *disparate impact of DP* (Bagdasaryan et al., 2019), which studies how DP-ML models exacerbate data imbalance in the training set. Our work differs as we consider IDP and focus on the **underrepresentation of data from the more private owners**. The training set itself is not necessarily imbalanced.

# 3 METHODOLOGY

## 3.1 IDP-INDUCED UTILITY IMBALANCE

We consider the setting where the training set still contains disjoint groups of data $g_1, g_2, \cdots, g_N$ with respective optimization objectives $\mathcal{L}_n(\boldsymbol{\theta})$, but the training set $D$ can be fairly balanced and each group is of similar size. In this section, we show that **MID, BOO and utility imbalance can still occur when owners of data from different groups have different IDP requirements**. For example, when owners of positive-case data of stigmatized diseases prefer stronger privacy in classification tasks, IDP can induce utility imbalance (see App. A.3 for more examples).

We first give the intuition why utility imbalance can still arise from both the SAMPLE and SCALE variants of IDP-SGD on balanced datasets. We demonstrate with 2 data owners **BLUE** (less private) and **RED** (more private) and plot visualizations in Fig. 2.

- **SAMPLE**: In SAMPLE, data owners share the same clipping threshold but have different sampling rates. App. C.1.1 shows that the sampling rate for a less private owner can be significantly larger than a more private one. Thus, in Fig. 2(a), there are more gradients from **BLUE** than **RED** at each iteration. Since all gradients are clipped to the same magnitude, the sum of clipped gradients from **BLUE** is much larger in magnitude than **RED**, causing MID to occur. Similarly, the training focuses more on reducing the expected loss of **BLUE** than **RED** and BOO occurs.
- **SCALE**: In SCALE, data owners share the same sampling rate but have different clipping thresholds. App. C.1.1 shows that the clipping threshold for a less private owner can be significantly larger than a more private one. Thus, in Fig. 2(b), although the numbers of gradients from both owners are similar, each clipped gradient from **BLUE** has a much larger magnitude than **RED**. The magnitude of **BLUE**'s summed gradients is thus much larger than **RED** and MID occurs. Such biased gradients similarly focus more on reducing the expected loss of **BLUE** than **RED** and BOO occurs.

Next, we present the following theorem to theoretically justify why data from the more private owners might not have a decreasing loss under IDP (and hence their utilities may drop):

**Theorem 3.1** (Informal). *Consider two data owners with subsets $D_1, D_2$, sampling rates $q_1, q_2$ and clipping thresholds $C_1, C_2$. At each iteration $t$, if the following condition (**Cond. (2)**)*

$$1 + \cos(\angle_t)\frac{q_2|D_2|C_2}{q_1|D_1|C_1} > \varsigma \tag{2}$$

*is satisfied with $\varsigma > 0$ being a problem-specific term, by choosing a sufficiently small learning rate $\eta$, the expected losses for both owners are guaranteed to decrease under standard assumptions. When Cond. (2) is not satisfied (e.g., $\leq 0$), the loss for owner 1 may increase (MID).*

App. C.1.2 proves and further discusses the theorem. Cond. (2) suggests that when the two owners have different optimization objectives ($\angle_t$ is large and $\cos(\angle_t) < 0$) and similar number of data ($|D_1| \approx |D_2|$), the more private owner 1 who has a much smaller sampling rate $q_1$ or clipping threshold $C_1$ in the denominator (hence left-hand side of Cond. (2) $\leq 0$) could suffer from an increasing loss. Unlike Cond. (1) (for data imbalance), Cond. (2) (for IDP-induced utility imbalance) requires a problem-specific term $\varsigma > 0$ due to IDP-SGD components such as individualized clipping which also makes it harder to satisfy. For the IDP-SGD algorithm, as the clipping thresholds are fixed, Cond. (2) may remain unsatisfied across many iterations for some owners (e.g., the more private owner or minority group). Thus, **MID becomes longer and more severe.**

**Undesirability of IDP-induced utility imbalance.**    If the ML model has significantly lower utility on some group (e.g., due to positive-case patients having stronger IDP requirements), the model owner and subsequent users with data in such group who are impacted by the model decisions are affected. Thus, the model owner must address such IDP-induced utility imbalance during training to improve the utilities of worst-off groups during deployment (see App. A.3).

**Unsuitability of existing works.**    App. B.3 gives a detailed review of existing works (possibly loosely) related to utility imbalance and their unsuitability to tackle IDP-induced utility imbalance under IDP. To summarize, we classify existing works into three categories:

**(1) Correcting data/class imbalance**: This category is the most relevant as they also seek to correct utility imbalance across groups. Their core idea is to control the per-group summed gradients $\left\|\mathbf{G}_{t,g_n}\right\|$ in Cond. 1 to obtain more balanced gradients to reduce MID and BOO. We note that adapting such works either (a) violate IDP requirements (e.g., by upscaling or oversampling the minority groups (He & Garcia, 2009)), (b) under-utilize the IDP budgets and give lower overall utility (e.g., by downscaling or undersampling the majority groups (He & Garcia, 2009)), or (c) have their effects eliminated by clipping (e.g., rescaling the loss function (Japkowicz, 2000)).

**(2) Correcting DP-exacerbated accuracy disparity** (Bagdasaryan et al., 2019; Tran et al., 2021a; Xu et al., 2021): These works do not satisfy IDP and are tailored to DP's disparate impact from clipping and noise addition, yet IDP-induced utility imbalance arises from an orthogonal cause.

**(3) Achieving group/individual fairness**:    These works seek to achieve similar **predictions/behaviors** across protected groups, which greatly differs from our goal to achieve similar **model utilities** across data owners. Thus, we do not compare with this category of works.

Thus, there is no prior work that addresses IDP-induced utility imbalance or can be easily adapted to do so. Our work seeks to fill this gap while still adhering to each owner's IDP preference.

## 3.2    THE INO-SGD ALGORITHM

The imbalanced gradient, sampling rate and clipping threshold (Fig. 2 & Cond. 2) in IDP-SGD cause the IDP-induced utility imbalance (MID and BOO) and must be addressed. While gradients from the more private owners cannot be upscaled due to IDP, approaches that downscale gradients from the less private owners would waste privacy budgets and lead to a lower overall utility due to the privacy-utility tradeoff (see App. C.2.1 for further discussions and D.1.3 for empirical evidence). How can we instead **strategically discard the less important information (e.g., from less private owners) while retaining the most important information**? The importance of gradient information of each datum in batch $B_t$ at iteration $t$ directly correlates with their individual loss values (Kawaguchi & Lu, 2020; Shrivastava et al., 2016). Thus, at first glance, the solution should keep only

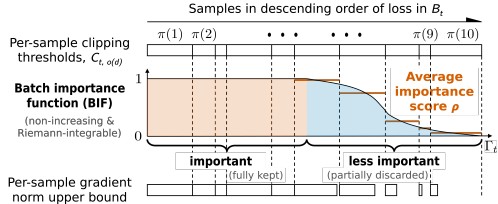

Figure 3: **Graphical illustration of the INO-SGD algorithm.** At iteration $t$, a batch $B_t$ (e.g., 10 data) is sampled. The gradients are computed and sorted by descending loss. INO-SGD calculates the average importance score of each gradient by integrating the importance function within its associated interval. By multiplying the clipped gradients to their scores, important gradients are fully kept while less important ones are partially discarded.

the gradients with highest-$\mu$ losses or discard the gradients with smallest-$\mu$ losses. However, these simple approaches violate IDP by increasing the algorithm's modular sensitivity (see App. C.2.1).

To exploit the order of losses[2] while still preserving IDP, we devise a unique algorithm, ***individualized noisy ordered SGD*** (INO-SGD) (Fig. 3). The key intuition behind INO-SGD is as follows: When a new datum is added, INO-SGD first examines if its gradient $\mathbf{g}_d$ is important based on the rank of $d$'s loss. If it is deemed important, INO-SGD simply clips it to its clipping threshold $C_{o(d)}$; otherwise, INO-SGD restricts its norm to an upper bound smaller than $C_{o(d)}$, which makes room to upscale other more important gradients while the total change is capped by $C_{o(d)}$, thus ensuring modular sensitivity $\Delta_{\mathcal{A}}^d \leq C_{o(d)}$ and achieving IDP (further intuition is given in Sec. 3.3.1).

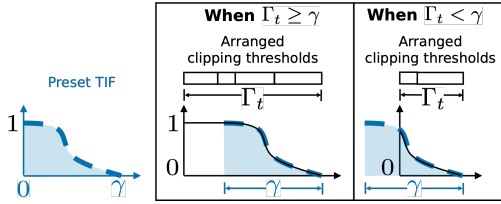

Figure 4: **At each iteration $t$, BIF $f_t$ (solid line) is constructed by transforming TIF $f_{\text{tail}}$ (blue dashed line).** The $x$-axes refer to the position of each ordered gradient piece and the $y$-axes refer to its importance score.

Specifically, INO-SGD adopts a "continuized" view of each datum's gradient as a collection of infinitely small gradient pieces whose loss values are all equal to the datum's loss (see Fig. 3). For each datum $d$, at most a total of $C_{o(d)}$ of its gradient pieces are kept and the rest are discarded to ensure modular sensitivity $\Delta_{\mathcal{A}}^d \leq C_{o(d)}$. Hence, the sum of gradient pieces is at most $\Gamma_t := \sum_{d \in B_t} C_{o(d)}$. Suppose all kept gradient pieces are ordered in descending order of loss. An *importance function* assigns an importance score between 0 (least important) and 1 (important) to each gradient piece, which indicates how much of the piece will be eventually kept. Moreover, since the summed gradient will be made noisy, INO-SGD only assigns low importance scores to a **limited** number of gradient pieces to maintain a reasonably high signal-to-noise ratio. This is captured by a preset *tail importance function*: only gradient pieces falling within the tail of length $\gamma$ are assigned scores $< 1$.

**Definition 3.2** (TIF). For a fixed length $\gamma > 0$, a *tail importance function* (TIF) $f_{\text{tail}} : [0, \gamma] \to [0, 1]$ is a non-increasing, Riemann-integrable function such that for any $c \in [0, \gamma]$, $f_{\text{tail}}(c)$ represents the importance of gradient piece at the $(c/\gamma)$-th quantile over the tail of length $\gamma$.

Based on TIF, for each sampled batch $B_t$, we define its unique *batch importance function* (BIF) by assigning importance score 1 to the gradient pieces outside the tail (i.e., they are important) as shown in Fig. 3. In other words, the domain of BIF includes all gradient pieces within $[0, \Gamma_t]$, where $\Gamma_t$ is the sum of clipping thresholds of data in $B_t$. The transformation from TIF to BIF is shown in Fig. 4 and BIF is formally defined in App. C.2.3. By integrating the importance scores of all gradient pieces from a gradient $\mathbf{g}_{\pi_t(k)}$ ranked $k$-th in the batch, INO-SGD calculates the average importance score $\rho_k$ of this gradient. The norm of this gradient is then restricted to $\rho_k C_{o(\pi_t(k))}$ (an easy way as shown in Alg. 1 is to simply scale the clipped gradient by $\rho_k$). In the end, steps similar to IDP-SGD are performed: summing up the clipped gradients (weighted by importance scores), adding noise to the summed gradient, estimating the average gradient[3] and performing gradient descent. A detailed pseudocode of INO-SGD is given in Alg. 1. We highlight that the runtime of INO-SGD is almost the same as IDP-SGD (App. C.2.4) as the time taken for backpropagation (Line 10) dominates the time needed for the additional sorting and integration (Line 5 and 8).

---

**Algorithm 1 The INO-SGD algorithm.**

**Input:** Each owner $n \in [N]$'s datasets $D_n$, sampling rates $q_n$, clipping thresholds $C_n$; initial model parameters $\boldsymbol{\theta}_0$; noise scale $\sigma$; TIF $f_{\text{tail}}$.
**Output:** Trained model parameters $\boldsymbol{\theta}_T$.
1: $B_t \leftarrow \emptyset; b \leftarrow \sum_{n \in [N]} q_n |D_n|$
2: **for** iteration $t \in [T]$ **do**
3:    $B_t \leftarrow \bigcup_n$ subset sampled from $D_n$ w.p. $q_n$
4:    $f_t \leftarrow$ BIF computed from $f_{\text{tail}}$ and $B_t$
5:    $B_t^{\text{s}} \leftarrow B_t$ sorted by order of losses
6:    $\pi_t \leftarrow$ function mapping $k \in [|B_t|]$ to index of $B_t^{\text{s}}$'s $k$-th datum in $B_t$
7:    $(c_k)_{k \in [|B_t|]} \leftarrow$ cumulative sums of first $k$ data's clipping thresholds in $B_t^{\text{s}}$
8:    $(\rho_k)_{k \in [|B_t|]} \leftarrow$ average importance scores $\left( \int_{c_{k-1}}^{c_k} f_t / (c_k - c_{k-1}) \; \mathrm{d}c \right)_{k \in [|B_t|]}$
9:    **for** datum $d$ in $B_t$ **do**
10:      $\mathbf{g}_d \leftarrow \nabla_{\boldsymbol{\theta}} \ell(\boldsymbol{\theta}_{t-1}; d)$
11:      $\bar{\mathbf{g}}_d \leftarrow \mathbf{g}_d / \max\{1, \|\mathbf{g}_d\| / C_{o(d)}\}$
12:    **end for**
13:    $\overline{\mathbf{G}}_t \leftarrow \sum_{k=1}^{|B_t|} \rho_k \bar{\mathbf{g}}_{\pi_t(k)}$
14:    $\widetilde{\mathbf{G}}_t \leftarrow \left( \overline{\mathbf{G}}_t + \mathcal{N}(\mathbf{0}_r, \sigma^2 \mathbf{I}_{r \times r}) \right) / b$
15:    $\boldsymbol{\theta}_t \leftarrow \boldsymbol{\theta}_{t-1} - \eta \widetilde{\mathbf{G}}_t$
16: **end for**
17: **return** updated model parameters $\boldsymbol{\theta}_T$

---

[2]Our algorithm and analysis also work with other sorting order. See App. C.2.2 for discussion.
[3]Using the expected batch size $b := \sum_{n \in [N]} q_n |D_n|$ instead of the actual $|B_t|$ avoids extra privacy loss.

**Choice of TIF.** To flexibly control the TIF shape via a few parameters, we recommend using a Beta distribution $\text{Beta}(\alpha, \beta)$ to model the model owner's belief regarding the fraction of important data. A larger $\alpha$ would increase the weights of the more important gradients within the tail while a larger $\beta$ should reduce the weights of the less important gradients within the tail. The value of a horizontal flip of Beta c.d.f. $f_{\text{tail}}(c) \coloneqq \int_0^{1-c/\gamma} x^{\alpha-1}(1-x)^{\beta-1}\, \mathrm{d}x / \int_0^1 x^{\alpha-1}(1-x)^{\beta-1}\, \mathrm{d}x$ at a point represents its importance based on the belief. In App. C.2.5, we include a guideline on how to tune the above hyperparameters in practice. More generally, one can interpret TIF as a *survival function* (David & Mitchel, 2012) that is specially defined on a finite range. The model owner can also use other functions they deem interpretable (we give an example in App. C.2.5).

## 3.3 THEORETICAL ANALYSIS OF INO-SGD

### 3.3.1 PRIVACY GUARANTEE OF INO-SGD

**Theorem 3.3** (Privacy of INO-SGD)**.** *For any TIF $f_{\text{tail}}$ and the corresponding BIF $f_t$ at each iteration $t \in [T]$, INO-SGD satisfies $(\boldsymbol{\alpha}, \bar{\epsilon})$-IRDP, where $\bar{\epsilon}_n = 2T\alpha_n C_n^2 q_n^2/\sigma^2$.*

To prove Thm. 3.3 (App. C.3.1), it is crucial to show that for any datum $d$, the modular sensitivity $\Delta_{\mathcal{A}}^d$ at iteration $t$ is bounded by $d$'s clipping threshold $C_{o(d)}$. When $d$ is added to batch $B_t$, the impact of less important data (with loss smaller than $d$) will not change due to the fixed-length tail of BIF. The importance score of more important data may increase (as they are "pushed" to the left by $C_{o(d)}$). However, we ensure the total change in the summed gradients and $d$'s clipped gradient is bounded by $C_{o(d)}$. Our proof makes use of the triangle inequality, the bound on each gradient's $L^2$-norm, and the telescoping sum created by our BIFs. Notably, as INO-SGD consumes privacy budgets **at the same rate** as IDP-SGD, we are not sacrificing number of iterations for better learning dynamics. In App. D.4.5, we empirically validate the privacy guarantee of INO-SGD by showing that the *LiRA membership inference attack* (Carlini et al., 2022) cannot effectively distinguish which data are used for training.

*Remark.* The use of order of loss in each batch (e.g., sorting) does not incur extra privacy costs, as such order at iteration $t$ comes from the ML model obtained at iteration $t-1$, which is already protected by (I)DP in earlier rounds (i.e., similar to the gradients we obtain at iteration $t$ for each sampled datum). Thus, although INO-SGD uses the gradients and the order of loss, as we bound its (modular) sensitivity at each iteration $t$, it strictly satisfies IDP by adaptive composition.

**INO-SGM mechanism.** We generalize INO-SGD to a general IDP mechanism based on the *sampled Gaussian mechanism* (SGM) in App. B.1.3 as follows (proven and discussed in App. C.3.2):

**Theorem 3.4** (INO-SGM, informal)**.** *Let $g$ be a function mapping an arbitrary set of data $\mathcal{B}$ to $\mathbb{R}^r$ such that $\|g(d)\|$ is upper bounded by $\Delta_{\mathcal{A}}^{o(d)}$. Let $\varpi$ be a score-based function mapping $\mathcal{B}$ to some order. Suppose $B \coloneqq \{d : d \text{ is sampled with probability } q_{o(d)}\}$. Then releasing $\sum_{k=1}^{|B|} \rho_k g(\varpi(B)(k)) + \mathcal{N}(\mathbf{0}_r, \sigma^2 \mathbf{I}_{r \times r})$ satisfies IRDP, where $\rho_k$ is set in the same way as in BIF.*

### 3.3.2 INO-SGD ADDRESSES IDP-INDUCED UTILITY IMBALANCE

When INO-SGD strategically downweights data with lower losses, it mitigates MID for more private owners with higher losses. This can be seen by applying Thm. 3.1 with the owners with higher and lower loss as owners 1 and 2, respectively. When $\cos(\angle_t)$ is negative, INO-SGD effectively decreases the numerator, increases the left-hand side of Cond. 2, potentially beyond $\varsigma$. Now we analyze the **optimization objective** of INO-SGD. The cumulative clipping thresholds $(c_k)_{k \in [|B_t|]}$ in Line 8 of Alg. 1 fall in different positions across iterations, which complicates the theoretical analysis. Thus, we analyze a *uniform-privacy-extension* (UPE) dataset $\widehat{D}$ instead. In $\widehat{D}$, each owner may have different number of data based on their original sampling rates and clipping thresholds ($\widehat{D}$ is imbalanced). To illustrate, consider the SAMPLE algorithm with different sampling rates[4]. Suppose we sample multiple subsets $(B_\tau)_{\tau \in [\mathcal{T}]}$ for a large $\mathcal{T}$ according to sampling rates $(q_n)_{n \in [N]}$. We construct the UPE dataset $\widehat{D} \coloneqq \bigcup_{\tau=1}^{\mathcal{T}} B_\tau$. Running INO-SGD on $\widehat{D}$ with all sampling rates $= 1/\mathcal{T}$ is approximately equivalent to running INO-SGD on $D$ as the sampled batches are similar.

The vanilla IDP-SGD is effectively optimizing $\mathcal{L}(\boldsymbol{\theta}; \widehat{D})$. Since $\widehat{D}$ is imbalanced, IDP-SGD suffers from MID and BOO as discussed in Sec. 2.2. In contrast, our INO-SGD is **less affected by MID**

---

[4]Analysis for SCALE is similar.

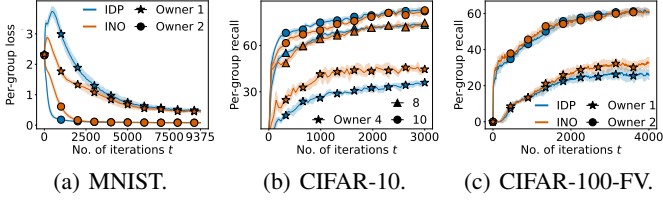

(a) MNIST.  (b) CIFAR-10.  (c) CIFAR-100-FV.

Figure 5: **Per-group utility corresponding to different owners for IDP-SGD and INO-SGD.** INO-SGD significantly improves model utility ($\sim 10\%$ accuracy) for the more private owners without lowering the utility for the less private owners.

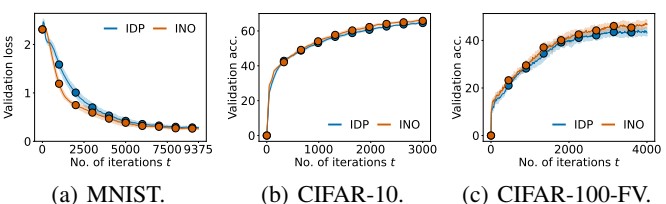

(a) MNIST.  (b) CIFAR-10.  (c) CIFAR-100-FV.

Figure 7: **Comparison of overall model utility between IDP-SGD and INO-SGD.** INO-SGD consistently improves/preserves the overall model performance.

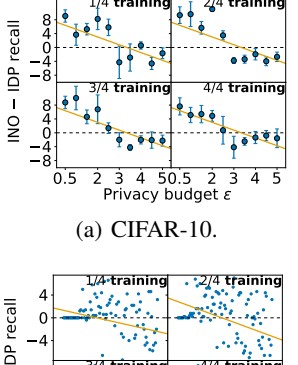

(a) CIFAR-10.

(b) CIFAR-100.

Figure 6: **INO-SGD's per-group recall minus IDP-SGD's.** Model utilities for the more private owners show higher increase.

**and BOO** as it effectively optimizes a different objective $\mathcal{L}_{f_{\text{tail}}}$ (see App. C.3.3 for verification that the gradient at each iteration is an unbiased estimate of derivative of $\mathcal{L}_{f_{\text{tail}}}$):

**Theorem 3.5** (Objective of INO-SGD). INO-SGD *specified by TIF $f_{\text{tail}}$ effectively minimizes*

$$\mathcal{L}_{f_{\text{tail}}}(\boldsymbol{\theta}; \widehat{D}) = \frac{1}{K} \sum_{k=1}^{K} w_k \ell(\boldsymbol{\theta}; d_{\pi_K(k)}),$$

*where $K$ is the number of data in $\widehat{D}$ and $\mathbf{w} = (w_k)_{k=1}^{K}$ is a non-increasing sequence of weighting coefficients less than 1 and $\pi_K(k)$ represents the index of datum with $k$-th largest loss in $\widehat{D}$.*

As the weight sequence is non-increasing, a smaller $k$ assigns a larger weight $w_k$ to the $k$-th largest loss in $\widehat{D}$. Intuitively, data that are harder to learn (e.g., belonging to more private owners/near the decision boundary) are assigned larger weights than easier ones (e.g., belonging to less private owners). Hence, **the more private groups have a larger weight in the objective, thus mitigating MID and BOO.** The INO-SGD objective $\mathcal{L}_{f_{\text{tail}}}$ can also be viewed as (see App. C.3.4 for details)

- the *mixed conditional value-at-risk* (Ogryczak, 2000) of loss $\ell$ (i.e., a weighted expectation of worse-case losses). Thus, data from more private owners with worse/larger losses are improved.
- an objective that better corrects for utility imbalance than average loss when minimized and is less affected by outliers with large loss than the *maximal loss* (Shalev-Shwartz & Wexler, 2016).
- an *ordered weighted average* (OWA) function with positive non-increasing weights. Weymark (1981) shows that such functions satisfy the *Pigou-Dalton principle* (Pigou, 1912; Dalton, 1920): if the model can reduce a larger individual loss (likely from a more private owner) by increasing another smaller individual loss (likely from a less private owner), the OWA loss decreases. Thus, minimizing the OWA loss reduces imbalance across owners with different IDP requirements.

## 4 EXPERIMENTS

In this section, we empirically compare INO-SGD with IDP-SGD under a variety of settings. App. D.1 describes our setup (e.g., data owners and their privacy budgets) in detail. To summarize[5], we train Papernot et al. (2021)'s convolutional neural network models for better performance under (I)DP on the MNIST (LeCun et al., 1998), CIFAR-10 or CIFAR-100 (Krizhevsky & Hinton, 2009) dataset. Each data owner holds data from 1 (CIFAR-10, CIFAR-100) or multiple distinct classes

---

[5]App. D also contains additional experiments under more datasets, models and settings.

(MNIST) with no overlap. Additionally, for CIFAR-100-FV, we consider a subset of CIFAR-100 with 2 data owners each possessing a domain of data FISH and VEHICLE respectively. In all experiments, owners with smaller index require stronger privacy. We focus on two questions: (**B**alance) *Does* INO-SGD *improve model performance on the worse-off validation groups (measured by per-group loss/recall[6]) corresponding to the more private data owners during training and reduce the performance gap between the more private and less private groups throughout training?* (**U**tility) *Since balance often comes at a cost, does* INO-SGD *achieve the above by sacrificing overall model utility (measured by validation loss/accuracy)?* We answer (**B**) in Sec. 4.1 and (**U**) in Sec. 4.2. Several other desirable properties of INO-SGD are discussed in Sec. 4.3. Our implementation can be found at `https://github.com/snoidetx/ino-sgd`.

## 4.1 ADDRESSING UTILITY IMBALANCE

In Fig. 5, we plot the per-group loss/recall corresponding to each data owner across iterations for IDP-SGD and INO-SGD. **For groups corresponding to the more private owners (e.g., ★), INO-SGD gives consistently better performance throughout learning**: the INO-SGD performance starts to increase earlier and faster than IDP-SGD while ending with better performance when training completes. We stress that **MID occurs at different time based on ML task complexity**. In simpler tasks like MNIST, MID occurs at around $1/4$ of the entire training stage, while in more difficult tasks like CIFAR-10, privacy budgets are used up before MID even ends. This aligns with our emphasis that **MID is more significant under (I)DP**. App. D.2 includes results for more datasets (e.g., real-world clinical datasets), models (e.g., ResNet-18 (He et al., 2016)) and settings (e.g., more variation in privacy budgets; less heterogeneity among owners; presence of label noises). Moreover, it also includes additional results that show how INO-SGD improves the worst-group utility and hence further validate that INO-SGD addresses utility imbalance in Fig. 12.

To investigate how model performances under INO-SGD change from IDP-SGD for data owners with different privacy budgets, we consider the CIFAR-10 and CIFAR-100 datasets where we have many data owners with various privacy budgets. For both datasets, we plot graphs of changes in model performance against privacy budgets to show INO-SGD's effectiveness for different privacy budgets in Fig. 6. We also plot their linear regression line in orange. **Across training stages, INO-SGD significantly increases the recall on validation data groups associated with more private owners (left) at the expense of a moderate decrease in the recall on less private ones (right).** By the end of training, both CIFAR-10 and CIFAR-100 datasets show statistically significant *Pearson correlations* between the changes in recall and privacy budgets ($p$-value $< 0.001$ in both cases). In particular, the scatter plot of CIFAR-100 appears noisier because the complexity of each class in CIFAR-100 varies and has higher variance than CIFAR-10. A more private data owner with easier-to-learn data might attain similar utility as another less private data owner with difficult data.

## 4.2 PRESERVING OVERALL MODEL UTILITY

Although the main goal of INO-SGD is to correct utility imbalance, Fig. 7 demonstrates that INO-SGD also results in equal or slightly better overall (aggregated) model utility. For all experiments, **INO-SGD generally achieves a smaller loss/higher accuracy than IDP-SGD throughout training**. This verifies our intuition in Sec. 3.2 that by dropping certain information from the sampled gradients, we are not wasting privacy budgets but rather deleting information that negatively affects model performance. The results also imply that vanilla IDP-SGD leads to a suboptimal privacy-utility tradeoff, while INO-SGD improves over it. In App. D.3, we include more results using larger datasets and models under various settings and observe the same trend.

## 4.3 OTHER EMPIRICAL BENEFITS AND DISCUSSIONS

- **Better within-owner learning dynamics:** In real life, a data owner itself may own data from multiple heterogeneous groups. Such groups may have different sizes and learning complexities, which cause utility imbalance to occur within the owner itself. In App. D.4.1, we demonstrate that INO-SGD addresses this too and results in better within-owner learning dynamics.

---

[6]We include some results based on per-group/overall loss since we analyze and solve the problem theoretically through the lens of loss. The recall/accuracy results are provided for all experiments in App. D.

- **Robustness to hyperparameter choice:** In App. D.4.2, our **ablation studies** show that INO-SGD performance is better than IDP-SGD across different choices of the TIF $f_{\text{tail}}$, including the tail length $\gamma$, specific values of $\alpha$ and $\beta$ when we construct $f_{\text{tail}}$ using a Beta distribution, or more generally the specific forms of TIF. This shows the practicability of INO-SGD.
- **Pareto superiority**: The model owner can also choose to set TIF more aggressively to trade some overall utility for less utility imbalance. In App. D.4.3, we demonstrate that INO-SGD Pareto-dominates simple methods (that satisfy IDP) in such cases.

## 5  CONCLUSION AND DISCUSSION

In this work, we identify and analyze the problem of IDP-induced utility imbalance and explain why it differs from standard data imbalance. We then propose the INO-SGD algorithm and theoretically analyze its properties. Lastly, we empirically show that INO-SGD addresses the utility imbalance problem on a range of different datasets. One possible limitation (discussed in App. E.1) is that increasing the utility of more private owners might reduce that of the less private ones. While this could be explained by the *IDP-balance-utility tradeoff* (see below and App. E.2), future work can consider exploring the Pareto frontier characterizing this tradeoff or other methods to address IDP-induced utility imbalance. Future work can also explore other use cases of our proposed unique IDP mechanism, INO-SGM. In App. F, we answer other questions a reader may have.

**IDP-balance-utility tradeoff.**   The disparate impact of DP and the *privacy-disparity-utility trade-off*[7] (Bagdasaryan et al., 2019) has been a well-known drawback that limits the performance of DP-ML models. In this work, we identify that the situation becomes even worse under IDP. In Sec. 3.1, we explain that even if the original training set is balanced and there is no underrepresented group, IDP(-SGD) would forcefully create underrepresented groups because of the individualized privacy preferences. We term this the IDP-balance-utility tradeoff. **Our work successfully gets closer to the boundary of the IDP-balance-utility tradeoff**, because we are able to achieve better balance without losing the overall model utility. Nonetheless, we are still facing the IDP-balance-utility tradeoff. More discussions are included in App. E.2. Future research may theoretically characterize this tradeoff or come up with new methods that tackle it.

---

[7]As mentioned in Sec. 3.1, these are orthogonal to this work.

ETHICS STATEMENT

This paper presents work whose goal is to advance the field of machine learning (ML). In particular, it identifies and addresses the problem of IDP-induced utility imbalance where the trained ML model has lower utilities on certain groups during deployment. This problem is justified and significant as the model owner and subsequent users with data in such groups who are impacted by the model decisions are negatively affected. Moreover, our work emphasizes data privacy, which protects the privacy of individuals. Therefore, our research promotes ethical AI with positive societal impact.

REPRODUCIBILITY STATEMENT

In App. D.1, we describe in detail the devices, datasets, models and settings used in all our experiments. In each experiment subsection, we also clearly explain how each experiment is done. We also provide our source code in the supplementary materials which includes the necessary information for reproducibility.

ACKNOWLEDGMENTS

This research is supported by the National Research Foundation, Singapore under its AI Singapore Programme (AISG Award No: AISG3-RP-2022-029). Xiao Tian and Jue Fan are supported by Agency for Science, Technology and Research (A*STAR) Graduate Academy. The authors would also like to thank the anonymous reviewers and AC for their helpful feedback. The authors would also extend special thanks to Prof. Jonathan Scarlett for his valuable comments and suggestions at the earlier stage of this research.

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

# A OVERVIEW

## A.1 SUMMARY OF THIS WORK

The following figure gives a summary of this work.

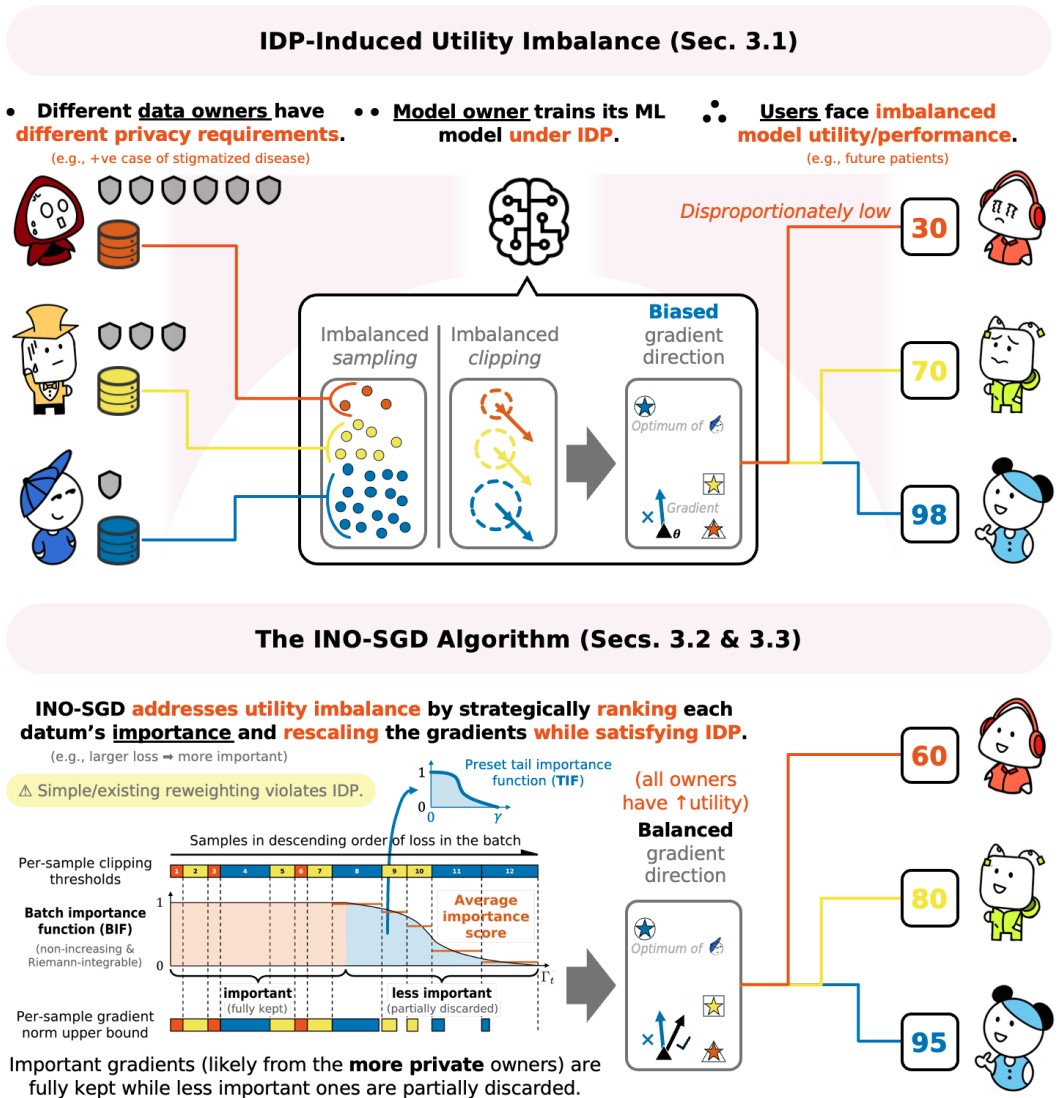

Figure 8: **Overview of IDP-induced utility imbalance and the INO-SGD algorithm.**

## A.2 SUMMARY OF NOTATIONS

Below gives a summary of all notations used in this paper.

**Constants and Variables**

| | |
|---|---|
| $\angle_t$ | Angle between group summed gradients at iteration $t$ |
| $\emptyset$ | Empty set |
| $\mathbf{0}_r$ | $r$-length vector of 0's |
| $\mathcal{A}$ | Algorithm; ML training algorithm |
| $\alpha$ | Rényi order |
| $\boldsymbol{\alpha}$ | Vector of per-owner Rényi orders |
| $\alpha_n$ | Rényi order computed for data owner $n$ |
| $\alpha$ | Beta distribution parameter |
| $B$ | Sampled batch from $D$ |
| $B_t$ | Sampled batch at iteration $t$ |
| $B_{t,n}$ | Sampled batch at iteration $t$ from data owner $n$ |
| $B_t^s$ | sorted $B_t$ based on descending order of individual losses |
| $\mathcal{B}$ | Arbitrary set of data |
| $b$ | Expected batch size |
| $\beta$ | Beta distribution parameter |
| $C$ | Clipping threshold |
| $c$ | Point over the tail of length $\gamma$ |
| $\mathbf{c}$ | Vector of accumulated clipping threshold |
| $c_k$ | Accumulated clipping threshold until (and including) the $k$-th ranked datum |
| $C_n$ | Clipping threshold of data owner $n$ |
| $D$ | Dataset |
| $\widehat{D}$ | UPE dataset associated with $D$ |
| $D^d$ | Neighboring dataset of $D$ with regards to datum $d$ |
| $D_n$ | Dataset of data owner $n$ |
| $\widehat{D}_n$ | UPE dataset of data owner $n$ |
| $\mathcal{D}$ | Set of datasets |
| $e$ | A random event |
| $d$ | Datum within a dataset |
| $\delta$ | Tolerance term in $(\epsilon, \delta)$-DP |
| $\epsilon$ | Privacy budget of a data owner; privacy level of a mechanism |
| $\boldsymbol{\epsilon}$ | Vector of per-owner privacy budgets |
| $\epsilon_n$ | Privacy budget of data owner $n$; |
| $\bar{\epsilon}$ | Rényi Privacy budget of a data owner |
| $\bar{\boldsymbol{\epsilon}}$ | Vector of Rényi privacy budgets of data owners |
| $\bar{\epsilon}_n$ | Rényi Privacy budget of data owner $n$ |
| $\eta$ | Learning rate |
| $F_g$ | Group fairness metric |
| $\mathcal{F}_t$ | Filtration up to the end of iteration $t$ |
| $\mathbf{G}_t$ | Gradient used for descent at iteration $t$ |
| $\mathbf{G}_{t,g_n}$ | Summed gradient of group $g_n$ at iteration $t$ |
| $\overline{\mathbf{G}}_t$ | Sum of (weighted) clipped gradient at iteration $t$ |
| $\widetilde{\mathbf{G}}_t$ | Perturbed gradient at iteration $t$ |
| $g_n$ | Group $n$ within a dataset |
| $\mathbf{g}_d$ | Individual gradient with regards to datum $d$ |
| $\mathbf{g}_{t,d}$ | Individual gradient with regards to datum $d$ at iteration $t$ |
| $\bar{\mathbf{g}}_d$ | Clipped individual gradient with regards to datum $d$ |
| $\bar{\mathbf{g}}_{t,d}$ | Clipped individual gradient with regards to datum $d$ at iteration $t$ |
| $\Gamma_t$ | Sum of clipping thresholds of sampled data at iteration $t$ |
| $\gamma$ | Length of tail |
| $\mathbf{I}_{r \times r}$ | $r \times r$ identity matrix |
| $K$ | Number of data in $\widehat{D}$ |

| | |
|---|---|
| $k$ | Index of datum in $\widehat{D}$ |
| $\kappa$ | Number used in fast integration in INO-SGD |
| $L_1$ | Lipschitz constant |
| $L^2$ | Quantities measured in 2D Lebesgue measure |
| $L_2$ | Smoothness constant |
| $\lambda$ | Level of CVaR |
| $N$ | Number of data owners |
| $n$ | Index of data owner |
| $\mathbf{O}$ | Subset of $\boldsymbol{\Theta}$ |
| $\boldsymbol{\Omega}$ | Set of algorithm outputs |
| $\boldsymbol{\omega}$ | Vector algorithm outputs |
| $\mathcal{P}$ | Uncertainty set |
| $\mathbf{p}$ | Possible weights within uncertainty set |
| $p_k$ | Possible weight assigned to $k$-th ranked datum |
| $\phi_{t,d}$ | Scaling factor to scale gradient of datum $d$ to its clipped gradient at iteration $t$ |
| $\bar{\phi}_{t,n}$ | Expected scaling factor of owner $n$ at iteration $t$ |
| $\bar{\phi}_{t,n}^{(2)}$ | Expected squared scaling factor of owner $n$ at iteration $t$ |
| $\psi$ | Problem-specific positive term used in Cond. 12 |
| $q$ | Sampling rate |
| $\mathbf{q}$ | Vector of individualized sampling rates |
| $\hat{q}$ | Uniform sampling rate in UPE dataset |
| $q_n$ | Sampling rate of data owner $n$ |
| $r$ | Number of parameters in an ML model |
| $\rho$ | Importance score |
| $\rho^d$ | Importance score of datum $d$ |
| $\rho_k$ | Importance score of the $k$-th ranked data |
| $s$ | Scoring function in INO-SGM |
| $\sigma$ | Noise scale |
| $\varsigma$ | Problem-specific positive term used in Cond. 5 |
| $\varsigma_n$ | Problem-specific positive term used in Eq. 13 and 14 |
| $T$ | Total number of iterations |
| $t$ | Index of iteration |
| $\mathcal{T}$ | Total number of sampling iterations to contruct $\widehat{D}$ |
| $\tau$ | Index of sampling iteration to construct $\widehat{D}$ |
| $\boldsymbol{\Theta}$ | Set of possible model parameters |
| $\boldsymbol{\theta}$ | Vector of model parameters |
| $\boldsymbol{\theta}^*$ | Optimal model parameter for ERM |
| $\boldsymbol{\theta}_t$ | Model parameters after iteration $t$ |
| $u$ | Possible value of $x$s |
| $v$ | Possible value of $x$s |
| $\mathbf{w}$ | Vector of weights in INO-SGD objective |
| $w_k$ | Weight associated with $k$-th ranked datum in INO-SGD objective |
| $\mathcal{X}_{\mathrm{s}}$ | Set of possible values of $x_{\mathrm{s}}$ |
| $\mathbf{x}$ | Unlabelled datum |
| $x_{\mathrm{s}}$ | A sensitive attribute |
| $\xi_n$ | Uniform bound for stochastic gradient from owner $n$ |
| $y$ | Label of data |
| $\hat{y}$ | Model's predicted label of data |
| $\mathbf{Z}_t$ | Gaussian noise added at iteration $t$ |

### Functions and Operators

| | |
|---|---|
| $\triangle$ | Symmetric difference between two sets |
| $\square$ | An arbitrary function term |
| $\boxed{+}$ | An arbitrary non-negative function term |
| $|\cdot|$ | Cardinality of set $\cdot$ |
| $\|\cdot\|$ | $L^2$-norm of $\cdot$ |

| | |
|---|---|
| $[\cdot]$ | Set $\{1, 2, \cdots, \cdot\}$ |
| $\Delta(\cdot)$ | Sensitivity of $\cdot$ |
| $\Delta_\cdot^\circ$ | Modular sensitivity of $\cdot$ with regards to datum $\circ$ |
| $\mathbb{E}[\cdot]$ | Expectation of $\cdot$ |
| $f_{\text{tail}}(\cdot)$ | Tail importance function |
| $f_t(\cdot)$ | Batch importance function at iteration $t$ |
| $o(\cdot)$ | Owner of $\cdot$; group membership function |
| $p_{\mathcal{A}}^D(\cdot)$ | Output distribution returned by algorithm $\mathcal{A}$ when taking in dataset $D$ |
| $\Pr[\cdot]$ | Probability of $\cdot$ |
| $\text{sort}(\cdot)$ | Sorted version of $\cdot$ in descending order |
| $\text{Beta}(\cdot, \circ)$ | Beta distribution |
| $\mathfrak{D}_\alpha(\cdot \| \circ)$ | Rényi divergence of order $\alpha$ between $\cdot$ and $\circ$ |
| $\mathfrak{D}_{\mathbf{x}}(\cdot, \circ)$ | Distance between $\cdot$ and $\circ$ over the domain of $\mathbf{x}$ |
| $\mathfrak{D}_y(\cdot, \circ)$ | Distance between $\cdot$ and $\circ$ over the domain of $y$ |
| $\mathcal{L}(\cdot; \circ)$ | Global loss function with parameter $\cdot$ and dataset $\circ$ |
| $\widehat{\mathcal{L}}(\cdot; \circ)$ | Global loss function in the UPE dataset |
| $\mathcal{L}_{f_{\text{tail}}}(\cdot; \circ)$ | INO-SGD objective with parameter $\cdot$ and dataset $\circ$ |
| $\mathcal{L}_\circ(\cdot)$ | Owner $\circ$'s loss function with parameter $\cdot$ |
| $\ell(\cdot; \circ)$ | Individual loss function with parameter $\cdot$ and datum $\circ$ |
| $\ell_k(\cdot)$ | Individual loss function with parameter $\cdot$ and the $k$-th datum |
| $\max\{\cdot, \circ\}$ | Larger value between $\cdot$ and $\circ$ |
| $\mathcal{N}(\cdot, \circ)$ | Gaussian distribution of mean $\cdot$ and variance $\circ$ |
| $\nabla_\circ \cdot$ | Derivative of $\cdot$ with regards to $\circ$ |
| $\pi(\cdot)$ | Index of datum whose loss is ranked $\cdot$ in the whole dataset |
| $\pi_t(\cdot)$ | Index of datum whose loss is ranked $\cdot$ in the batch at iteration $t$ |
| $\varpi(\cdot)$ | Function that maps a set of data to their order |

## A.3 USE CASES

In this section, we explain why this work is important in real life. **It is common for different groups or data owners to have different privacy preferences.** Data owners with certain attribute values/class labels (e.g., stigmatized diseases, race) could be more disadvantaged than others when data leakage occurs out of privacy attack, as this may bring them discrimination or denial of services (Best & Arseniev-Koehler, 2023; Lai & Tanner, 2022). Since DP is highly relevant in such fields to protect their privacy, our work's use cases are broad. For example, a hospital located at the center of a disease outbreak will have much more positive cases and require stronger privacy than a hospital located far away to protect their patients. Similarly, students from marginalized backgrounds may prefer stronger privacy; social media users discussing more sensitive topics may desire stronger privacy on their profiles; individuals with poor credit histories may want stronger privacy to avoid unnecessary loss of opportunities.

**Should model utility on more private data owners' data be lower? Our answer is no.** If the trained ML model has lower utility on some group of data within the data space, the model owner and subsequent users with data from such group will be negatively affected. For example, consider the case where a medical institute trains a disease model for a stigmatized disease. Just because positive-case data owners have stronger privacy does not mean the trained model should perform badly on data from positive-case groups during deployment, since subsequent positive-case patients who are affected by the model's decisions will bear the consequences.

This work shows in Sec. 3.1 that IDP could induce severe utility imbalance between the more private data owners and the less private data owners. Because of the reasons above, **the goal of this work is to correct IDP-induced utility imbalance**, that is, to improve model utility on the worst-off groups corresponding to the more private data owners and reduce the gap between model utilities on groups corresponding to the more private data owners and the less private data owners. Our solution INO-SGD successfully achieves this without sacrificing overall model utility. As the first such work, our work makes a significant contribution. Note that improving the utilities of worst-off groups (during deployment) is also an important goal in fairness (Dullerud et al., 2022; Zhao & Gordon, 2022) and distributional robustness (Chen et al., 2023; Qiao et al., 2025).

**IDP-SGD assumes homogeneous data owners.** A naïve approach to address different privacy requirements is to always maintain the strongest DP level. However, this would inevitably lead to a loss in overall model utility (see App. F Q1 for a detailed explanation). The original IDP-SGD work (Boenisch et al., 2024) assumes that each data owner holds similar data. The different privacy preferences come from their natural personalities, where around $34\%$ of people prefer high privacy, $43\%$ prefer medium privacy and $23\%$ prefer low privacy (Alaggan et al., 2016). This setting is applicable to ML tasks where training data are in general not very sensitive such as emoji suggestion. In contrast, our work realizes that the nature of data could lead to different privacy preferences too and identifies that this may lead to a severe utility imbalance problem.

**Individualized privacy progression.** We also foresee another potential use case of our work and recognize it as *individualized privacy progression* (IPP). IPP captures the change of privacy preferences over time. Due to the privacy-utility tradeoff (Makhdoumi & Fawaz, 2013), owners who specify low privacy budgets (resulting in low-utility models) may regret their previous choices and wish to increase their privacy budgets. Conversely, owners who set high privacy budgets may become increasingly aware of or averse to the risks of privacy leakage and desire to lower their privacy budgets. When data owners wish to increase their privacy budgets, they may do so to varying degrees based on the sensitivity of their data. If data owners wish to decrease their privacy budgets (assuming that the model has not been attacked yet), the ML model needs to be rolled back to an earlier timestamp in the training process. This reinforces the importance of maintaining a good learning dynamics throughout the model training process.

# B ADDITIONAL BACKGROUND AND RELATED WORKS

## B.1 DIFFERENTIAL PRIVACY

In this section, we provide a detailed background on *differential privacy* (DP) for unfamiliar readers.

Let $\mathcal{D}$ be the set of datasets. Consider a dataset $D \in \mathcal{D}$ and an algorithm $\mathcal{A} : \mathcal{D} \to \mathbf{\Omega}$ that takes in the dataset $D$ and produces some useful output $\boldsymbol{\omega} = \mathcal{A}(D) \in \mathbf{\Omega}$. In the context of (parametric) machine learning (ML), algorithm $\mathcal{A}$ could refer to the training algorithm that outputs the vector of trained model parameters. If algorithm $\mathcal{A}$ is deterministic and always outputs the same $\boldsymbol{\omega}$ given fixed input dataset $D$, then attackers can reconstruct the input dataset $D$ by inverting algorithm $\mathcal{A}$ (e.g., through *membership inference attack* (Shokri et al., 2017) in ML), which threatens the privacy of data owners. Therefore, when dataset $D$ contains sensitive data, algorithm $\mathcal{A}$ is usually algorithmically designed to be randomized instead and *differential privacy* (Dwork, 2006) (DP) is employed to assess the privacy-preserving capability of $\mathcal{A}$.

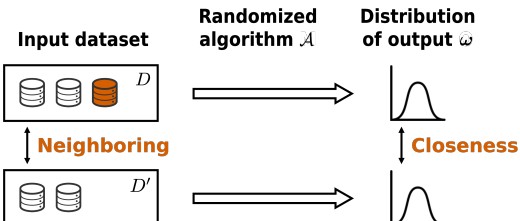

Figure 9: **Illustration of DP.** When a randomized algorithm $\mathcal{A}$ takes in a pair of neighboring datasets, we say $\mathcal{A}$ satisfies DP if its output distributions are always **close** to each other. The closeness of distributions is defined differently in different variants of DP.

DP works by comparing the distributions of randomized output $\boldsymbol{\theta}$ when algorithm $\mathcal{A}$ takes in a pair of *neighboring datasets* $D$ and $D^d$ that differ by only one datum $d$. Intuitively, if attackers cannot distinguish the distributions of $\mathcal{A}(D)$ and $\mathcal{A}(D^d)$, then they cannot reveal datum $d$ since they are not certain whether $d$ is even in the dataset. If all such pairs of neighboring datasets result in indistinguishable output distributions through $\mathcal{A}$, then all data are private. Therefore, the formal definition of (pure) DP follows:

**Definition B.1** ($\epsilon$-DP (Dwork, 2006))**.** A randomized algorithm $\mathcal{A} : \mathcal{D} \to \mathbf{\Omega}$ satisfies $\epsilon$-*differential privacy* ($\epsilon$-DP) if for any datum $d$, any pair of neighboring datasets $D, D^d \in \mathcal{D}$ such that $D \triangle D^d =$

$\{d\}$, and every possible output set $\mathbf{O} \subseteq \mathbf{\Omega}$, we have

$$\Pr[\mathcal{A}(D) \in \mathbf{O}] \le e^{\epsilon} \cdot \Pr[\mathcal{A}(D^d) \in \mathbf{O}], \tag{3}$$

where $\epsilon \ge 0$ is called the *privacy budget* of data owners or *privacy level* of algorithm $\mathcal{A}$, which measures how private the algorithm $\mathcal{A}$ is.

When privacy budget $\epsilon = 0$, the algorithm $\mathcal{A}$ is absolutely private. In other words, all pairs of neighboring datasets behave exactly the same through $\mathcal{A}$. Iteratively, they all behave the same as the empty set $\emptyset$. Thus, algorithm $\mathcal{A}$ is useless in this case. An example of such an algorithm would be $\forall D \in \mathcal{D} [\mathcal{A}(D) = \mathcal{N}(0,1)]$, which outputs a random Gaussian noise regardless of the input dataset. When privacy budget $\epsilon$ becomes larger, the distributions of $\mathcal{A}(D)$ and $\mathcal{A}(D^d)$ become less close and more distinguishable. The algorithm $\mathcal{A}$ becomes less private. On the other hand, this also means that the algorithm output $\boldsymbol{\theta}$ is impacted by the data to a larger extent, and hence the output should be more useful or informative. This is commonly known as the *privacy-utility tradeoff* (Makhdoumi & Fawaz, 2013).

Nevertheless, $\epsilon$-DP is often considered to be too strict in practice. This is because its definition requires a bounded logarithmic difference $\log \frac{\Pr[\mathcal{A}(D^d) \in \mathbf{O}]}{\Pr[\mathcal{A}(D) \in \mathbf{O}]}$ for any $\mathbf{O} \subseteq \mathbf{\Omega}$, although for some particular output $\boldsymbol{\omega} \in \mathbf{\Omega}$, it is almost impossible for algorithm $\mathcal{A}$ to output such $\boldsymbol{\omega}$ (i.e., $\Pr[\mathcal{A}(D) = \boldsymbol{\omega}] \approx 0$ and $\Pr[\mathcal{A}(D^d) = \boldsymbol{\omega}] \approx 0$). To make sure the bound $\epsilon$ holds for such edge cases, a great amount of extra perturbation and randomization needs to be added to algorithm $\mathcal{A}$, which damages its utility. This phenomenon is more severe in complex mechanisms such as model training in ML. To mitigate this issue, two common relaxations of $\epsilon$-DP are introduced below.

### B.1.1 $(\epsilon, \delta)$-DIFFERENTIAL PRIVACY

The most straightforward relaxation of $\epsilon$-DP is to add a tolerance term $\delta$ to the right-hand side of Eq. 3. Such relaxation is called $(\epsilon, \delta)$-*DP*, which aims to solve the edge case issue mentioned in the previous section by allowing a very small probability $\delta \in (0, 1)$ that the logarithmic difference bound $\epsilon$ is violated, i.e.,

**Definition B.2** ($(\epsilon, \delta)$-DP (Dwork et al., 2006)). A randomized algorithm $\mathcal{A} : \mathcal{D} \to \mathbf{\Omega}$ satisfies $(\epsilon, \delta)$-*differential privacy* ($(\epsilon, \delta)$-DP) if for any datum $d$, any pair of neighboring datasets $D, D^d \in \mathcal{D}$ such that $D \triangle D^d = \{d\}$, and every possible output set $\mathbf{O} \subseteq \mathbf{\Omega}$, we have

$$\Pr[\mathcal{A}(D) \in \mathbf{O}] \le e^{\epsilon} \cdot \Pr[\mathcal{A}(D^d) \in \mathbf{O}] + \delta,$$

where $\epsilon \ge 0$ is the *privacy budget* set by data owners or *privacy level* of algorithm $\mathcal{A}$ (a more private data owner sets a smaller $\epsilon$ for stronger privacy), and $\delta \in [0, 1)$ is a tolerance term which allows for a very small probability that the privacy level $\epsilon$ is violated.

While $(\epsilon, \delta)$-DP indeed gives better utility in most applications, its major drawback is that it is hard to compute the resulting $(\epsilon, \delta)$-DP level when two or more $(\epsilon, \delta)$-DP mechanisms are sequentially composed together[8] (i.e, $\mathcal{A}_2(\mathcal{A}_1(D))$). In fact, because of the existence of tolerance $\delta$, it is now intractable to obtain an exact $(\epsilon, \delta)$ bound for sequentially composed mechanisms, since both multiplication and addition operations appear in the definition. Therefore, for complex DP mechanisms such as ML model training, it is challenging to do *privacy accounting*, that is, to maintain a tight and accurate bound on the privacy consumption as training progresses. Existing techniques such as *moments accountant* (Abadi et al., 2016) provide relatively good accounting and maintain a relatively good model utility.

### B.1.2 RÉNYI DIFFERENTIAL PRIVACY

To make privacy accounting easier and more accurate, alternative variants of DP are proposed. One of such variants is based on the *Rényi divergence of order $\alpha$ (Van Erven & Harremos, 2014)*, $\mathfrak{D}_{\alpha}$, between the output distributions $p_{\mathcal{A}}^D$ and $p_{\mathcal{A}}^{D^d}$ corresponding to neighboring datasets $D$ and $D^d$. Specifically,

**Definition B.3** (Rényi Divergence (Van Erven & Harremos, 2014)). The *Rényi divergence of order $\alpha$* between distributions $p_{\mathcal{A}}^D$ and $p_{\mathcal{A}}^{D^d}$, $\mathfrak{D}_{\alpha}(p_{\mathcal{A}}^D \| p_{\mathcal{A}}^{D^d})$, is defined as follows:

$$\mathfrak{D}_{\alpha}(p_{\mathcal{A}}^D \| p_{\mathcal{A}}^{D^d}) := \frac{1}{\alpha - 1} \ln \int_{\mathbf{\Omega}} \Pr[\mathcal{A}(D^d) = \boldsymbol{\omega}] \cdot \left( \frac{\Pr[\mathcal{A}(D) = \boldsymbol{\omega}]}{\Pr[\mathcal{A}(D^d) = \boldsymbol{\omega}]} \right)^{\alpha} d\boldsymbol{\omega}. \tag{4}$$

---

[8]In comparison, for pure $\epsilon$-DP, we can simply sum up the exponents for composed DP mechanisms.

In particular, as $\alpha \to 1$, the above Rényi divergence approaches the Kullback–Leibler (KL) divergence. A smaller Rényi divergence between distributions $p_{\mathcal{A}}^{D}$ and $p_{\mathcal{A}}^{D^d}$ means that the two distributions are closer, thus indicating stronger privacy. Formally,

**Definition B.4** (($\alpha, \bar{\epsilon}$)-RDP (Mironov, 2017)). A randomized algorithm $\mathcal{A} : \mathcal{D} \to \mathbf{\Omega}$ satisfies ($\alpha, \bar{\epsilon}$)-*Rényi differential privacy* (($\alpha, \bar{\epsilon}$)-RDP) if for every pair of neighboring datasets $D, D^d \in \mathcal{D}$, we have

$$\mathfrak{D}_\alpha \left( p_{\mathcal{A}}^{D} \,\middle\|\, p_{\mathcal{A}}^{D^d} \right) \leq \bar{\epsilon},$$

where $\bar{\epsilon} \geq 0$ is the *Rényi privacy level* of algorithm $\mathcal{A}$.

The RDP bound of sequentially composed RDP mechanisms is simple and tight:

**Theorem B.5** (Sequential Composition of RDP (Mironov, 2017)). *Suppose mechanism $\mathcal{A}_1$ satisfies* ($\alpha, \bar{\epsilon}_1$)*-RDP and $\mathcal{A}_2$ satisfies* ($\alpha, \bar{\epsilon}_2$)*-RDP, then their sequential composition* ($\mathcal{A}_1(D), \mathcal{A}_2(D)$) *that releases both outputs satisfies* ($\alpha, \bar{\epsilon}_1 + \bar{\epsilon}_2$)*-RDP.*

Moreover, RDP algorithms satisfy ($\epsilon, \delta$)-DP via the following theorem:

**Theorem B.6** (Relation to ($\epsilon, \delta$)-DP (Balle et al., 2020b)). *A randomized algorithm $\mathcal{A}$ that satisfies* ($\alpha, \bar{\epsilon}$)*-Rényi DP also satisfies* ($\bar{\epsilon} + \log((\alpha - 1)/\alpha) - (\log \delta + \log \alpha)/(\alpha - 1), \delta$)*-DP for any $\delta \in (0, 1)$.*

For data owners, it is clearly easier for them to specify their privacy preferences using the ($\epsilon, \delta$)-DP notion because both $\epsilon$ and $\delta$ have a straightforward meaning. Specifically, $\epsilon$ is usually set to be larger than 0.1 and lower than 10, while $\delta$ is set to a small number (e.g., $10^{-5}$). Meanwhile, they may find it challenging to determine the appropriate value for Rényi order $\alpha$ and corresponding Rényi privacy budget $\bar{\epsilon}$. In constrast, model owners need to trace the privacy consumption when they train their ML models and thus ($\alpha, \bar{\epsilon}$)-RDP is more suitable for them. Since Thm. B.6 allows for the conversion between both notions, it becomes a common choice **for data owners to use** ($\epsilon, \delta$)**-DP to specify privacy preferences** and **for model owners to use** ($\alpha, \bar{\epsilon}$)**-RDP to perform privacy accounting** (known as *RDP accountant*).

### B.1.3 SAMPLED GAUSSIAN MECHANISM

The *sampled Gaussian mechanism* (SGM) is one of the most common DP mechanisms which converts any non-private algorithm $\mathcal{A}$ to one that satisfies DP. It has two components, the *noise addition mechanism* and the *sampling mechanism*.

**Noise addition mechanism.** The noise addition mechanism is the most common technique to achieve DP. It works by perturbing the non-private algorithm output $\boldsymbol{\omega}$ with certain noise of magnitude comparable to the difference in algorithm outputs given by neighboring datasets, such that the perturbed outputs become indistinguishable to attackers.

A quantity called *sensitivity* is used to represent the maximum difference in algorithm outputs given by neighboring datasets, which is defined formally below (we focus on the $L^2$ case which is the most practical):

**Definition B.7** (Sensitivity). The *($L^2$-)sensitivity* of an (non-private) algorithm $\mathcal{A}$ is defined as the $L^2$-norm of maximum change in mechanism outputs when it takes in a pair of neighboring datasets, that is,

$$\Delta_{\mathcal{A}} := \sup_{\substack{d \in D \\ D, D^d \in \mathcal{D} \\ D \triangle D^d = \{d\}}} \|\mathcal{A}(D) - \mathcal{A}(D^d)\|,$$

where $D, D^d$ are neighboring datasets.

*Clipping* (Abadi et al., 2016) is commonly used to control the $L^2$-sensitivity of an algorithm $\mathcal{A}$. For a vector $\boldsymbol{\omega}$ and a predetermined *clipping threshold* $C > 0$, the clipped vector is

$$\bar{\boldsymbol{\omega}} := \frac{\boldsymbol{\omega}}{\max\{1, \|\boldsymbol{\omega}\|/C\}},$$

where we scale down $\boldsymbol{\omega}$ if its $L^2$-norm exceeds $C$ and do nothing otherwise. For example, given $C = 5$, the vector $\binom{6}{8}$ will be clipped to $\binom{3}{4}$. Another vector $\binom{2}{2}$ will not be clipped since its norm

is already less than $C = 5$. By returning the clipped output for mechanism $\mathcal{A}$, we can limit its sensitivity to at most $C$ when we add a datum to or remove a datum from the input dataset $D$.

After computing or controlling the sensitivity of algorithm $\mathcal{A}$, model owners can add noise of corresponding scale to achieve different level of privacy. The most practical way is to add Gaussian noise and the privacy guarantee is given below:

**Theorem B.8** (Gaussian Mechanism (Dwork et al., 2006))**.** *For any algorithm $\mathcal{A}$ with output dimensionality $r$, privacy budget $\epsilon$ and tolerance $\delta \in (0, 1)$, adding a Gaussian noise of $\mathbf{0}_r$ ($r$-length vector of $0$'s) mean and $\sigma^2 \mathbf{I}_{r \times r}$ ($r \times r$ identity matrix) variance to the output achieves $(\epsilon, \delta)$-DP, where*

$$\sigma = \frac{\Delta_{\mathcal{A}} \sqrt{2 \log(1.25/\delta)}}{\epsilon}.$$

**Sampling mechanism.** Privacy can be amplified by *sampling* (Balle et al., 2020a). Suppose instead that we sample a subset $B \subseteq D$ by independently sampling each datum in $D$ with probability $q$ (i.e., *Poisson sampling*). We call $q$ the *sampling rate*. Privacy is amplified through Poisson sampling because for any neighboring datasets, the datum by which they differ has probability $1 - q$ of not being sampled, in which case the mechanism outputs on neighboring datasets will be exactly the same.

Combining the Gaussian mechanism and sampling mechanism together, we have the *sampled Gaussian mechanism* (SGM), which refers to the process where we first sample a subset $B \subseteq D$ using Poisson sampling, pass it into algorithm $\mathcal{A}$, clip the output to clipping threshold $C$ and perturb the output with Gaussian noise. SGM achieves the following privacy guarantee:

**Theorem B.9** (Privacy of SGM (Mironov et al., 2019))**.** *An SGM with sampling rate $q$, sensitivity $\Delta_{\mathcal{A}}$ and noise scale $\sigma$ satisfies $(\alpha, \bar{\epsilon})$-RDP, where*

$$\bar{\epsilon} = \frac{2\alpha q^2 \Delta_{\mathcal{A}}^2}{\sigma^2}.$$

### B.1.4 DIFFERENTIALLY PRIVATE STOCHASTIC GRADIENT DESCENT

Based on SGM, *differentially private stochastic gradient descent* (DP-SGD) (Abadi et al., 2016) is a widely used DP algorithm to train ML models (DP-ML). Consider the *empirical risk minimization* (ERM) formulation of ML. Let $\mathcal{L}(\boldsymbol{\theta}; D) := \sum_{d \in D} \ell(\boldsymbol{\theta}; d)$ be the empirical loss function we wish to minimize, where $\ell(\boldsymbol{\theta}; d)$ is the individual loss associated with datum $d$. To approach the minimizer $\boldsymbol{\theta}^*$, SGD works by iteratively sampling a batch of data and use their averaged or summed gradients to update the model parameter. In contrast, DP-SGD works by iteratively applying SGM to compute the noisy batch gradients $\widetilde{\mathbf{G}}_t \leftarrow (1/b) \sum_{d \in B_t} \bar{\mathbf{g}}_d + \mathcal{N}(\mathbf{0}_r, \sigma^2 \mathbf{I}_{r \times r})$ used to update model parameters $\boldsymbol{\theta}$ at each iteration $t$. Here, $B_t$ is the Poisson sampled batch using sampling rate $b/|D|$ to achieve the expected batch size $b$. To control the sensitivity of the batch gradient, each individual gradient $\mathbf{g}_d \leftarrow \nabla_{\boldsymbol{\theta}} \ell(\boldsymbol{\theta}_{t-1}, d)$ is "clipped" to a predetermined clipping threshold $C$ using $\bar{\mathbf{g}}_d \leftarrow \mathbf{g}_d / \max\{1, \|\mathbf{g}_d\|/C\}$ such that the sensitivity of their average is at most $C$. The privacy of DP-SGD is thus guaranteed by adaptive composition of Thm. B.9 as follows.

**Theorem B.10** (Privacy of DP-SGD (Mironov et al., 2019))**.** *The DP-SGD algorithm running for $T$ iterations with sampling rate $q$, clipping threshold $C$ and noise scale $\sigma$ satisfies $(\alpha, \bar{\epsilon})$-RDP, where*

$$\bar{\epsilon} = \frac{2T\alpha q^2 C^2}{\sigma^2}.$$

The detailed algorithm is given in Alg. 2 below:

---

**Algorithm 2 The DP-SGD algorithm.**

---

**Input:** Dataset $D$; initial model parameters $\boldsymbol{\theta}_0$; sampling rates $q$; clipping threshold $C$; noise scale $\sigma$.
**Output:** Trained model parameters $\boldsymbol{\theta}_T$.

1: $b \leftarrow q|D|$
2: **for** iteration $t \in [T]$ **do**
3:     $B_t \leftarrow$ subset sampled from $D$ with rate $q$
4:     **for** datum $d$ in $B_t$ **do**
5:        $\mathbf{g}_d \leftarrow \nabla_{\boldsymbol{\theta}} \ell(\boldsymbol{\theta}_{t-1}, d)$
6:        $\bar{\mathbf{g}}_d \leftarrow \mathbf{g}_d / \max\{1, \|\mathbf{g}_d\|/C\}$
7:     **end for**
8:     $\overline{\mathbf{G}}_t \leftarrow \sum_{d \in B_t} \bar{\mathbf{g}}_d$
9:     $\widetilde{\mathbf{G}}_t \leftarrow \frac{1}{b}\left(\overline{\mathbf{G}}_t + \mathcal{N}(\mathbf{0}_r, \sigma^2 \mathbf{I}_{r \times r})\right)$
10:     $\boldsymbol{\theta}_t \leftarrow \boldsymbol{\theta}_{t-1} - \eta \widetilde{\mathbf{G}}_t$
11: **end for**
12: **return** updated model parameters $\boldsymbol{\theta}_T$

---

### B.2 INDIVIDUALIZED DIFFERENTIAL PRIVACY

Readers may refer to the main paper for detailed backgrounds on IDP and the IDP-SGD algorithm. The pseudocode of IDP-SGD is given in Alg. 3 below:

---

**Algorithm 3 The IDP-SGD algorithm.**

---

**Input:** Datasets of all data owners $D_1, D_2, \cdots, D_N$; initial model parameters $\boldsymbol{\theta}_0$; per-owner sampling rates $q_1, q_2, \cdots, q_N$; per-owner clipping thresholds $C_1, C_2, \cdots, C_N$; noise scale $\sigma$.
**Output:** Trained model parameters $\boldsymbol{\theta}_T$.

1: $b \leftarrow \sum_{n=1}^{N} q_n |D_n|$
2: **for** iteration $t \in [T]$ **do**
3:     **for** owner $n \in [N]$ **do**
4:        $B_{t,n} \leftarrow$ subset sampled from $D_n$ with rate $q_n$
5:     **end for**
6:     $B_t \leftarrow \bigcup_{n=1}^{N} B_{t,n}$
7:     **for** datum $d$ in $B_t$ **do**
8:        $\mathbf{g}_d \leftarrow \nabla_{\boldsymbol{\theta}} \ell(\boldsymbol{\theta}_{t-1}, d)$
9:        $\bar{\mathbf{g}}_d \leftarrow \mathbf{g}_d / \max\{1, \|\mathbf{g}_d\|/C_{o(d)}\}$
10:     **end for**
11:     $\overline{\mathbf{G}}_t \leftarrow \sum_{k=1}^{|B_t|} \bar{\mathbf{g}}_{\pi_t(k)}$
12:     $\widetilde{\mathbf{G}}_t \leftarrow \frac{1}{b}\left(\overline{\mathbf{G}}_t + \mathcal{N}(\mathbf{0}_r, \sigma^2 \mathbf{I}_{r \times r})\right)$
13:     $\boldsymbol{\theta}_t \leftarrow \boldsymbol{\theta}_{t-1} - \eta \widetilde{\mathbf{G}}_t$
14: **end for**
15: **return** updated model parameters $\boldsymbol{\theta}_T$

---

For ease of reference we also include the pseudocodes for the SAMPLE and SCALE variants of IDP-SGD respectively below. The SAMPLE algorithm assigns different sampling rate but the same clipping threshold for different data owners:

---

**Algorithm 4 The SAMPLE algorithm.**

---

**Input:** Datasets of all data owners $D_1, D_2, \cdots, D_N$; initial model parameters $\boldsymbol{\theta}_0$; per-owner sampling rates $q_1, q_2, \cdots, q_N$; clipping threshold $C$; noise scale $\sigma$.
**Output:** Trained model parameters $\boldsymbol{\theta}_T$.

1: $b \leftarrow \sum_{n=1}^{N} q_n |D_n|$
2: **for** iteration $t \in [T]$ **do**
3:     **for** owner $n \in [N]$ **do**
4:        $B_{t,n} \leftarrow$ subset sampled from $D_n$ with rate $q_n$

```
5:     end for
6:     B_t ← ⋃_{n=1}^{N} B_{t,n}
7:     for datum d in B_t do
8:         g_d ← ∇_θ ℓ(θ_{t-1}, d)
9:         ḡ_d ← g_d/ max{1, ‖g_d‖/C}
10:    end for
11:    Ḡ_t ← ∑_{k=1}^{|B_t|} ḡ_{π_t(k)}
12:    G̃_t ← (1/b)(Ḡ_t + 𝒩(0_r, σ²I_{r×r}))
13:    θ_t ← θ_{t-1} − η G̃_t
14: end for
15: return updated model parameters θ_T
```

The SCALE algorithm assigns different clipping thresholds but the same sampling rate for different data owners:

---

**Algorithm 5 The SCALE algorithm.**

---

**Input:** Datasets of all data owners $D_1, D_2, \cdots, D_N$; initial model parameters $\boldsymbol{\theta}_0$; sampling rate $q$; per-owner clipping thresholds $C_1, C_2, \cdots, C_N$; noise scale $\sigma$.
**Output:** Trained model parameters $\boldsymbol{\theta}_T$.

```
1: b ← ∑_{n=1}^{N} q_n|D_n|
2: for iteration t ∈ [T] do
3:     B_t ← subset sampled from D with rate q
4:     for datum d in B_t do
5:         g_d ← ∇_θ ℓ(θ_{t-1}, d)
6:         ḡ_d ← g_d/ max{1, ‖g_d‖/C_{o(d)}}
7:     end for
8:     Ḡ_t ← ∑_{k=1}^{|B_t|} ḡ_{π_t(k)}
9:     G̃_t ← (1/b)(Ḡ_t + 𝒩(0_r, σ²I_{r×r}))
10:    θ_t ← θ_{t-1} − η G̃_t
11: end for
12: return updated model parameters θ_T
```

---

Specially, the two algorithms may be used jointly (i.e., different sampling rates and different clipping thresholds) which results in Alg. 3 and they all satisfy the same privacy guarantee (Thm. 2.2).

### B.3  COMPARISON OF IMBALANCE AND FAIRNESS NOTIONS

In this section, we discuss various notions related to utility imbalance across groups, including **(1)** correcting data/class imbalance; **(2)** correcting DP-exacerbated accuracy disparity (Bagdasaryan et al., 2019); and **(3)** achieving group/individual fairness. We will explain why and how our work, which focuses on IDP-induced utility imbalance/accuracy disparity, is related to **(1)**, somewhat related to but different from **(2)** and significantly different from **(3)**.

**(1) Data/class imbalance.**  As introduced in the main paper, data imbalance refers to (†) the imbalance in sizes of different groups in the training set, where each group optimizes a different objective (e.g., each group holds a class (i.e., class imbalance) or a lot more of one majority class over the others). Existing works on (i) can be broadly grouped into two categories: **data-wise approaches** and **optimization-wise approaches**. Data-wise approaches (He & Garcia, 2009) focus on manipulating the distribution of original dataset or sampled data at each iteration, so that the number of gradients for each group is similar. *Oversampling* works by sampling more data from the minority groups, but is infeasible under IDP due to the limited privacy budget of more private data owners. *Undersampling* works by selecting fewer data from the majority group and making their quantity comparable to the minority group, but is also unsuitable under IDP as it would worsen the model utility when data are already scarce. Optimization-wise approaches, on the other hand, focus on modifying the optimization objectives or algorithms. Many works (Japkowicz, 2000; Lin et al., 2017) attempt to

rescale the loss function in order to obtain gradients of different magnitudes for different groups, but their effects will be eliminated by gradient clipping in DP/IDP. Anand et al. (1993) and Francazi et al. (2023) rescale the obtained gradients directly to balance gradients from each group, but this will in turn change the modular sensitivity of IDP-SGD and hence violate the privacy requirement of the algorithm.

We would like to restate that **our setting differs as we do not assume (†)**, though we recognize data imbalance as a closely related problem since they both exhibit MID and BOO.

**(2) DP-exacerbated accuracy disparity.** Bagdasaryan et al. (2019) first identifies that if there exist underrepresented groups in the training set (i.e., (†) the training set is imbalanced), training with DP algorithms would make such groups even more underrepresented in the trained ML model, which is reflected as even lower accuracy/utility on data from such groups (thus the term "DP-exacerbated"). In particular, it explains how clipping and noise addition of DP jointly lead to such a disparate impact and worsen accuracy disparity between the well-represented groups and underrepresented groups. In contrast, our work does not assume (†) the training set is imbalanced. Instead, we identify that IDP itself can induce utility imbalance. Thus, **our work is orthogonal to the DP-exacerbated accuracy disparity line of works**. Since our work also operates under (I)DP, we use a different term "utility imbalance" to avoid confusion.

Nevertheless, since (ii) and our work both address accuracy disparity/utility imbalance, we give a quick review of existing works on DP-exacerbated accuracy disparity. We thoroughly examine follow-up works of (Bagdasaryan et al., 2019) and classify them as follows. (Farrand et al., 2020; Ganev, 2021; Ganev et al., 2022; Uniyal et al., 2022) analyze the problem of DP-exacerbated accuracy disparity theoretically/empirically. Tran et al. (2021a) uses an alternative objective function which penalizes different (clipped) gradients and (trace of) Hessians among groups, which the authors identify as sources of the disparate impact of DP; Xu et al. (2021) computes different clipping thresholds for different data using additional privacy costs in order to control the contribution of each datum; Esipova et al. (2023) spends additional privacy budgets at each iteration to compute a strict clipping bound (that is larger than the clipping threshold) and discards the gradients larger than the clipping bound; Kulynych et al. (2022) adopts importance sampling on each group's data based on the group size; Knolle et al. (2023) adds a regularizing term to the loss function that penalizes large individual gradient norms which bring larger bias when clipped. Notably, these works operate under standard DP and assume uniform privacy budgets across all data owners. Thus, the impact of an additional privacy cost is the same for each owner. In the IDP setting, however, the same additional privacy cost consumes a larger fraction of the more private data owners' privacy budgets, which worsens the IDP-induced utility imbalance as we had novelly pointed out in Sec. 3.1. Moreover, these works are tailored to the disparate impact of DP caused by its clipping and noise addition components, while our utility imbalance arises from an orthogonal cause. Thus, they cannot address IDP-induced utility imbalance. For example, when using Tran et al. (2021a)'s alternative objective function, the different sampling rates and/or clipping thresholds introduced by IDP-SGD could still lead to MID, BOO and utility imbalance. Last but not least, it is unclear how these works can even be adapted to satisfy IDP.

**(3) Group/individual fairness.** Here we will explain how group/individual fairness works focus on achieving similar model **predictions/behaviors** across different groups/individuals, **which significantly differs from** our work and works in (i) and (ii) which focus on achieving similar model **utilities/accuracies** across different groups/data owners.

- **Group fairness:** Consider a binary classification problem with class label $y \in \{0, 1\}$ representing whether an outcome is positive (e.g., whether a candidate is qualified to a position). Let $\hat{y}$ denote the model prediction. Let $x_s$ denote a sensitive attribute (e.g., race) and $\mathcal{X}_s$ denote the set of its possible values. Each value corresponds to a group. A general definition of *group fairness*, $F_g$, is as follows (Tran et al., 2025):

$$F_g := \max_{u,v \in \mathcal{X}_s} \left( \Pr[\hat{y} = 1 | x_s = u, e] - \Pr[\hat{y} = 1 | x_s = v, e] \right),$$

where $u, v$ are possible values of $x_s$, $e$ is a random event, and the model owner needs to minimize $F_g$ or ensure it is bounded when training their models. When $e = \emptyset$, $F_g$ measures the *statistical parity* (Darlington, 1971), which means that whether an outcome is positive should be independent of the sensitive attribute; when $e$ is the event $y = 1$, $F_g$ measures the *equality of opportunity*

(Hardt et al., 2016), which means that all qualified candidates should get similar opportunity of positive outcome; when $e = y$, $F_g$ measures the *equality of odds* (Hardt et al., 2016), which is stricter than equality of opportunity and requires similar probability even when $y$ is negative. To this end, readers may understand why such works have a different goal from ours: it seeks to achieve similar model predictions/behaviors across different groups (which may lead to a greater imbalance in utilities on some validation sets). Works addressing group fairness under DP include (Ding et al., 2020; Jagielski et al., 2019; Lowy et al., 2023; Tran et al., 2021b; Xian et al., 2024; Xu et al., 2019; Zhang et al., 2021).

• **Individual fairness:** Similar to group fairness, *individual fairness* (Dwork et al., 2012) seeks to achieve similar model predictions/behaviors across different individuals, which is satisfied when

$$\forall \mathbf{x}, \mathbf{x}' \ [\mathfrak{D}_y(\hat{y}, \hat{y}') \le \mathfrak{D}_\mathbf{x}(\mathbf{x}, \mathbf{x}')] ,$$

where $\mathfrak{D}_y$ is some distance metric over the predicted labels and $\mathfrak{D}_\mathbf{x}$ is some distance metric over the inputs. Likewise, its focus is on the model predictions $\hat{y}$ and differs from ours.

**Since the goal of group and individual fairness significantly differ from ours, we do not compare with works under this category (iii).**

## C ADDITIONAL THEORETICAL RESULTS

### C.1 IDP-INDUCED UTILITY IMBALANCE

#### C.1.1 IMBALANCED SAMPLING RATES OR CLIPPING THRESHOLDS

In Fig. 10, we show the imbalance in sampling rates and clipping thresholds of data owners that occur in the SAMPLE and SCALE variants of IDP-SGD respectively, computed using an RDP accountant. For example, for SAMPLE a less private owner with $\epsilon = 8$ can have a 10 times larger sampling rate (0.19) than a more private owner with $\epsilon = 0.6$ (0.019) when clipping threshold $C = 1$ and noise scale $\sigma = 4$; likewise, for SCALE a less private owner with $\epsilon = 3$ can have a 10 times larger clipping threshold (3.29) than a more private owner with $\epsilon = 0.5$ (0.33) when sampling rate $q = 0.05$. These lead to the imbalanced distribution of gradients in the sampled batch $B_t$ (Fig. 2) and cause the utility imbalance problem as explained in Sec. 3.1.

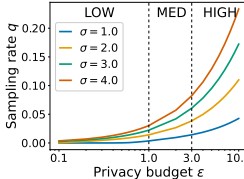

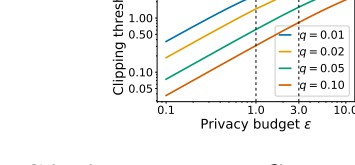

(a) **Sampling rates $q$ corresponding to different privacy budgets $\epsilon$ with added noise scales $\sigma = 1.0$ to $4.0$ in SAMPLE.** Specifically, we set tolerance $\delta = 10^{-5}$, clipping threshold $C = 1.0$ and number of iterations $T = 1000$.

(b) **Clipping thresholds $C$ corresponding to different privacy budgets $\epsilon$ with sampling rate $q = 0.01$ to $0.10$ in SCALE.** Specifically, we set tolerance $\delta = 10^{-5}$, added noise scale $\sigma = 4.0$ and number of iterations $T = 1000$.

Figure 10: **Imbalance of individualized sampling rates and clipping thresholds under IDP-SGD's SAMPLE and SCALE variants.** (a) and (b) show the discrepancies in sampling rates and clipping thresholds for different privacy preferences.

#### C.1.2 PROOF AND DISCUSSIONS FOR THEOREM 3.1

In this section, we state the formal version of Thm. 3.1, prove it and provide some additional discussions. Specifically, the first part of the theorem (Eq. (5)) informs when the MID phenomenon **cannot** happen and therefore how to mitigate it; the second part of the theorem shows that MID **can** happen when the condition is not satisfied. Without loss of generality, we suppose there are two data owners 1 and 2.

**Theorem 3.1** (Condition for Per-Owner Monotone Decrease of Loss in IDP-SGD, formal). *Assume each data owner $n$'s objective function $\mathcal{L}_n$ is $L_1$-Lipschitz and $L_2$-smooth. Assume the stochastic gradient sampled for each owner satisfies uniform boundedness with regards to the owner's full gradient (i.e., $\mathrm{Pr}[\|\mathbf{g}_{t,d} - \nabla_{\boldsymbol{\theta}}\mathcal{L}_{o(d)}(\boldsymbol{\theta}_{t-1})\|^2 \le \xi_{o(d)}^2] = 1$ for some constant $\xi_n$ for each owner $n$).*

*Consider two data owners with dataset, sampling rate and clipping threshold $(D_1, q_1, C_1)$ and $(D_2, q_2, C_2)$, respectively. Recall that their corresponding loss objectives are $\mathcal{L}_1$ and $\mathcal{L}_2$. At each iteration $t$, when the following condition*

$$1 + \cos(\angle_t)\frac{q_2|D_2|C_2}{q_1|D_1|C_1} > \varsigma \tag{5}$$

*is satisfied with $\varsigma > 0$ being a problem-specific term, by choosing a sufficiently small learning rate $\eta$, the losses for both owners are guaranteed to decrease in expectation. Moreover, whenever*

$$1 + \cos(\angle_t)\frac{q_2|D_2|C_2}{q_1|D_1|C_1} < 0,$$

*there exist ML tasks such that the loss for some owner increases, so the above condition is necessary to guarantee a decreasing loss for both owners.*

*Remark.*

- The exact definitions of $\varsigma$ and "sufficiently small" are formally stated in the full proof.
- Although the theorem looks comparable to the original Francazi et al. (2023)'s condition, proving it is significantly more difficult due to the consideration of various components of IDP, including individualized sampling rates $q_n$, individualized clipping thresholds $C_n$, the gradient clipping mechanism that introduces clipping bias to each owner's true gradient and the injected Gaussian noise to guarantee DP.

*Interpretation.* Thm. 3.1 explicitly illustrates the effects of individualized sampling rates $q_n$ and individualized clipping thresholds $C_n$ on per-owner model utility. We now explain what it informs about the utility imbalance problem as we demonstrate graphically in Sec. 3.1. As shown at the start of this section (App. C.1.1), IDP-SGD itself would induce a severe imbalance in the individualized sampling rates $q_n$ and individualized clipping thresholds $C_n$. Therefore, even if the original training set is balanced ($|D_1| \approx |D_2|$), the ratio $\frac{q_2|D_2|C_2}{q_1|D_1|C_1}$ would still be large (without loss of generality assume owner 1 is more private). Moreover, the angle $\angle_t$ is expected to be large as the directions of each owner's full gradient differ when the two groups have different optimization objectives. As $\cos(\angle_t)$ and hence the left-hand side of Eq. 5 are negative, the condition is not satisfied. Thus, when the loss for the less private owner 2 decreases, the loss for the more private owner 1 may not (which explains why MID can be induced by IDP-SGD too). Unlike the original condition (Eq. 1), the ratio $\frac{q_2|D_2|C_2}{q_1|D_1|C_1}$ stays constant throughout the training (in contrast to the data imbalance setting where the corresponding ratio in Eq. 1 gradually decreases) and hence the MID period can be even longer, which worsens the utility imbalance problem.

*Relation to* INO-SGD. From our proof, it is also clear that the larger the left-hand side of Eq. 5 is, the less severe the utility imbalance problem will be, even when the left-hand side is negative. Our solution, the INO-SGD algorithm, aims to increase the left-hand side as much as possible without hurting the overall model utility. By down-weighting the less important data within each batch, INO-SGD is empirically reducing the clipping thresholds $C_n$ for the less private data owners in a strategic way that incurs no penalty on overall model utility by fully utilizing each owner's privacy budget.

*Justification of assumptions.* Our assumptions in Thm. 3.1 are reasonable and in line with existing imbalance/DP literatures. The smoothness assumption on each owner's objective function is often used to characterize the change in loss when a parameter update is made, as a less restrictive assumption than convexity or strong convexity. It is used in recent imbalance works such as (Francazi et al., 2023; Zhang et al., 2025) and DP works such as (Arora et al., 2023; Chen et al., 2020b; Das et al., 2023; Sha et al., 2024). The uniform boundedness assumption is used to bound the stochastic in the sampled gradients, which is widely used in optimization, especially clipping, literature (Li et al., 2023; Zhang et al., 2020; 2019).

*Proof outline.* Since the full proof below is rather long, we provide a **shorter outline** here for better understanding of the proof.

1. We use the symbol ☐ to represent a term and ⊞ to represent a non-negative term. Note that we only need to prove for one owner since the proof for the other owner is identical by symmetry. We first upper bound the **change in loss** $\mathcal{L}_1(\boldsymbol{\theta})$ using the smoothness assumption (see Eq. 6). It can be easily seen that term (B) in Eq. 6 is non-negative, and the whole right-hand side can be written in the format

$$-\eta \left( \Box - \eta \boxplus \right).$$

To this end, if we manage to rewrite and bound terms (A) well such that in the above expression, ☐ $> 0$, then by choosing a learning rate $\eta$ smaller than ☐/⊞, the above expression is negative and we guarantee the decreasing loss. We eventually achieve this in Eq. 11 of the format

$$-\eta \left( \Box - \boxplus - \eta \boxplus \right),$$

where the first ☐ gives a raw version of the target condition (Eq. 12), the middle ⊞ is from an unavoidable clipping bias (which does not appear in the original Francazi et al. (2023)'s non-private setting). Thus, we need the first ☐ to be larger than the middle ⊞ to guarantee a decreasing loss, which leads to the right-hand side of the final condition (Eq. 5) not being 0.

   (a) Term (A) involves the inner product of owner 1's full gradient and the noisy summed clipped gradient $\widetilde{\mathbf{G}}_t$ from both owners. To analyze the imbalance between owners, we break down $\widetilde{\mathbf{G}}_t$ to three components: (i) summed clipped gradient of owner 1 (term (C)); (ii) summed clipped gradient of owner 2 (term (D)); (iii) DP noise. The final bound is given in Eq. 9.
   - For (i) and (ii), we need to handle the *clipping bias*, that is, the different directions of each owner's full gradient and (expectation of) each owner's summed clipped gradient. We handle this by breaking down the summed clipped gradient to the full gradient plus the clipping bias (in our analysis of terms (C) and (D)). The clipping bias is captured by a term, $\phi_{t,d}$, which is the scaling factor of each gradient to obtain its clipped gradient.
   - Therefore, (ii) involves the inner product of owner 1's full gradient and owner 2's full gradient, which highlights the role of $\cos(\angle_t)$.
   - (iii) vanishes as we take expectation because the DP noise is of zero mean.

   (b) Although term (B) is non-negative, we derive a bound in order to better understand it. Similarly, we break down $\widetilde{\mathbf{G}}_t$ to three components: (i) summed clipped gradient of owner 1 (term (C)); (ii) summed clipped gradient of owner 2 (term (D)); (iii) DP noise. Since (B) involves a quadratic term, we need to additionally consider the variance of the three components. The final bound is given in Eq. 10.
   - For (i) and (ii), the variances can be bounded using $\xi_1^2$ and $\xi_2^2$ because of our uniform boundedness assumption (terms (F) and (G)).
   - For (iii), the variance is $r\sigma^2$ since the DP noise is Gaussian.

2. Now we have obtained the desired form and a raw condition (Eq. 12) to guarantee decreasing loss which involves the sum of clipping factors $\phi_{t,d}$. Note that the number of clipping factors in the sum is determined by the number of data sampled from each owner, i.e., $q_n|D_n|$. The clipping factor is determined by the clipping threshold of each owner, i.e., $C_n$. So we just have to further rewrite and bound the raw condition (Eq. 12) to a version that involves $q_n|D_n|$ and $C_n$ to understand the role of $q_n$ and $C_n$, which is exactly the final version (Eq. 5).
   - We observe that the function mapping each owner's gradient to its clipping factor threshold is Lipschitz. Moreover, the mean of each owner's gradient is the owner's full gradient. Therefore, we can derive a bound on the clipping factor using each owner's full gradient (Eq. 13 and 14).
   - By substituting the above bound into the raw condition (Eq. 12), the norm of each owner's full gradient is replaced by their clipping thresholds (Eq. 5), which explains why without data imbalance, IDP itself can induce MID and utility imbalance.

The **full proof** of Theorem 3.1 is as follows.

*Full proof.* We focus on the objective $\mathcal{L}_1(\boldsymbol{\theta})$ for owner 1 (proof for owner 2 is the same by symmetry). Since $\mathcal{L}_1$ is $L_2$-smooth, we have

$$\mathcal{L}_1(\boldsymbol{\theta}_t) - \mathcal{L}_1(\boldsymbol{\theta}_{t-1}) \le \underbrace{-\eta \left\langle \nabla_{\boldsymbol{\theta}} \mathcal{L}_1(\boldsymbol{\theta}_{t-1}), \widetilde{\mathbf{G}}_t \right\rangle}_{(A)} + \underbrace{\frac{L_2 \eta^2}{2} \left\| \widetilde{\mathbf{G}}_t \right\|^2}_{(B)}. \tag{6}$$

Let $\mathbf{Z}_t \sim \mathcal{N}(\mathbf{0}_r, \sigma^2 \mathbf{I}_{r \times r})$ denote the random Gaussian noise added at iteration $t$ (to ensure IDP). Let $\mathbf{g}_{t,d}$ and $\bar{\mathbf{g}}_{t,d}$ denote the gradient and clipped gradient associated with datum $d$ at iteration $t$ respectively. Let $\phi_{t,d} := \min\{1, C_{o(d)}/\|\mathbf{g}_{t,d}\|\} = 1/\max\{1, \|\mathbf{g}_{t,d}\|/C_{o(d)}\} \in (0,1]$ denote the scaling

factor to scale a gradient to its corresponding clipped gradient, i.e., $\bar{\mathbf{g}}_{t,d} = \phi_{t,d}\mathbf{g}_{t,d}$. Then,

$$
\begin{aligned}
\text{Term (A)} &= -\eta \left\langle \nabla_{\boldsymbol{\theta}}\mathcal{L}_1(\boldsymbol{\theta}_{t-1}), \frac{1}{b}\left(\overline{\mathbf{G}}_t + \mathbf{Z}_t\right) \right\rangle \\
&= -\eta \left\langle \nabla_{\boldsymbol{\theta}}\mathcal{L}_1(\boldsymbol{\theta}_{t-1}), \frac{1}{b}\sum_{d\in B_{t,1}}\bar{\mathbf{g}}_{t,d} + \frac{1}{b}\sum_{d\in B_{t,2}}\bar{\mathbf{g}}_{t,d} + \frac{1}{b}\mathbf{Z}_t \right\rangle \\
&= -\eta \left\langle \nabla_{\boldsymbol{\theta}}\mathcal{L}_1(\boldsymbol{\theta}_{t-1}), \frac{1}{b}\sum_{d\in B_{t,1}}\bar{\mathbf{g}}_{t,d} \right\rangle - \eta \left\langle \nabla_{\boldsymbol{\theta}}\mathcal{L}_1(\boldsymbol{\theta}_{t-1}), \frac{1}{b}\sum_{d\in B_{t,2}}\bar{\mathbf{g}}_{t,d} \right\rangle - \eta \left\langle \nabla_{\boldsymbol{\theta}}\mathcal{L}_1(\boldsymbol{\theta}_{t-1}), \frac{1}{b}\mathbf{Z}_t \right\rangle \\
&= -\eta \left\langle \nabla_{\boldsymbol{\theta}}\mathcal{L}_1(\boldsymbol{\theta}_{t-1}), \frac{1}{b}\sum_{d\in B_{t,1}}\phi_{t,d}\mathbf{g}_{t,d} \right\rangle - \eta \left\langle \nabla_{\boldsymbol{\theta}}\mathcal{L}_1(\boldsymbol{\theta}_{t-1}), \frac{1}{b}\sum_{d\in B_{t,2}}\phi_{t,d}\mathbf{g}_{t,d} \right\rangle - \eta \left\langle \nabla_{\boldsymbol{\theta}}\mathcal{L}_1(\boldsymbol{\theta}_{t-1}), \frac{1}{b}\mathbf{Z}_t \right\rangle \\
&= \underbrace{-\frac{\eta}{b}\sum_{d\in B_{t,1}}\langle\nabla_{\boldsymbol{\theta}}\mathcal{L}_1(\boldsymbol{\theta}_{t-1}),\phi_{t,d}\mathbf{g}_{t,d}\rangle}_{\text{(C)}} \underbrace{-\frac{\eta}{b}\sum_{d\in B_{t,2}}\langle\nabla_{\boldsymbol{\theta}}\mathcal{L}_1(\boldsymbol{\theta}_{t-1}),\phi_{t,d}\mathbf{g}_{t,d}\rangle}_{\text{(D)}} -\eta\left\langle\nabla_{\boldsymbol{\theta}}\mathcal{L}_1(\boldsymbol{\theta}_{t-1}),\frac{1}{b}\mathbf{Z}_t\right\rangle .
\end{aligned}
$$

$$
\begin{aligned}
\text{Term (C)} &= -\frac{\eta}{b}\sum_{d\in B_{t,1}}\langle\nabla_{\boldsymbol{\theta}}\mathcal{L}_1(\boldsymbol{\theta}_{t-1}),\ \phi_{t,d}\nabla_{\boldsymbol{\theta}}\mathcal{L}_1(\boldsymbol{\theta}_{t-1}) - (\phi_{t,d}\nabla_{\boldsymbol{\theta}}\mathcal{L}_1(\boldsymbol{\theta}_{t-1}) - \phi_{t,d}\mathbf{g}_{t,d})\rangle \\
&= -\frac{\eta}{b}\sum_{d\in B_{t,1}}\phi_{t,d}\|\nabla_{\boldsymbol{\theta}}\mathcal{L}_1(\boldsymbol{\theta}_{t-1})\|^2 - \phi_{t,d}\langle\nabla_{\boldsymbol{\theta}}\mathcal{L}_1(\boldsymbol{\theta}_{t-1}),\nabla_{\boldsymbol{\theta}}\mathcal{L}_1(\boldsymbol{\theta}_{t-1}) - \mathbf{g}_{t,d}\rangle \\
&\overset{(1)}{\leq} -\frac{\eta}{b}\sum_{d\in B_{t,1}}\phi_{t,d}\|\nabla_{\boldsymbol{\theta}}\mathcal{L}_1(\boldsymbol{\theta}_{t-1})\|^2 - \phi_{t,d}\|\nabla_{\boldsymbol{\theta}}\mathcal{L}_1(\boldsymbol{\theta}_{t-1})\|\|\nabla_{\boldsymbol{\theta}}\mathcal{L}_1(\boldsymbol{\theta}_{t-1}) - \mathbf{g}_{t,d}\| \\
&\overset{(2)}{\leq} -\frac{\eta}{b}\sum_{d\in B_{t,1}}\phi_{t,d}\|\nabla_{\boldsymbol{\theta}}\mathcal{L}_1(\boldsymbol{\theta}_{t-1})\|^2 - \phi_{t,d}\xi_1\|\nabla_{\boldsymbol{\theta}}\mathcal{L}_1(\boldsymbol{\theta}_{t-1})\|,
\end{aligned}
$$

where (1) is due to Cauchy–Schwarz inequality and (2) is due to the uniform boundedness assumption. Likewise,

$$
\begin{aligned}
\text{Term (D)} &\leq -\frac{\eta}{b}\sum_{d\in B_{t,2}}\phi_{t,d}\langle\nabla_{\boldsymbol{\theta}}\mathcal{L}_1(\boldsymbol{\theta}_{t-1}),\nabla_{\boldsymbol{\theta}}\mathcal{L}_2(\boldsymbol{\theta}_{t-1})\rangle - \phi_{t,d}\xi_2\|\nabla_{\boldsymbol{\theta}}\mathcal{L}_1(\boldsymbol{\theta}_{t-1})\| \\
&= -\frac{\eta}{b}\sum_{d\in B_{t,2}}\phi_{t,d}\|\nabla_{\boldsymbol{\theta}}\mathcal{L}_1(\boldsymbol{\theta}_{t-1})\|\|\nabla_{\boldsymbol{\theta}}\mathcal{L}_2(\boldsymbol{\theta}_{t-1})\|\cos(\angle_t) - \phi_{t,d}\xi_2\|\nabla_{\boldsymbol{\theta}}\mathcal{L}_1(\boldsymbol{\theta}_{t-1})\|,
\end{aligned}
$$

where (recall that) $\angle_t$ is the angle between the two owner's repective summed gradients $\nabla_{\boldsymbol{\theta}}\mathcal{L}_1(\boldsymbol{\theta}_{t-1})$ and $\nabla_{\boldsymbol{\theta}}\mathcal{L}_2(\boldsymbol{\theta}_{t-1})$. Therefore,

$$
\begin{aligned}
\text{Term (A)} &\leq -\frac{\eta}{b}\sum_{d\in B_{t,1}}\left(\phi_{t,d}\|\nabla_{\boldsymbol{\theta}}\mathcal{L}_1(\boldsymbol{\theta}_{t-1})\|^2 - \phi_{t,d}\xi_1\|\nabla_{\boldsymbol{\theta}}\mathcal{L}_1(\boldsymbol{\theta}_{t-1})\|\right) \\
&\quad -\frac{\eta}{b}\sum_{d\in B_{t,2}}\left(\phi_{t,d}\|\nabla_{\boldsymbol{\theta}}\mathcal{L}_1(\boldsymbol{\theta}_{t-1})\|\|\nabla_{\boldsymbol{\theta}}\mathcal{L}_2(\boldsymbol{\theta}_{t-1})\|\cos(\angle_t) - \phi_{t,d}\xi_2\|\nabla_{\boldsymbol{\theta}}\mathcal{L}_1(\boldsymbol{\theta}_{t-1})\|\right) \\
&\quad -\eta\left\langle\nabla_{\boldsymbol{\theta}}\mathcal{L}_1(\boldsymbol{\theta}_{t-1}),\frac{1}{b}\mathbf{Z}_t\right\rangle .
\end{aligned}
\tag{7}
$$

Now we move on to term (B):

$$
\begin{aligned}
\text{Term (B)} &= \frac{L_2\eta^2}{2}\left\|\frac{1}{b}(\overline{\mathbf{G}}_t + \mathbf{Z}_t)\right\|^2 \\
&= \frac{L_2\eta^2}{2b^2}\langle\overline{\mathbf{G}}_t + \mathbf{Z}_t, \overline{\mathbf{G}}_t + \mathbf{Z}_t\rangle \\
&= \underbrace{\frac{L_2\eta^2}{2b^2}\|\overline{\mathbf{G}}_t\|^2}_{\text{(E)}} + \frac{L_2\eta^2}{b^2}\langle\overline{\mathbf{G}}_t, \mathbf{Z}_t\rangle + \frac{L_2\eta^2}{2b^2}\|\mathbf{Z}_t\|^2 .
\end{aligned}
$$

$$\text{Term (E)} = \frac{L_2\eta^2}{2b^2}\left\|\sum_{d\in B_{t,1}}\bar{\mathbf{g}}_{t,d} + \sum_{d\in B_{t,2}}\bar{\mathbf{g}}_{t,d}\right\|^2$$

$$\overset{(3)}{\leq} \underbrace{\frac{L_2\eta^2}{b^2}\left\|\sum_{d\in B_{t,1}}\bar{\mathbf{g}}_{t,d}\right\|^2}_{(F)} + \underbrace{\frac{L_2\eta^2}{b^2}\left\|\sum_{d\in B_{t,2}}\bar{\mathbf{g}}_{t,d}\right\|^2}_{(G)},$$

where (3) is due to Cauchy-Schwarz inequality.

$$\text{Term (F)} \overset{(4)}{\leq} \frac{L_2\eta^2|B_{t,1}|}{b^2}\sum_{d\in B_{t,1}}\|\bar{\mathbf{g}}_{t,d}\|^2$$

$$= \frac{L_2\eta^2|B_{t,1}|}{b^2}\sum_{d\in B_{t,1}}\|\phi_{t,d}\mathbf{g}_{t,d}\|^2$$

$$= \frac{L_2\eta^2|B_{t,1}|}{b^2}\sum_{d\in B_{t,1}}\phi_{t,d}^2\|\nabla_{\boldsymbol{\theta}}\mathcal{L}_1(\boldsymbol{\theta}_{t-1}) - (\nabla_{\boldsymbol{\theta}}\mathcal{L}_1(\boldsymbol{\theta}_{t-1}) - \mathbf{g}_{t,d})\|^2$$

$$= \frac{L_2\eta^2|B_{t,1}|}{b^2}\sum_{d\in B_{t,1}}\phi_{t,d}^2\left(\|\nabla_{\boldsymbol{\theta}}\mathcal{L}_1(\boldsymbol{\theta}_{t-1})\|^2 - 2\langle\nabla_{\boldsymbol{\theta}}\mathcal{L}_1(\boldsymbol{\theta}_{t-1}), \nabla_{\boldsymbol{\theta}}\mathcal{L}_1(\boldsymbol{\theta}_{t-1}) - \mathbf{g}_{t,d}\rangle + \|\nabla_{\boldsymbol{\theta}}\mathcal{L}_1(\boldsymbol{\theta}_{t-1}) - \mathbf{g}_{t,d}\|^2\right)$$

$$\overset{(5)}{\leq} \frac{L_2\eta^2|B_{t,1}|}{b^2}\sum_{d\in B_{t,1}}\phi_{t,d}^2\left(\|\nabla_{\boldsymbol{\theta}}\mathcal{L}_1(\boldsymbol{\theta}_{t-1})\|^2 + 2\|\nabla_{\boldsymbol{\theta}}\mathcal{L}_1(\boldsymbol{\theta}_{t-1})\|\|\nabla_{\boldsymbol{\theta}}\mathcal{L}_1(\boldsymbol{\theta}_{t-1}) - \mathbf{g}_{t,d}\| + \|\nabla_{\boldsymbol{\theta}}\mathcal{L}_1(\boldsymbol{\theta}_{t-1}) - \mathbf{g}_{t,d}\|^2\right)$$

$$\overset{(6)}{\leq} \frac{L_2\eta^2|B_{t,1}|}{b^2}\sum_{d\in B_{t,1}}\phi_{t,d}^2\left(\|\nabla_{\boldsymbol{\theta}}\mathcal{L}_1(\boldsymbol{\theta}_{t-1})\|^2 + 2\|\nabla_{\boldsymbol{\theta}}\mathcal{L}_1(\boldsymbol{\theta}_{t-1})\|\xi_1 + \xi_1^2\right),$$

where (4) and (5) are due to Cauchy-Schwarz inequality, (6) is due to the uniform boundedness assumption. Likewise,

$$\text{Term (G)} \leq \frac{L_2\eta^2|B_{t,2}|}{b^2}\sum_{d\in B_{t,2}}\phi_{t,d}^2\left(\|\nabla_{\boldsymbol{\theta}}\mathcal{L}_2(\boldsymbol{\theta}_{t-1})\|^2 + 2\|\nabla_{\boldsymbol{\theta}}\mathcal{L}_2(\boldsymbol{\theta}_{t-1})\|\xi_2 + \xi_2^2\right).$$

Therefore,

$$\text{Term (B)} \leq \frac{L_2\eta^2|B_{t,1}|}{b^2}\sum_{d\in B_{t,1}}\phi_{t,d}^2\left(\|\nabla_{\boldsymbol{\theta}}\mathcal{L}_1(\boldsymbol{\theta}_{t-1})\|^2 + 2\|\nabla_{\boldsymbol{\theta}}\mathcal{L}_1(\boldsymbol{\theta}_{t-1})\|\xi_1 + \xi_1^2\right)$$

$$+ \frac{L_2\eta^2|B_{t,2}|}{b^2}\sum_{d\in B_{t,2}}\phi_{t,d}^2\left(\|\nabla_{\boldsymbol{\theta}}\mathcal{L}_2(\boldsymbol{\theta}_{t-1})\|^2 + 2\|\nabla_{\boldsymbol{\theta}}\mathcal{L}_2(\boldsymbol{\theta}_{t-1})\|\xi_2 + \xi_2^2\right)$$

$$+ \frac{L_2\eta^2}{b^2}\langle\overline{\mathbf{G}}_t, \mathbf{Z}_t\rangle + \frac{L_2\eta^2}{2b^2}\|\mathbf{Z}_t\|^2. \tag{8}$$

Let $\mathcal{F}_{t-1}$ denote the filtration at iteration $t$ (i.e., all the algorithm histories up to the start of iteration $t$). Now consider the expectation of terms (A) (Eq. 7) and (B) (Eq. 8) given filtration $\mathcal{F}_{t-1}$ at iteration $t$ (i.e., now that we know we are already at a particular $\boldsymbol{\theta}_{t-1}$, what happens to the loss after we update the parameters at iteration $t$?):

$$\mathbb{E}[\text{term (A)} \mid \mathcal{F}_{t-1}] \leq \mathbb{E}\left[-\frac{\eta}{b}\sum_{d\in B_{t,1}}\left(\phi_{t,d}\|\nabla_{\boldsymbol{\theta}}\mathcal{L}_1(\boldsymbol{\theta}_{t-1})\|^2 - \phi_{t,d}\xi_1\|\nabla_{\boldsymbol{\theta}}\mathcal{L}_1(\boldsymbol{\theta}_{t-1})\|\right)\right.$$

$$\left. -\frac{\eta}{b}\sum_{d\in B_{t,2}}\left(\phi_{t,d}\|\nabla_{\boldsymbol{\theta}}\mathcal{L}_1(\boldsymbol{\theta}_{t-1})\|\|\nabla_{\boldsymbol{\theta}}\mathcal{L}_2(\boldsymbol{\theta}_{t-1})\|\cos(\angle_t) - \phi_{t,d}\xi_2\|\nabla_{\boldsymbol{\theta}}\mathcal{L}_1(\boldsymbol{\theta}_{t-1})\|\right)\right|\mathcal{F}_{t-1}\right]$$

$$\underbrace{-\mathbb{E}\left[\eta\left\langle\nabla_{\boldsymbol{\theta}}\mathcal{L}_1(\boldsymbol{\theta}_{t-1}), \frac{1}{b}\mathbf{Z}_t\right\rangle\bigg|\mathcal{F}_{t-1}\right]}_{(H)}.$$

Term (H) $= 0$ because $\mathbf{Z}_t$ is a Gaussian noise with zero mean. For the terms before (H), given filtration $\mathcal{F}_{t-1}$, everything, except $B_{t,1}, B_{t,2}$ and hence $\phi_{t,d}$, are constant. Like other works that theoretically analyze DP (Abadi et al., 2016; Bassily et al., 2019), we assume $|B_{t,1}| = q_1|D_1|$ and $|B_{t,2}| = q_2|D_2|$. Let $\bar{\phi}_{t,1} := \mathbb{E}_{d \in D_1}[\phi_{t,d}|\mathcal{F}_{t-1}]$ and similarly $\bar{\phi}_{t,2} := \mathbb{E}_{d \in D_2}[\phi_{t,d}|\mathcal{F}_{t-1}]$. We have

$$
\begin{aligned}
\mathbb{E}[\text{term (A)} \mid \mathcal{F}_{t-1}] \leq &-\frac{\eta q_1|D_1|}{b}\left(\bar{\phi}_{t,1}\|\nabla_{\boldsymbol{\theta}}\mathcal{L}_1(\boldsymbol{\theta}_{t-1})\|^2 - \bar{\phi}_{t,1}\xi_1\|\nabla_{\boldsymbol{\theta}}\mathcal{L}_1(\boldsymbol{\theta}_{t-1})\|\right) \\
&-\frac{\eta q_2|D_2|}{b}\left(\bar{\phi}_{t,2}\|\nabla_{\boldsymbol{\theta}}\mathcal{L}_1(\boldsymbol{\theta}_{t-1})\|\|\nabla_{\boldsymbol{\theta}}\mathcal{L}_2(\boldsymbol{\theta}_{t-1})\|\cos(\angle_t) - \phi_{t,2}\xi_2\|\nabla_{\boldsymbol{\theta}}\mathcal{L}_1(\boldsymbol{\theta}_{t-1})\|\right) \\
= &-\frac{\eta}{b}\big(q_1|D_1|\bar{\phi}_{t,1}\|\nabla_{\boldsymbol{\theta}}\mathcal{L}_1(\boldsymbol{\theta}_{t-1})\|^2 - q_1|D_1|\bar{\phi}_{t,1}\xi_1\|\nabla_{\boldsymbol{\theta}}\mathcal{L}_1(\boldsymbol{\theta}_{t-1})\| \\
&+ q_2|D_2|\bar{\phi}_{t,2}\|\nabla_{\boldsymbol{\theta}}\mathcal{L}_1(\boldsymbol{\theta}_{t-1})\|\|\nabla_{\boldsymbol{\theta}}\mathcal{L}_2(\boldsymbol{\theta}_{t-1})\|\cos(\angle_t) - q_2|D_2|\bar{\phi}_{t,2}\xi_2\|\nabla_{\boldsymbol{\theta}}\mathcal{L}_1(\boldsymbol{\theta}_{t-1})\|\big).
\end{aligned}
$$
(9)

Now we move on to the expectation of term (B):

$$
\begin{aligned}
\mathbb{E}[\text{term (B)} \mid \mathcal{F}_{t-1}] \leq \mathbb{E}\bigg[&\frac{L_2\eta^2|B_{t,1}|}{b^2}\sum_{d \in B_{t,1}}\phi_{t,d}^2\left(\|\nabla_{\boldsymbol{\theta}}\mathcal{L}_1(\boldsymbol{\theta}_{t-1})\|^2 + 2\|\nabla_{\boldsymbol{\theta}}\mathcal{L}_1(\boldsymbol{\theta}_{t-1})\|\xi_1 + \xi_1^2\right) \\
&+ \frac{L_2\eta^2|B_{t,2}|}{b^2}\sum_{d \in B_{t,2}}\phi_{t,d}^2\left(\|\nabla_{\boldsymbol{\theta}}\mathcal{L}_2(\boldsymbol{\theta}_{t-1})\|^2 + 2\|\nabla_{\boldsymbol{\theta}}\mathcal{L}_2(\boldsymbol{\theta}_{t-1})\|\xi_2 + \xi_2^2\right)\bigg|\mathcal{F}_{t-1}\bigg] \\
&+ \underbrace{\mathbb{E}\left[\frac{L_2\eta^2}{b^2}\langle\overline{\mathbf{G}}_t, \mathbf{Z}_t\rangle\bigg|\mathcal{F}_{t-1}\right]}_{(I)} + \underbrace{\mathbb{E}\left[\frac{L_2\eta^2}{2b^2}\|\mathbf{Z}_t\|^2\bigg|\mathcal{F}_{t-1}\right]}_{(J)}.
\end{aligned}
$$

Since the Gaussian noise $\mathbf{Z}_t \sim \mathcal{N}(\mathbf{0}_r, \sigma^2\mathbf{I}_{r \times r})$, term (I) $= 0$ and term (J) $= \frac{L_2\eta^2 r\sigma^2}{2b^2}$. We additionally denote $\bar{\phi}_{t,1}^{(2)} := \mathbb{E}_{d \in D_1}[\phi_{t,d}^2|\mathcal{F}_{t-1}]$ and similarly $\bar{\phi}_{t,2}^{(2)} := \mathbb{E}_{d \in D_2}[\phi_{t,d}^2|\mathcal{F}_{t-1}]$. Then,

$$
\begin{aligned}
\mathbb{E}[\text{term (B)} \mid \mathcal{F}_{t-1}] \leq -\frac{\eta}{b}\bigg(&-\frac{L_2\eta}{b}\Big(q_1^2|D_1|^2\bar{\phi}_{t,1}^{(2)}\left(\|\nabla_{\boldsymbol{\theta}}\mathcal{L}_1(\boldsymbol{\theta}_{t-1})\|^2 + 2\|\nabla_{\boldsymbol{\theta}}\mathcal{L}_1(\boldsymbol{\theta}_{t-1})\|\xi_1 + \xi_1^2\right) \\
&+ q_2^2|D_2|^2\bar{\phi}_{t,d}^{(2)}\left(\|\nabla_{\boldsymbol{\theta}}\mathcal{L}_2(\boldsymbol{\theta}_{t-1})\|^2 + 2\|\nabla_{\boldsymbol{\theta}}\mathcal{L}_2(\boldsymbol{\theta}_{t-1})\|\xi_2 + \xi_2^2 + \frac{r\sigma^2}{2}\Big)\bigg).
\end{aligned}
$$
(10)

Combining Eq. 9 and 10,

$$
\begin{aligned}
\mathbb{E}[\mathcal{L}_1(\boldsymbol{\theta}_t) \mid \mathcal{F}_{t-1}] - \mathcal{L}_1(\boldsymbol{\theta}_{t-1}) \leq -\frac{\eta}{b}\bigg(&q_1|D_1|\bar{\phi}_{t,1}\|\nabla_{\boldsymbol{\theta}}\mathcal{L}_1(\boldsymbol{\theta}_{t-1})\|^2 \\
&+ q_2|D_2|\bar{\phi}_{t,2}\|\nabla_{\boldsymbol{\theta}}\mathcal{L}_1(\boldsymbol{\theta}_{t-1})\|\|\nabla_{\boldsymbol{\theta}}\mathcal{L}_2(\boldsymbol{\theta}_{t-1})\|\cos(\angle_t) \\
&- q_1|D_1|\bar{\phi}_{t,1}\xi_1\|\nabla_{\boldsymbol{\theta}}\mathcal{L}_1(\boldsymbol{\theta}_{t-1})\| - q_2|D_2|\bar{\phi}_{t,2}\xi_2\|\nabla_{\boldsymbol{\theta}}\mathcal{L}_1(\boldsymbol{\theta}_{t-1})\| \\
&- \frac{L_2\eta}{b}\Big(q_1^2|D_1|^2\bar{\phi}_{t,1}^{(2)}\left(\|\nabla_{\boldsymbol{\theta}}\mathcal{L}_1(\boldsymbol{\theta}_{t-1})\|^2 + 2\|\nabla_{\boldsymbol{\theta}}\mathcal{L}_1(\boldsymbol{\theta}_{t-1})\|\xi_1 + \xi_1^2\right) \\
&+ q_2^2|D_2|^2\bar{\phi}_{t,d}^{(2)}\left(\|\nabla_{\boldsymbol{\theta}}\mathcal{L}_2(\boldsymbol{\theta}_{t-1})\|^2 + 2\|\nabla_{\boldsymbol{\theta}}\mathcal{L}_2(\boldsymbol{\theta}_{t-1})\|\xi_2 + \xi_2^2 + \frac{r\sigma^2}{2}\Big)\bigg) \\
= -\frac{\eta}{b}\bigg(&\underbrace{\left(q_1|D_1|\bar{\phi}_{t,1} + q_2|D_2|\bar{\phi}_{t,2}\frac{\|\nabla_{\boldsymbol{\theta}}\mathcal{L}_2(\boldsymbol{\theta}_{t-1})\|}{\|\nabla_{\boldsymbol{\theta}}\mathcal{L}_1(\boldsymbol{\theta}_{t-1})\|}\cos(\angle_t)\right)}_{(K)}\|\nabla_{\boldsymbol{\theta}}\mathcal{L}_1(\boldsymbol{\theta}_{t-1})\|^2 \\
&\underbrace{-\left(q_1|D_1|\bar{\phi}_{t,1}\xi_1 + q_2|D_2|\bar{\phi}_{t,2}\xi_2\right)\|\nabla_{\boldsymbol{\theta}}\mathcal{L}_1(\boldsymbol{\theta}_{t-1})\|}_{(L)} \\
&\underbrace{-\frac{L_2\eta}{b}\Big(q_1^2|D_1|^2\bar{\phi}_{t,1}^{(2)}\left(\|\nabla_{\boldsymbol{\theta}}\mathcal{L}_1(\boldsymbol{\theta}_{t-1})\|^2 + 2\|\nabla_{\boldsymbol{\theta}}\mathcal{L}_1(\boldsymbol{\theta}_{t-1})\|\xi_1 + \xi_1^2\right)}_{(M), \text{ part 1}}
\end{aligned}
$$

$$+q_2^2|D_2|^2\bar{\phi}_{t,d}^{(2)}\left(\|\nabla_{\boldsymbol{\theta}}\mathcal{L}_2(\boldsymbol{\theta}_{t-1})\|^2 + 2\|\nabla_{\boldsymbol{\theta}}\mathcal{L}_2(\boldsymbol{\theta}_{t-1})\|\xi_2 + \xi_2^2) + \frac{r\sigma^2}{2}\right)\bigg)\bigg).$$

$$\underbrace{\phantom{+q_2^2|D_2|^2\bar{\phi}_{t,d}^{(2)}\left(\|\nabla_{\boldsymbol{\theta}}\mathcal{L}_2(\boldsymbol{\theta}_{t-1})\|^2 + 2\|\nabla_{\boldsymbol{\theta}}\mathcal{L}_2(\boldsymbol{\theta}_{t-1})\|\xi_2 + \xi_2^2) + \frac{r\sigma^2}{2}\right)}}_{\text{(M), part 2}}$$

$$(11)$$

In order to guarantee a decreasing loss in expectation, the learning rate $\eta$ should be chosen such that the above term $\leq 0$. Note that terms (L) and (M) are never positive. Thus, term (K) must be positive, and it should be positive enough at least to compensate term (L) (since we can control term (M) by varying the step size $\eta$). Let $\psi \coloneqq \left(q_1|D_1|\bar{\phi}_{t,1}\xi_1 + q_2|D_2|\bar{\phi}_{t,2}\xi_2\right)\|\nabla_{\boldsymbol{\theta}}\mathcal{L}_1(\boldsymbol{\theta}_{t-1})\|/\|\nabla_{\boldsymbol{\theta}}\mathcal{L}_1(\boldsymbol{\theta}_{t-1})\|^2$ (clearly $\psi \geq 0$) and we arrive at the following condition:

$$q_1|D_1|\bar{\phi}_{t,1} + q_2|D_2|\bar{\phi}_{t,2}\frac{\|\nabla_{\boldsymbol{\theta}}\mathcal{L}_2(\boldsymbol{\theta}_{t-1})\|}{\|\nabla_{\boldsymbol{\theta}}\mathcal{L}_1(\boldsymbol{\theta}_{t-1})\|}\cos(\angle_t) > \psi. \tag{12}$$

$q_1|D_1|$ and $q_2|D_2|$ are clearly controlled by the individualized sampling rates $q_1$ and $q_2$ specified by IDP-SGD (e.g., the SAMPLE variant). Now we will break down $\bar{\phi}_{t,1}$ and $\bar{\phi}_{t,2}$ to see how the above is also controlled by the individualized clipping thresholds $C_1$ and $C_2$ specified by IDP-SGD (e.g., the SCALE variant). We have $\bar{\phi}_{t,1} = \mathbb{E}_{d\in D_1}[\phi_{t,d}|\mathcal{F}_{t-1}] = \mathbb{E}_{d\in D_1}[\min\{1, C_1/\|\mathbf{g}_{t,d}\|\}|\mathcal{F}_{t-1}]$. The function $\min\{1, C_1/x\}$ has derivative $-C_1 x^{-2}$ on $(C_1, +\infty)$. Thus, when $x \in [\|\nabla_{\boldsymbol{\theta}}\mathcal{L}_1(\boldsymbol{\theta}_{t-1})\| - \xi_1, \|\nabla_{\boldsymbol{\theta}}\mathcal{L}_1(\boldsymbol{\theta}_{t-1})\| + \xi_1]$ (by uniform boundedness and triangle inequality), the function $\min\{1, C_1/x\}$ is locally Lipschitz with Lipschitz constant $C_1/(\max\{\|\nabla_{\boldsymbol{\theta}}\mathcal{L}_1(\boldsymbol{\theta}_{t-1})\| - \xi_1, C_1\})^2$. Also, the $L^2$-norm $\|\cdot\|$ is 1-Lipschitz. Thus, we have

$$\left|\bar{\phi}_{t,1} - \min\left\{1, \frac{C_1}{\|\nabla_{\boldsymbol{\theta}}\mathcal{L}_1(\boldsymbol{\theta}_{t-1})\|}\right\}\right| \leq \frac{C_1}{(\max\{\|\nabla_{\boldsymbol{\theta}}\mathcal{L}_1(\boldsymbol{\theta}_{t-1})\| - \xi_1, C_1\})^2}\cdot\xi_1. \tag{13}$$

Similarly,

$$\left|\bar{\phi}_{t,2} - \min\left\{1, \frac{C_2}{\|\nabla_{\boldsymbol{\theta}}\mathcal{L}_2(\boldsymbol{\theta}_{t-1})\|}\right\}\right| \leq \frac{C_2}{(\max\{\|\nabla_{\boldsymbol{\theta}}\mathcal{L}_2(\boldsymbol{\theta}_{t-1})\| - \xi_2, C_2\})^2}\cdot\xi_2. \tag{14}$$

That is, $\bar{\phi}_{t,1}$ lies in a small region around the clipping factor for owner 1's full gradient, $\min\{1, C_1/\|\nabla_{\boldsymbol{\theta}}\mathcal{L}_1(\boldsymbol{\theta}_{t-1})\|\}$ and $\bar{\phi}_{t,2}$ lies in a small region around the clipping factor for owner 2's full gradient, $\min\{1, C_2/\|\nabla_{\boldsymbol{\theta}}\mathcal{L}_2(\boldsymbol{\theta}_{t-1})\|\}$. Denote the right-hand sides of Eq. 13 and Eq. 14 as $\varsigma_1 > 0$ and $\varsigma_2 > 0$ respectively. To guarantee Eq. 12 to hold, we need

$$q_1|D_1|\min\left\{1, \frac{C_1}{\|\nabla_{\boldsymbol{\theta}}\mathcal{L}_1(\boldsymbol{\theta}_{t-1})\|}\right\} + q_2|D_2|\min\left\{1, \frac{C_2}{\|\nabla_{\boldsymbol{\theta}}\mathcal{L}_2(\boldsymbol{\theta}_{t-1})\|}\right\}\frac{\|\nabla_{\boldsymbol{\theta}}\mathcal{L}_2(\boldsymbol{\theta}_{t-1})\|}{\|\nabla_{\boldsymbol{\theta}}\mathcal{L}_1(\boldsymbol{\theta}_{t-1})\|}\cos(\angle_t)$$
$$> q_1|D_1|\varsigma_1 + q_2|D_2|\varsigma_2 + \psi.$$

To fully utilize the privacy budgets, except at the very end of training, the clipping thresholds $C_1$ and $C_2$ should be set smaller than the raw gradients (works on *adaptive clipping thresholds* (Bu et al., 2023; Xia et al., 2023) actually enforce this). Thus, the above reduces to

$$q_1|D_1|C_1/\|\nabla_{\boldsymbol{\theta}}\mathcal{L}_1(\boldsymbol{\theta}_{t-1})\| + q_2|D_2|C_2\cos(\angle_t)/\|\nabla_{\boldsymbol{\theta}}\mathcal{L}_1(\boldsymbol{\theta}_{t-1})\| > q_1|D_1|\varsigma_1 + q_2|D_2|\varsigma_2 + \psi,$$

which requires $q_1|D_1|C_1 + q_2|D_2|C_2\cos(\angle_t) > 0$ even when the bound is tightest (i.e., $\varsigma_1 = \varsigma_2 = 0$). Therefore, letting $\varsigma = \|\nabla_{\boldsymbol{\theta}}\mathcal{L}_1(\boldsymbol{\theta}_{t-1})\|(q_1|D_1|\varsigma_1 + q_2|D_2|\varsigma_2 + \psi)/(q_1|D_1|C_1)$ (which is always positive), the condition can be reduced to Eq. 5.

When Eq. 5 and hence Eq. 12 are satisfied, referring to the expected loss change in Eq. 11, by choosing a $\eta$ smaller than

$$\frac{\underbrace{\left(q_1|D_1|\bar{\phi}_{t,1} + q_2|D_2|\bar{\phi}_{t,2}\frac{\|\nabla_{\boldsymbol{\theta}}\mathcal{L}_2(\boldsymbol{\theta}_{t-1})\|}{\|\nabla_{\boldsymbol{\theta}}\mathcal{L}_1(\boldsymbol{\theta}_{t-1})\|}\cos(\angle_t)\right)}_{\text{(K)}}\|\nabla_{\boldsymbol{\theta}}\mathcal{L}_1(\boldsymbol{\theta}_{t-1})\|^2 - \underbrace{\left(q_1|D_1|\bar{\phi}_{t,1}\xi_1 + q_2|D_2|\bar{\phi}_{t,2}\xi_2\right)\|\nabla_{\boldsymbol{\theta}}\mathcal{L}_1(\boldsymbol{\theta}_{t-1})\|}_{\text{(L)}}}{\underbrace{\frac{L_2}{b}\left(q_1^2|D_1|^2\bar{\phi}_{t,1}^{(2)}\left(\|\nabla_{\boldsymbol{\theta}}\mathcal{L}_1(\boldsymbol{\theta}_{t-1})\|^2 + 2\|\nabla_{\boldsymbol{\theta}}\mathcal{L}_1(\boldsymbol{\theta}_{t-1})\|\xi_1 + \xi_1^2\right) + q_2^2|D_2|^2\bar{\phi}_{t,d}^{(2)}\left(\|\nabla_{\boldsymbol{\theta}}\mathcal{L}_2(\boldsymbol{\theta}_{t-1})\|^2 + 2\|\nabla_{\boldsymbol{\theta}}\mathcal{L}_2(\boldsymbol{\theta}_{t-1})\|\xi_2 + \xi_2^2\right) + \frac{r\sigma^2}{2}\right)}_{\text{(M) without } \eta}},$$

the loss is expected to decrease. Note that both the numerator and the denominator are guaranteed to be positive by Eq. 12.

*Tightness of the result.* We first show that whenever $1 + \cos(\angle_t)\frac{q_2|D_2|C_2}{q_1|D_1|C_1} < 0$, the losses for both owners are not guaranteed to decrease together. Consider number of parameters $r = 2$ and objectives $\mathcal{L}_1(\boldsymbol{\theta}) = \theta_1^2 + (100 - \theta_2)^2$ and $\mathcal{L}_2(\boldsymbol{\theta}) = (100 - \theta_1)^2 + \theta_2^2$, the clipping thresholds for owner 1 and 2 are $C_1 = 1$ and $C_2$ respectively and 1 datum is sampled from each owner, i.e., $q_1|D_1| = q_2|D_2| = 1$. Suppose now $\boldsymbol{\theta}_{t-1} = \binom{40}{40}$. Suppose that the mean gradient of each owner $n$ is the actual gradient $\nabla_{\boldsymbol{\theta}}\mathcal{L}_n(\boldsymbol{\theta}_{t-1})$, that is, $\binom{80}{-120}$ for owner 1 and $\binom{-120}{80}$ for owner 2. In this case, $\cos(\angle_t) = (80 \times (-120) + (-120) \times 80)/(120^2 + 80^2) = -12/13$. Suppose that for both owners, if their respective mean gradient is $\mathbf{g}$, the stochastic gradient is a random sample from $\binom{\text{Uniform}(g_1-50, g_1+50)}{\text{Uniform}(g_2-5, g_2+5)}$. Assuming that the DP noise is negligibly small[9], the expected sum of clipped gradient is

$$\begin{pmatrix} \frac{1}{10^3}\int_{30}^{130}\int_{-125}^{-115} \frac{g_1}{\sqrt{g_1^2+g_2^2}} \, \mathrm{d}g_2 \, \mathrm{d}g_1 \\ \frac{1}{10^3}\int_{30}^{130}\int_{-125}^{-115} \frac{g_2}{\sqrt{g_1^2+g_2^2}} \, \mathrm{d}g_2 \, \mathrm{d}g_1 \end{pmatrix} + \begin{pmatrix} \frac{1}{10^3}\int_{-170}^{-70}\int_{75}^{85} \frac{C_2 g_1}{\sqrt{g_1^2+g_2^2}} \, \mathrm{d}g_2 \, \mathrm{d}g_1 \\ \frac{1}{10^3}\int_{-170}^{-70}\int_{75}^{85} \frac{C_2 g_2}{\sqrt{g_1^2+g_2^2}} \, \mathrm{d}g_2 \, \mathrm{d}g_1 \end{pmatrix} \approx \begin{pmatrix} 0.532 \\ -0.830 \end{pmatrix} + C_2 \begin{pmatrix} -0.816 \\ 0.566 \end{pmatrix}.$$

The loss for owner 1 will decrease only when the expected summed gradient lies between $\binom{-120}{-80}$ and owner 1's mean gradient, i.e., $C_2 < 1.067$, that is, $1 + \cos(\angle_t)\frac{q_2|D_2|C_2}{q_1|D_1|C_1} < 0.0148$. This suffices to show that whenever $1 + \cos(\angle_t)\frac{q_2|D_2|C_2}{q_1|D_1|C_1} < 0$, the losses for both owners are not guaranteed to decrease together. This also shows that the problem-specific positive term $\varsigma$ in the right-hand side of Eq. 5 is necessary, as compared with the original Francazi et al. (2023)'s condition (Eq. 1).

In our proof, the bound for term (A) in Eq. 7 can achieve equality when all the stochastic gradients for an owner $n$ are in the same direction and their norms are either $\|\nabla_{\boldsymbol{\theta}}\mathcal{L}_n(\boldsymbol{\theta}_{t-1}) - \xi_n\|$ or $\|\nabla_{\boldsymbol{\theta}}\mathcal{L}_n(\boldsymbol{\theta}_{t-1}) + \xi_n\|$, so it is tight. This is important because only term (A) is responsible for the condition (Eq. 5). Term (B) is only related to term (M) in Eq. 11 and thus even if a tighter bound can be attained, one will arrive at a similar result by choosing a slightly larger learning rate $\eta$. Therefore, we choose a reasonably good bound for (B).

## C.2 THE INO-SGD ALGORITHM

### C.2.1 UNSUITABILITY OF STRAIGHTFORWARD APPROACHES

In this section, we give a list of straightforward approaches to address IDP-induced utility imbalance that might be considered at first glance. We will argue why all of them are inadequate:

- **Enforcing the lowest privacy budget for all owners** (e.g., by undersampling or using a smaller clipping thresholds): This approach aligns with DP-SGD, which under-utilizes IDP budgets and has a significantly lower overall model utility compared to IDP-SGD (Boenisch et al., 2024) (see empirical evidence in App. D.4.3). Our algorithm will fully utilize each owners' IDP budgets instead.
- **Randomly downscaling owners with higher privacy budgets** (e.g., by down-weighting or undersampling): This approach has several issues. Firstly, for multiple data owners and privacy budgets, it is unclear how to set (i) the cutoff between less and more private owners and (ii) how much to downscale from each less private owner. For (i), for example, for 10 owners with $\epsilon_n \in [0.5, 5]$ evenly, it is unsuitable to directly use the mean $2.25$ as the cutoff since the impact of privacy does not scale linearly with privacy budgets and is hard to quantify *a priori*; for (ii), downscaling in the same way from owners with $\epsilon_n = 3$ and $\epsilon_n = 5$ is unsuitable, as there is still imbalance between these two and downscaling owner 3 incurs different utility loss than owner 5. Due to (i) and (ii), when there are $N$ owners, finding the optimal number of data to drop from **each** owner requires $O(N)$ hyperparameters, making the approach inadequate. Secondly, this approach also under-utilizes IDP budgets and leads to a worse overall model utility. We provide empirical evidence in App. D.1.3 and App. D.4.3. Thirdly, when some data from less private owners still have higher loss (e.g., a harder-to-predict one), randomly downscaling would undesirably worsen utility imbalance. Our algorithm would order data by their losses to identify the important data we should fully keep. This is sensible as the importance of gradient information of each datum in batch $B_t$ at iteration $t$ directly correlates with their individual loss values (Kawaguchi & Lu, 2020; Shrivastava et al., 2016).

---

[9]In our proof, we can see that the DP noise appears only in term (M) which can be compensated by a sufficiently small learning rate $\eta$.

- **Keeping samples with top-$\mu$ losses**: As an example, consider a batch $B$ with 16 data and we keep only the gradients with top-8 losses. A private datum $d$ is added to batch $B$ and its loss is ranked 1st. The original 8-th datum $d'$ will no longer be in top-8 and will be discarded. Suppose that the clipping thresholds of $d$ and $d'$ are $C_{o(d)}$ and $C_{o(d')}$ (recall that $o(d)$ refers to owner of $d$). The $L^2$-norm of change in the sum of gradients within batch $B$ can reach a maximum of $C_{o(d)} + C_{o(d')}$ (by triangle inequality), which contradicts the requirement that the addition of $d$ should only cause a change of $L^2$-norm $C_{o(d)}$ in the sum of gradients. Thus, keeping samples with top-$\mu$ losses does not satisfy DP (when all clipping thresholds equate to $C$, the change can reach $2C$) nor IDP.
- **Discarding samples with smallest-$\mu$ losses**: Consider the DP case when all clipping thresholds are equal. Suppose again that a batch $B$ of 16 data is sampled and we remove the smallest-8 losses. When a private datum $d$ is added to $B$, if its loss is larger than the datum with 9-th loss, it will be kept and the norm of change in the summed gradient is at most $C_{o(d)}$; if its loss is smaller than the datum with 9-th loss, it will be discarded and the original 9-th datum $d'$'s gradient will be kept instead of discarded. The norm of change in the summed gradient is still at most $C_{o(d')}$. When $d$ is instead already in $B$ and will be removed from $B$, the analysis will be similar to the norm of change when $d$ is added to $B \setminus \{d\}$. Thus, this approach actually satisfies DP. However, it does not satisfy IDP. In the case of IDP, when $d$ is added to the batch and its loss is smaller than the 9-th loss, it will be discarded. The original 9-th datum, say $d'$, will be kept instead of discarded, causing a change in the summed gradient with norm $C_{o(d')}$, which can be larger than $C_{o(d)}$. Thus, the modular sensitivity $\Delta_{\mathcal{A}}^d$ is no longer $C_{o(d)}$ and this approach does not satisfy IDP.

### C.2.2 ALTERNATIVE SORTING ORDERS

In the main paper we propose using the order of loss to rank the importance of data within each batch. However, it can in fact work with many other sorting orders since it is an INO-SGM mechanism as introduced in Sec. 3.3.1 and App. C.3.2.

One example is a mixed order: we may rank data primarily based on their ownership (i.e., data from the more private data owners are ranked higher) followed by loss values within each owner. In this way, ideally, gradients from the more private data will always be ranked very high and they will never fall into the tail region and be discarded or down-weighted. However, we find that this order does not perform as well as directly using the order of loss:

- Sometimes, the less private owner's data can be difficult to learn and still have large losses and vice versa. INO-SGD using this order would undesirably downweight such data by assigning them low importance scores.
- The least private owner's data are always at the tail, and they could never be learnt. For example, in our **CM-PC** experiments (see App. D.1) with 10 data owners, the least private owner 10's maximum accuracy is only $0.2\%$ throughout the learning dynamics using this order, which means that its data is always severely down-weighted and barely learnt.

### C.2.3 BATCH IMPORTANCE FUNCTION

Based on the transformation described in Sec. 3.2, we include the formal definition of BIF below:

**Definition C.2** (BIF)**.** Given a preset TIF $f_{\text{tail}} : [0, \gamma] \to [0, 1]$ and a sampled batch $B_t$ of data whose sum of clipping thresholds equals $\Gamma_t$, the *batch importance function* (BIF) $f_t : [0, \Gamma_t] \to [0, 1]$ is defined in the following way (under the notation of ordered pairs):

- If $\Gamma_t \geq \gamma$, then $f_t = \{(c, 1) : c \in [0, \Gamma_t - \gamma]\} \cup \{(c, f_{\text{tail}}(c - (\Gamma_t - \gamma)) : c \in (\Gamma_t - \gamma, \Gamma_t])\}$;

- If $\Gamma_t < \gamma$, then $f_t = \{(c, f_{\text{tail}}(c + (\gamma - \Gamma_t))) : c \in [0, \Gamma_t]\}$.

Since TIF is non-increasing and Riemann-integrable, it can be clearly seen that the BIF is also non-increasing and Riemann-integrable, which justifies how INO-SGD computes per-sample importance score via integration.

### C.2.4 RUNTIME OF INO-SGD

We highlight that the runtime of INO-SGD does not differ much from that of IDP-SGD despite the additional operations, i.e., sorting and integrating:

- The most costly steps, including backpropagation and per-sample gradient processing (clipping) are shared by both IDP-SGD and INO-SGD.
- The time taken to sort the gradients in batch $B_t$ is $\mathcal{O}(|B_t| \log |B_t|)$, which is negligible compared with the backpropagation time.
- Simple TIFs, such as a step function (e.g., in App. C.2.5), can be integrated exactly in $\mathcal{O}(1)$ time. The integration of more complex TIFs can be parallelized within each batch. Moreover, since the same TIF is used throughout training, the model owner can pre-compute a *fast integration array*:
  1. Pick a number $\kappa$ much larger than the expected sum of clipping thresholds $\sum_{n \in [N]} q_n |D_n| C_n$;
  2. Compute a BIF $f_\kappa : [0, \kappa] \to [0, 1]$ based on the TIF $f_{\text{tail}}$;
  3. Compute the cumulative integrals $F_\kappa(x) := \int_0^x f_\kappa(c) \, \mathrm{d}c$ for $c = 0, \cdots, \kappa$ up to a certain precision. The $F_\kappa(x)$'s form the fast integration array.
  4. At each iteration $t$, $\int_{c_1}^{c_2} f_t(c) \, \mathrm{d}c = \int_{c_1 + \kappa - \Gamma_t}^{c_2 + \kappa - \Gamma_t} f_\kappa(c) \, \mathrm{d}c = F_\kappa(c_2 + \kappa - \Gamma_t) - F_\kappa(c_1 + \kappa - \Gamma_t)$ becomes a simple $\mathcal{O}(1)$ lookup.

On CIFAR-10 with batch size $1024$, the wallclock runtime of 1 epoch in seconds (averaged over 10 epochs) for IDP-SGD and INO-SGD are $11.80 \pm 1.21$ and $11.89 \pm 0.64$ respectively, which supports our claim that INO-SGD introduces negligible computational overhead.

### C.2.5 Remarks on Tail Importance Functions

In this section we first give an example of an alternative TIF: the model owner can use a step function that splits the $\gamma$-tail into partitions and specifies constant importance scores within each partition: $1/2$ for the first, $1/4$ for the second, $\cdots$, 0 for the last. App. D.4.2 shows that INO-SGD can still give competitive performance for such functions.

Although INO-SGD may work with various forms of TIFs, we still recommend the Beta-distribution-based TIF mentioned in Sec. 3.2 in the main paper, which allows the model owner to flexibly control the TIF shape with only a few parameters. Below is a guideline on how to tune these hyperparameters in practice:

- The TIF with some tail length $\gamma_1$ and some $\beta_1$ has a similar shape to another TIF with a larger tail length $\gamma_2 > \gamma_1$ and larger beta $\beta_2 > \beta_1$, so it suffices to fix a moderately large tail length $\gamma$ (e.g., 50% of the total clipping threshold) to only tune $\alpha$ and $\beta$; OR
- fix $\alpha$ and $\beta$ (e.g., 1 and 1) and set the tail length to (or only tune the tail length around) a certain percentage of the expected sum of clipping thresholds for each batch.

We demonstrate that **INO-SGD is robust to hyperparameter choice** in Sec. 4.3 and App. D.4.2. Thus, tuning according to the above guidelines is practical in real deployments. When the model owner is willing to trade more overall utility for less imbalance, they can manage the tradeoff by increasing the tail length $\gamma$ (the easiest way), or setting a larger $\alpha$ and smaller $\beta$.

### C.3 Theoretical Analysis of INO-SGD

### C.3.1 Proof and Discussions for Theorem 3.3

In this section, we give a formal proof of Thm. 3.3 (privacy of INO-SGD). Consider the sub-algorithm in Line 7-16 of Alg. 1 which takes in a batch of data $B_t$ and returns the sum of clipped gradients $\overline{\mathbf{G}}_t$. Let $c_0 = 0$ and $c_k$ denote the accumulated clipping threshold until (and including) the $k$-th ranked datum $d_{\pi_t(k)}$. We first prove that for any datum $d$, the modular sensitivity $\Delta_{\mathcal{A}}^d$ of the algorithm at iteration $t$ is bounded by $d$'s corresponding clipping threshold $C_{o(d)}$. Suppose that a new datum $d_{\mathrm{a}}$ is added to the batch $B_t$. Without loss of generality, assume that $d_{\mathrm{a}}$ is ranked the $a$-th in the new batch $B_t \cup \{d_{\mathrm{a}}\}$. The importance function changes from $f_t$ to $f_t^{\mathrm{a}}$. It can be seen that all data that are ranked higher than $d_{\mathrm{a}}$ receive larger gradient weights while data that are ranked lower than $d_{\mathrm{a}}$ are not affected. Therefore, the change in the sum of clipped gradients $\overline{\mathbf{G}}_t$ consists of $d_{\mathrm{a}}$'s (weighted) clipped gradient and the increase in (weighted) clipped gradients of all data that are ranked higher than $d_{\mathrm{a}}$, that is,

$$\frac{\int_{c_{a-1}}^{c_a} f_t^{\mathrm{a}} \, \mathrm{d}c}{C_{o(d)}} \cdot \overline{\mathbf{g}}_{d_{\mathrm{a}}} + \sum_{k=1}^{a-1} \left( \frac{\int_{c_{k-1}}^{c_k} f_t^{\mathrm{a}} \, \mathrm{d}c}{c_k - c_{k-1}} - \frac{\int_{c_{k-1}}^{c_k} f_t \, \mathrm{d}c}{c_k - c_{k-1}} \right) \cdot \overline{\mathbf{g}}_{d_k}. \tag{15}$$

Because of how BIF is constructed from TIF, we have $\int_{c_{k-1}}^{c_k} f_t^a \; dc - \int_{c_{k-1}}^{c_k} f_t \; dc = \int_{c_{k-1}}^{c_k} f_t^a \; dc - \int_{c_{k-1}+C_{o(d_a)}}^{c_k+C_{o(d_a)}} f_t^a \; dc = \int_{c_{k-1}}^{c_{k-1}+C_{o(d_a)}} f_t^a \; dc - \int_{c_k}^{c_k+C_{o(d_a)}} f_t^a \; dc$. Thus,

$$\left\| \frac{\int_{c_{a-1}}^{c_a} f_t^a \; dc}{C_{o(d_a)}} \cdot \bar{\mathbf{g}}_{d_a} + \sum_{k=1}^{a-1} \left( \frac{\int_{c_{k-1}}^{c_k} f_t^a \; dc}{c_k - c_{k-1}} - \frac{\int_{c_{k-1}}^{c_k} f_t \; dc}{c_k - c_{k-1}} \right) \cdot \bar{\mathbf{g}}_{d_k} \right\|$$

$$\overset{(1)}{\leq} \frac{\int_{c_{a-1}}^{c_a} f_t^a \; dc}{C_{o(d_a)}} \|\bar{\mathbf{g}}_{d_a}\| + \sum_{k=1}^{a-1} \frac{\int_{c_{k-1}}^{c_{k-1}+C_{o(d_a)}} f_t^a \; dc - \int_{c_k}^{c_k+C_{o(d_a)}} f_t^a \; dc}{c_k - c_{k-1}} \|\bar{\mathbf{g}}_{d_k}\|$$

$$= \int_{c_{a-1}}^{c_a} f_t^a \; dc \cdot \frac{\|\bar{\mathbf{g}}_{d_a}\|}{C_{o(d_a)}} + \sum_{k=1}^{a-1} \left( \int_{c_{k-1}}^{c_{k-1}+C_{o(d_a)}} f_t^a \; dc - \int_{c_k}^{c_k+C_{o(d_a)}} f_t^a \; dc \right) \cdot \frac{\|\bar{\mathbf{g}}_{d_k}\|}{c_k - c_{k-1}}$$

$$\overset{(2)}{\leq} \int_{c_{a-1}}^{c_a} f_t^a \; dc + \sum_{k=1}^{a-1} \left( \int_{c_{k-1}}^{c_{k-1}+C_{o(d_a)}} f_t^a \; dc - \int_{c_k}^{c_k+C_{o(d_a)}} f_t^a \; dc \right)$$

$$\overset{(3)}{=} \int_{c_{a-1}}^{c_a} f_t^a \; dc + \int_{c_0}^{c_0+C_{o(d_a)}} f_t^a \; dc - \int_{c_{a-1}}^{c_a} f_t^a \; dc$$

$$= \int_{c_0}^{c_0+C_{o(d_a)}} f_t^a \; dc$$

$$\leq C_{o(d_a)}.$$

Here (1) is because of triangle inequality and the fact that BIF is non-increasing; (2) is because $\bar{\mathbf{g}}_d = \mathbf{g}_d / \max\{1, \|\mathbf{g}_d\|/C_{o(d)}\}$ and hence $\|\bar{\mathbf{g}}_d\| \leq C_{o(d)}$ for any datum $d$; (3) is by telescoping sum.

When a datum $d_d$ is deleted from the batch $B_t$, the change in sum of clipped gradients is equal to the case where $d_d$ is added to the batch $B_t \setminus \{d_d\}$. The $L^2$-norm of the change is hence bounded by $d_d$'s clipping threshold $C_{o(d_d)}$. Therefore, we have proven the modular sensitivity statement.

For data owner $n \in [N]$ whose sampling rate is $q_n$ at iteration $t$ and whose clipping threshold is $C_n$ (and thus the modular sensitivity corresponding to its data), each iteration of the INO-SGD algorithm satisfies $(\alpha_n, \bar{\epsilon}_n)$-RDP by Thm. B.9 (since it is an SGM), where

$$\bar{\epsilon}_n = 2\alpha_n \frac{C_n^2 q_n^2}{\sigma_t^2}. \tag{16}$$

By composition of RDP along all $T$ iterations (Thm. B.5), we have shown Thm. 3.3. $\qquad\square$

*Remark.* Note that the use of order of loss in each batch does not incur extra privacy costs, as such order at iteration $t$ comes from the ML model obtained at iteration $t-1$, which is itself a standard output under *adaptive composition* (i.e., similar to the gradients we obtain at iteration $t$ for each sampled datum). Thus, for a mechanism using the gradients and/or the order of loss, as long as we bound its (modular) sensitivity at iteration $t$, it strictly satisfies IDP by adaptive composition.

*Remark.* We assume that the model owners have decided on a TIF beforehand (e.g., based on past experience) and do not consider the cost of tuning any hyperparameter. This is consistent with other works that devise (I)DP algorithms to solve certain problems (e.g., Xu et al. (2021); Esipova et al. (2023); Lowy et al. (2023)). Bounding the DP cost in tuning hyperparameters can be addressed by DP hyperparameter tuning works (Sopa et al., 2025; Wang et al., 2023; Ding & Wu, 2025). When the privacy budget for hyperparameter selection is severely limited, the practitioner can consider a few hyperparameter tuning choices based on the following guidelines for the Beta distribution-based TIF. The practitioner can fix a moderately large tail length $\gamma$ (e.g., $50\%$ of total clipping threshold) to only tune $\alpha$ and $\beta$; when privacy budget is extremely limited, the practitioner can fix $\alpha$ and $\beta$ (e.g., 1, 1) and set tail length to (or only tune it around) a fixed percentage of the expected sum of clipping thresholds for each batch, and we show **INO-SGD is robust to hyperparameter choices** in Fig. 16 and Fig. 17.

### C.3.2 THE INO-SGM MECHANISM

In Thm. 3.4, we give a description of the INO-SGM mechanism. We provide the full version below and prove it.

**Theorem 3.4** (INO-SGM, formal). *Let $g$ be a function mapping an arbitrary set of data $\mathcal{B}$ to $\mathbb{R}^r$ such that $\|g(d)\|$ is upper bounded by $\Delta_{\mathcal{A}}^{o(d)}$. Let $\varpi_s$ be a function mapping $\mathcal{B}$ to some order that works in the following way:*

- *$s : \mathbb{R}^r \to \mathbb{R}$ is a scoring function that can assign a score to each datum in $\mathcal{B}$.*
- *For a subset $B \subseteq \mathcal{B}$, $\varpi_s(B)$ returns an order of data in $B$ such that $\varpi_s(B)(k)$ returns the datum $d$ with $k$-th largest score $s(d)$ in $B$.*

*Suppose $B := \{d : d \text{ is sampled with probability } q_{o(d)}\}$ is a sampled subset of data. Then releasing $\sum_{k=1}^{|B|} \rho_k g(\varpi_s(B)(k)) + \mathcal{N}(\mathbf{0}_r, \sigma^2 \mathbf{I}_{r \times r})$ satisfies IRDP, where $\rho_k$ is set in the same way as in BIF.*

More specifically, $g$ represents a function that takes in a datum and returns the information we wish to obtain from the datum; $\varpi_s$ constructs the ordering of any sampled subset $\mathcal{B}$ such that $\varpi_s(B)(k)$ gives the $k$-th ranked datum in $\mathcal{B}$. This can be used to represent the importance, certainty, usefulness or any other properties of data. By summing up $\rho_k g(\varpi_s(B)(k))$, information from better data is assigned a larger weight and thus we will in general obtain useful results. The INO-SGM mechanism can potentially be applied to any field where order of data is important and can be used to improve the outcome.

Now we provide a proof of Thm. 3.4. When a new datum $d_a$ is added to $B$ and assuming $d_a$ is ranked the $a$-th in $B \cup \{d_a\}$, the importance function changes from $f$ to $f^a$. Similar to the proof of Thm. 3.3, the change in the sum of obtained information consists of $d_a$'s (weighted) information and the increase in (weighted) information of all data that are ranked higher than $d_a$, that is,

$$\frac{\int_{c_{a-1}}^{c_a} f^a \, \mathrm{d}c}{\Delta_{o(d)}} \cdot g(d_a) + \sum_{k=1}^{a-1} \left( \frac{\int_{c_{k-1}}^{c_k} f^a \, \mathrm{d}c}{c_k - c_{k-1}} - \frac{\int_{c_{k-1}}^{c_k} f \, \mathrm{d}c}{c_k - c_{k-1}} \right) \cdot g(d_k). \tag{17}$$

Because of how BIF is constructed from TIF, we have $\int_{c_{k-1}}^{c_k} f^a \, \mathrm{d}c - \int_{c_{k-1}}^{c_k} f \, \mathrm{d}c = \int_{c_{k-1}}^{c_k} f^a \, \mathrm{d}c - \int_{c_{k-1}+\Delta_{o(d_a)}}^{c_k+\Delta_{o(d_a)}} f^a \, \mathrm{d}c = \int_{c_{k-1}}^{c_k+\Delta_{o(d_a)}} f^a \, \mathrm{d}c - \int_{c_k}^{c_k+\Delta_{o(d_a)}} f^a \, \mathrm{d}c$. Thus,

$$\left\| \frac{\int_{c_{a-1}}^{c_a} f^a \, \mathrm{d}c}{\Delta_{o(d_a)}} \cdot g(d_a) + \sum_{k=1}^{a-1} \left( \frac{\int_{c_{k-1}}^{c_k} f^a \, \mathrm{d}c}{c_k - c_{k-1}} - \frac{\int_{c_{k-1}}^{c_k} f \, \mathrm{d}c}{c_k - c_{k-1}} \right) \cdot g(d_k) \right\|$$

$$\overset{(1)}{\leq} \frac{\int_{c_{a-1}}^{c_a} f^a \, \mathrm{d}c}{\Delta_{o(d_a)}} \|g(d_a)\| + \sum_{k=1}^{a-1} \frac{\int_{c_{k-1}}^{c_{k-1}+\Delta_{o(d_a)}} f^a \, \mathrm{d}c - \int_{c_k}^{c_k+\Delta_{o(d_a)}} f^a \, \mathrm{d}c}{c_k - c_{k-1}} \|g(d_k)\|$$

$$= \int_{c_{a-1}}^{c_a} f^a \, \mathrm{d}c \cdot \frac{\|g(d_a)\|}{\Delta_{o(d_a)}} + \sum_{k=1}^{a-1} \left( \int_{c_{k-1}}^{c_{k-1}+\Delta_{o(d_a)}} f^a \, \mathrm{d}c - \int_{c_k}^{c_k+\Delta_{o(d_a)}} f^a \, \mathrm{d}c \right) \cdot \frac{\|g(d_k)\|}{c_k - c_{k-1}}$$

$$\overset{(2)}{\leq} \int_{c_{a-1}}^{c_a} f^a \, \mathrm{d}c + \sum_{k=1}^{a-1} \left( \int_{c_{k-1}}^{c_{k-1}+\Delta_{o(d_a)}} f^a \, \mathrm{d}c - \int_{c_k}^{c_k+\Delta_{o(d_a)}} f^a \, \mathrm{d}c \right)$$

$$\overset{(3)}{=} \int_{c_{a-1}}^{c_a} f^a \, \mathrm{d}c + \int_{c_0}^{c_0+\Delta_{o(d_a)}} f^a \, \mathrm{d}c - \int_{c_{a-1}}^{c_a} f^a \, \mathrm{d}c$$

$$= \int_{c_0}^{c_0+\Delta_{o(d_a)}} f^a \, \mathrm{d}c$$

$$\leq \Delta_{o(d_a)}.$$

Here (1) is because of triangle inequality and the fact that BIF is non-increasing; (2) is because $\|g(d)\| \leq \Delta_{o(d)}$ for any datum $d$ as defined; (3) is by telescoping sum.

When a datum $d_d$ is deleted from the batch $B$, the change in sum of clipped gradients is equal to the case where $d_d$ is added to the batch $B \setminus \{d_d\}$. The $L^2$-norm of the change is hence bounded by $d_d$'s clipping threshold $\Delta_{o(d_d)}$, thus completing the proof for modular sensitivity.

For data owner $n \in [N]$ whose sampling rate is $q_n$ and whose clipping threshold is $\Delta_n$ (and thus the modular sensitivity corresponding to its data), the INO-SGM algorithm satisfies $(\alpha_n, \bar{\epsilon}_n)$-RDP by Thm. B.9, where

$$\bar{\epsilon}_n = 2\alpha_n \frac{\Delta_n^2 q_n^2}{\sigma^2}. \tag{18}$$

$\square$

*Remark.* Thm. 3.4 assumes functions $g$ and $s$ do not consume further privacy (i.e., they should be independent of dataset $\mathcal{B}$ (e.g., at the first iteration of SGD where the model parameters are randomly initialized) or are already protected by IRDP (e.g., at each iteration of INO-SGD).

### C.3.3  INO-SGD OBJECTIVE

In this section, we will give a formal proof of Thm. 3.5 (objective of INO-SGD). At iteration $t$, the batch $B_t$ sampled from dataset $D$ with per-owner sampling rates $\mathbf{q}$ is equivalent as a batch sampled from $\widehat{D}$ with uniform sampling rate $\hat{q}$. Let $b$ denote the (expected) batch size of $B_t$. For simplicity assume $C = 1^{10}$ and $\gamma$ is an integer[11]. The expectation of summed gradient (omitting gradient clipping and noise temporarily) is equal to

$$\mathbb{E}_{B_t}[\mathbf{G}_t] = \sum_{k=1}^{K} \Pr\left[\hat{d}_k \in B_t\right] \cdot \left(\Pr[\pi(k) \in \gamma\text{-tail}] \cdot \text{corresponding } \rho + \Pr[\pi(k) \notin \gamma\text{-tail}]\right)$$

$$\cdot \nabla \ell_{\pi(k)}(\boldsymbol{\theta})$$

$$= \sum_{k=1}^{K} \hat{q} \cdot \left( \sum_{i=0}^{\gamma-1} \binom{K-k}{i} \hat{q}^i (1-\hat{q})^{K-k-i} \int_{\gamma-(i+1)}^{\gamma-i} f_{\text{tg}} \, \mathrm{d}c + \right.$$
$$\left. \sum_{i=\gamma}^{K-k} \binom{K-k}{i} \hat{q}^i (1-\hat{q})^{K-k-i} \right) \cdot \nabla \ell_{\pi(k)}(\boldsymbol{\theta}).$$

The inner bracketed sum is obviously larger for smaller $k$ and it is always at most 1. By setting $w_k = \frac{K\hat{q}}{b}(\sum_{i=0}^{\gamma-1} \binom{K-k}{i} \hat{q}^i (1-\hat{q})^{K-k-i} \int_{\gamma-(i+1)}^{\gamma-i} f_{\text{tg}} \mathrm{d}c + \sum_{i=\gamma}^{K-k} \binom{K-k}{i} \hat{q}^i (1-\hat{q})^{K-k-i}) \cdot \nabla \ell_{\pi(k)}(\boldsymbol{\theta})$, we show that the expectation of $\frac{\mathbf{G}_t}{b}$ is equal to the gradient of the INO-SGD objective $\mathcal{L}_{f_{\text{tg}}}\left(\boldsymbol{\theta}, \widehat{D}\right)$. Therefore, we have proven Thm. 3.5.

The fact that vanilla SGD with gradient clipping optimizes the SGD objective function can be generalized to INO-SGD as a similar first-order optimization technique. Similar to vanilla clipped SGD, INO-SGD with clipping and noise addition also converges to the optimum of the stated objective while suffering from an unavoidable bias.  $\square$

### C.3.4  BENEFITS OF INO-SGD OBJECTIVE

In the main paper we give some brief summaries of interpretations of INO-SGD's objective. Here we give more details and discuss them further:

- We mentioned in the main paper that the INO-SGD objective can be viewed as the mixed conditional value-at-risk (mixed CVaR) of the loss function $\ell$, when $\ell$ is viewed as a random function of $\hat{d} \in \widehat{D}$. Specifically, CVaR and mixed CVaR are concepts from risk theory. The $\lambda$-level CVaR refers to the average loss when only the data with top $100\lambda\%$ losses remain in the dataset (i.e., $100\lambda\%$ worst-case tail of losses) and the mixed CVaR represents a weighted average of CVaRs at different $\lambda \in (0, 1]$ (i.e., different severity of worst-case). Thus, the trained model will improve the performance on data with the larger losses which are likely to belong to more private owners.
- We also mentioned that the INO-SGD objective can be viewed as an objective that better corrects for utility imbalance than average loss when minimized and is less affected by outliers/noises with large loss than the *maximal loss*. This is shown empirically in Fig. 11(h) and Fig. 14(h).
- An alternative interpretation is through *distributionally robust optimization* (DRO) (Goh & Sim, 2010). DRO studies how to preserve robust model performance if the actual data distribution during ML model deployment is different from the training data distribution. To achieve this, one typical solution is to consider distributions of training data that are different from the training set itself and train a model that can minimize the maximum loss among all these distributions.

---

[10]In DP analysis, only the ratio between clipping threshold and noise scale, called *noise multiplier*, matters.

[11]$\gamma$ can always be rounded up to the nearest integer by lengthening the tail.

Specifically, the INO-SGD loss $\mathcal{L}_{f_{\text{tg}}}\left(\boldsymbol{\theta}, \widehat{D}\right)$ recovers the DRO objective

$$\inf_{\boldsymbol{\theta}} \sup_{\mathbf{p} \in \mathcal{P}} \sum_{k=1}^{K} p_k \ell_k(\boldsymbol{\theta}),$$

where *uncertainty set* $\mathcal{P} = \{\mathbf{p} : \text{sort}(\mathbf{p}) \preceq \mathbf{w}; \sum_k p_k = 1; \mathbf{p} \geq \mathbf{0}\}$ denotes the set of possible true probabilities of each datum in $\widehat{D}$ occurring in the underlying distribution, $\text{sort}(\mathbf{p})$ denotes the sorted sequence of $\mathbf{p}$ in descending order and $\preceq$ denotes a partial order over decreasing sequences of length $K$ defined as $\mathbf{p} \preceq \mathbf{p}' \Leftrightarrow \forall k \in [K] \left[\sum_{j=1}^{k} p_j \leq \sum_{j=1}^{k} p'_j\right]$. In particular, the cardinality of the uncertainty set $\mathcal{P}$ increases and the trained model becomes more distributionally robust when the learner chooses a larger length of tail $\gamma$ or faster decaying tail importance function $f_{\text{tg}}$ (e.g, large $\alpha$ and small $\beta$ when using a Beta distribution). This explains why our INO-SGD can give better results than the vanilla IDP-SGD when IDP-induced utility imbalance occurs.

## D ADDITIONAL EXPERIMENT RESULTS

The experiments are run on the following machines:

- Ubuntu 18.04.5 LTS, Intel(R) Xeon(R) Gold 6226R (2.90GHz) with NVIDIA GeForce RTX 3080 GPUs (CUDA 11.3);
- Ubuntu 22.04.3 LTS, AMD EPYC 7763 with NVIDIA L40 GPUs (CUDA 12.3).

All the programs are written in Python and executed using Anaconda. The implementation details and hyperparameters used can be found from our codes in the supplementary materials.

### D.1 SETUP

#### D.1.1 DATASETS AND DATA OWNERS

We empirically evaluate our proposed method INO-SGD on various tabular, vision and language datasets. For each dataset, we first split the training set into $N$ disjoint subsets representing the datasets owned by $N$ data owners, in a way that data within each subset share similar traits and data across different subsets have different traits (e.g., we can split by classes or categories) to simulate the different subsets that could have different privacy preferences. We then assign each data owner a unique privacy budget ranging from low (private) to high (less private). Besides the standard settings (**MN**, **CM**, **CL**, **CH**, **CF**, **SU** (explained in detail below)), we consider the following special settings:

- **Real-world clinical datasets** where owners of (likely) positive-case data have stronger privacy requirements, which supports our motivation (**CA**, **PN**);
- **Noisy datasets** with random flipped labels (**CFN**);
- Each data owner possesses 1 datum and has its own privacy budget. Data owners with data from the same group have similar privacy budgets sampled from the same distribution with a unique mean associated with the group. This resonates with real-life scenarios where **data owners with sensitive data in general have less private budgets but can vary** (**CS**, **CFS**);
- **Sensitive data occasionally have weaker privacy requirements and vice versa** (**COL**, **COM**);
- Each owner has a fixed privacy budget but they hold a randomly sampled class in each experiment run. This can **eliminate the potential impact of class complexities on per-owner utility** (**CR**).

The details on our datasets and how we set the data owners are described below:

- **CA**: We use the UCI **CA**rdiotocography dataset (Campos & Bernardes, 2000), which contains 2126 tabular data of fetal cardiotocograms (CTGs) from 3 classes (NORMAL, SUSPICIOUS, PATHOLOGICAL) representing whether a fetus is healthy or not. We assign privacy budgets 3 (most private), 4 and 5 (least private) to pathological, suspicious and normal fetuses. Tolerance $\delta_n = 10^{-5}$ for all owners $n \in [N]$.
- **PN**: We use the **Pn**eumonia dataset (Kermany et al., 2018), which contains chest X-ray images from 3 classes (NORMAL, BACTERIALPNEUMONIA and VIRALPNEUMONIA). We randomly pick 1341 data from each class as the training set. We set 2 data owners: Owner 1 possesses NORMAL data and requires $(5, 10^{-5})$-DP (less private), while Owner 2 possesses BACTERIALPNEUMONIA and VIRALPNEUMONIA patients' data and requires $(0.5, 10^{-5})$-DP (more private).

- **MN**: We use the **MN**IST dataset (LeCun et al., 1998), whose training set contains 60000 grayscale images of handwritten digits from 0 to 9. In our experiment, we suppose that there are $N = 2$ data owners, with one possessing digits from 0 to 4 (30596 data) and another possessing digits from 5 to 9 (29404 data). To simulate individualized privacy preferences, we assign privacy budgets $\epsilon_1 = 0.1$ (private) and $\epsilon_2 = 1$ (less private) and tolerance $\delta_1 = \delta_2 = 10^{-5}$.
- **CM**: We use the **C**IFAR-10 dataset (Krizhevsky & Hinton, 2009), whose training set contains 50000 images of objects from 10 classes (e.g., CAT, DOG). We set $N = 10$ data owners, where owner $n$ possesses data from class $n - 1$. **CM** refers to a **M**ore private setting where we uniformly assign privacy budgets $\epsilon_n$ ranging from 0.5 (most private) to 5 (least private) and tolerance $\delta_n = 10^{-5}$ for all $n \in [N]$.
- **CL**: Similar to **CM**, but considers a **L**ess private setting where we uniformly assign privacy budgets $\epsilon_n$ ranging from 6 (most private) to 15 (least private) and tolerance $\delta_n = 10^{-5}$ for all $n \in [N]$.
- **CR**: Also similar to **CM**, but considers each data owner holding a **R**andomly sampled class in each run. Their privacy preferences remain the same. This could eliminate the impact of inherent difference in class complexities.
- **CS**: We use **C**IFAR-10 and consider the case where each owner possesses only 1 datum and owners possessing data from the same class have **S**imilar (but not same) privacy preferences. Specifically, each class's mean privacy budget uniformly ranges from 0.5 (most private) to 5 (least private), and each owner's privacy budget is uniformly sampled from class mean $\pm 0.25$.
- **COL**: We use **C**IFAR-10 and consider the case where occasionally some sensitive data have weaker privacy requirements and vice versa, that is, there are **O**verlaps among data owners. **COL** considers **L**ess overlaps where each owner possesses 90% data from one class and 10% data from any other classes.
- **COM**: Similar to **COL**, but considers **M**ore overlaps among data owners where each owner possesses 70% data from one class and 30% data from any other classes.
- **CH**: We use the **C**IFAR-**100** dataset (Krizhevsky & Hinton, 2009), whose training set contains 50000 images of objects from 100 classes (e.g., FLATFISH, BICYCLE). Every 5 classes come from one category (e.g., FLATFISH comes from the FISH category). We set $N = 100$ data owners, where owner $n$ possesses data from class $n - 1$. We uniformly assign privacy budgets $\epsilon_n$ ranging from 1 (most private) to 10 (least private) and tolerance $\delta_n = 10^{-5}$ for all $n \in [N]$.
- **CF**: We consider $N = 2$ data owners, each possessing data from one **category** (i.e., 5 classes). Specifically, Owner 1 owns the FISH category containing AQUARIUMFISH, FLATFISH, RAY, SHARK and TROUT classes, while Owner 2 owns the VEHICLES1 category containing BICYCLE, BUS, MOTORCYCLE, PICKUPTRUCK and TRAIN classes. Owner 1 requires $(1, 10^{-5})$-DP (private) while Owner 2 requires $(5, 10^{-5})$-DP (less private). We denote this as **C**IFAR-100-**FV**.
- **CFS**: Similar to **CF** and **CS**, each category's mean privacy budget is 1 and 5 respectively and each owner's privacy budget is sampled from the Gaussian distribution with mean = its category mean and standard deviation = 0.5.
- **CFN**: We use **C**IFAR-100-**F**V and consider the case where the dataset is more **N**oisy. Specifically, we introduce 10% random label flips.
- **SU**: The **SU**rnames dataset (Robertson, 2017) (**SU**) contains 20074 surnames from 18 different countries, with Country 4 being the majority country containing 9408 surnames. We consider $N = 2$ data owners, including a less private data owner who possesses data from the majority country and requires $(3, 10^{-5})$-DP and a more private data owner who possesses data from other countries and requires $(2, 10^{-5})$-DP. This is because data from other countries might be more difficult to obtain compared to the majority country with plenty of data.

For all the datasets listed above, we use their default split of training and validation sets in our experiments.

### D.1.2 MODELS AND TRAINING

In the main paper, for vision datasets (MNIST, CIFAR-10, CIFAR-100 and CIFAR-100-FV), we use a set of convolutional neural network (CNN) models proposed by Papernot et al. (2021) which cater to DP-ML and are widely used in existing literature (Tramèr & Boneh, 2021; Boenisch et al., 2024). These CNN models are smaller than popular models such as VGG (Simonyan & Zisserman, 2014) or ResNet (He et al., 2016), but are more suitable for DP-ML because the (expected) magnitude of added Gaussian noise (drawn from $\mathcal{N}(0, \sigma^2 \mathbf{I}_{r \times r})$) increases with the number of model parameters $r$. Also, they use Tanh activation functions which are shown to perform better in DP-ML. We

supplement these CNN experiments by considering more complex models and embeddings such as ResNet-18 (He et al., 2016) and SimCLRv2 (Chen et al., 2020a). For language dataset (Surnames), we use a character-level long short-term memory (LSTM) model from Boenisch et al. (2024), which is a type of recurrent neural network (RNN) model that treats each surname as a sequence of characters (i.e., letters and special characters).

The details on all the models used are listed below:

- **MLP**: Refers to a simple **M**ulti-**L**ayer **P**erceptron (MLP) model with 3 hidden layers each containing 47 neurons and ReLU activations.
- **PC**: Refers to **P**apernot et al. (2021)'s **C**NN models. For MNIST, this refers to a CNN model with 5 hidden layers (Conv-MaxPool-Conv-MaxPool-FC) with Tanh activation at each convolutional and fully connected layer; for CIFAR-10, CIFAR-100 and CIFAR-100-FV, this refers to a **C**NN model with 10 hidden layers (Conv-Conv-MaxPool-Conv-Conv-Maxpool-Conv-Conv-Maxpool-FC) with Tanh activation at each convolutional and fully connected layer.
- **RN**: Refers to a **R**es**N**et-18 (He et al., 2016) model trained from scratch.
- **SC**: Refers to a pretrained **S**im**C**LRv2 model (`r50_2x_sk1`) (Chen et al., 2020a) which we use to obtain embeddings and train a linear model upon them.
- **EN**: Refers to a pretrained **E**fficient**N**et B1 model (Tan & Le, 2019) which we used to obtain embeddings and train an MLP with 1 hidden layer containing 512 neurons and ReLU activations.
- **LS**: Refers to a character-level **LS**TM model (Embedding-LSTM-FC). Each input is mapped to a 64-dimensional embedding. The LSTM has a hidden size of 128, with internal linear layers projecting to 512 dimensions.

We will label each experiment in the format [dataset]-[model] (e.g., **MN-PC**). For each dataset-model combination, we choose from the two variants of IDP-SGD, SAMPLE and SCALE, at random since they usually have similar performances and it is not yet clearly understood which variant has advantages over another. Specifically, we use SAMPLE for Cardiotocography, MNIST, CIFAR-100, CIFAR-100-FV and Surnames datasets, and SCALE for Pneumonia, CIFAR datasets. The same setting is used when we test our method INO-SGD. Furthermore, we repeat each experiment for 5 times and show the means and standard deviations of results to tackle randomness.

### D.1.3    CHOICE OF BASELINE

Since this is the first work that recognizes and addresses IDP-induced utility imbalance, there is no prior work that mitigates the problem and is suitable as baseline. Therefore, we compare against the most important baseline IDP-SGD in all experiments.

In Sec. 3.1 and App. C.2.1, we have explained why existing techniques addressing utility imbalance (either in non-private or private setting) neither satisfy IDP nor can be easily adapted to do so. To add on, we explain why common techniques to address data imbalance in the **non-private** setting are **not suitable** in the IDP setting:

- **Strategically dropping data from the less private data owners before training** (e.g., with data pruning): This does not satisfy IDP. For neighboring datasets $D$ and $D^d$, if the dropping algorithm is deterministic, the resulting datasets $D'$ and $D^{d'}$ may not be neighboring, thus can be identified by attackers even if IDP-SGD is applied to $D'$ and $D^{d'}$; if the dropping algorithm is randomized, the resulting datasets $D'$ and $D^{d'}$ may not follow a similar distribution, thus can be identified too. Therefore, it remains non-trivial to design a dropping algorithm that preserves IDP, which is beyond the scope of this work.
- **Randomly dropping data from less private owners before training**: This has two issues. Firstly, for multiple data owners and privacy budgets, it is unclear how to set (i) the cutoff between less and more private owners and (ii) how much data to drop from each less private owner. For (i), for example, for 10 owners with $\epsilon_n \in [0.5, 5]$ evenly, it is unsuitable to directly use the mean 2.25 as the cutoff since the impact of privacy does not scale linearly with privacy budgets and is hard to quantify *a priori*; for (ii), dropping equal number of data from owners with $\epsilon_n = 3$ and $\epsilon_n = 5$ is unsuitable, as there is still imbalance between these two and dropping data from $\epsilon_n = 3$ incurs different utility loss than $\epsilon_n = 5$. Due to (i) and (ii), finding the optimal number of data to drop from each owner requires $O(N)$ hyperparameters, making it unsuitable as baseline. Secondly, this method may under-utilize the privacy budgets and data from the less private data owners. Due to the privacy-utility tradeoff (Makhdoumi & Fawaz, 2013), the overall utility

would be lower than our current baseline IDP-SGD which fully utilizes the privacy budgets of each owner. Below we show the overall validation accuracy (mean (standard deviation)) when we drop $10\%$ to $50\%$ data from less private owners (cutoff = mean):

Table 3: **Random undersampling degrades overall validation accuracy.**

| Setting | IDP-SGD | Drop $10\%$ | Drop $20\%$ | Drop $30\%$ | Drop $40\%$ | Drop $50\%$ | INO-SGD |
|---------|---------|-------------|-------------|-------------|-------------|-------------|---------|
| **CF-PC** | 46.4 (0.53) | 43.88 (0.77) | 43.56 (0.99) | 42.9 (1.67) | 42.06 (1.88) | 40.72 (2.51) | 48.74 (0.65) |
| **CL-PC** | 63.35 (1.28) | 62.48 (1.74) | 61.64 (2.31) | 60.41 (3.31) | 58.83 (4.11) | 56.47 (5.06) | 66.73 (0.24) |

- **Reducing less private owners' privacy budgets**: Similar to the above, it is still unclear how to set the cutoff and how much privacy budgets to reduce from each less private owner. Moreover, this method under-utilizes the privacy budgets and data from the less private owners. Due to the privacy-utility tradeoff (Makhdoumi & Fawaz, 2013), the overall utility would be lower than our current baseline IDP-SGD which fully utilizes the privacy budgets of each owner. We validate the above in App. D.4.3. Note that this includes other methods like reducing less private owners' sampling rates in SAMPLE and clipping thresholds in SCALE.

## D.2 Addressing Utility Imbalance

In the main paper, Fig. 5(a), (b) and (c) are under the **MN-PC**, **CL-PC** and **CF-PC** setting respectively; Fig. 6(a) and (b) are under the **CM-PC** and **CH-PC** setting respectively.

In Fig. 11, we present more experiments under various settings and more complex models (e.g., **SU-LS**, **CM-RN**, **CM-SC**) that illustrate how INO-SGD still addresses utility imbalance induced by IDP-SGD. In particular, for experiments with more data owners (e.g., **CH-PC**), we show 3 or 4 data owners whose privacy preferences range from low to high for a clearer visualization. We note the following points:

- Fig. 11(a)-(d) are real-world clinical datasets where owners of (likely) positive-case data have stronger privacy requirements, which reinstates our motivation that utility imbalance is undesirable for subsequent users of the trained ML model.
- Fig. 11(h) considers each category as a group since each owner's data contain label noises. It shows that INO-SGD's performance is robust to label noises.
- Fig. 11(q) and (r) consider possible real-world settings where each data owner only possesses 1 datum and can choose its own privacy budgets. Owners who possess sensitive data can generally prefer stronger privacy but have small variations among themselves. The two results show that INO-SGD is still effective under such settings.
- Fig. 11(s) and (t) consider possible real-world settings where occasionally sensitive data have weaker privacy requirements and vice versa. The two results show that INO-SGD is still effective under such settings.

*Remark.* Similar to MNIST (**MN-PC**), Pneumonia and CIFAR-10 with extracted embeddings (**PN-EF/CM-SC**) are also relatively easy-to-learn tasks for which MID happens at an earlier stage of training. In contrast, in more difficult tasks such as CIFAR-10 (**CM-PC/CM-RN**) and CIFAR-100 (**CH-PC**), privacy budgets are used up before MID even ends. This reinstates the importance of considering the entire learning dynamics and the benefit of our INO-SGD algorithm.

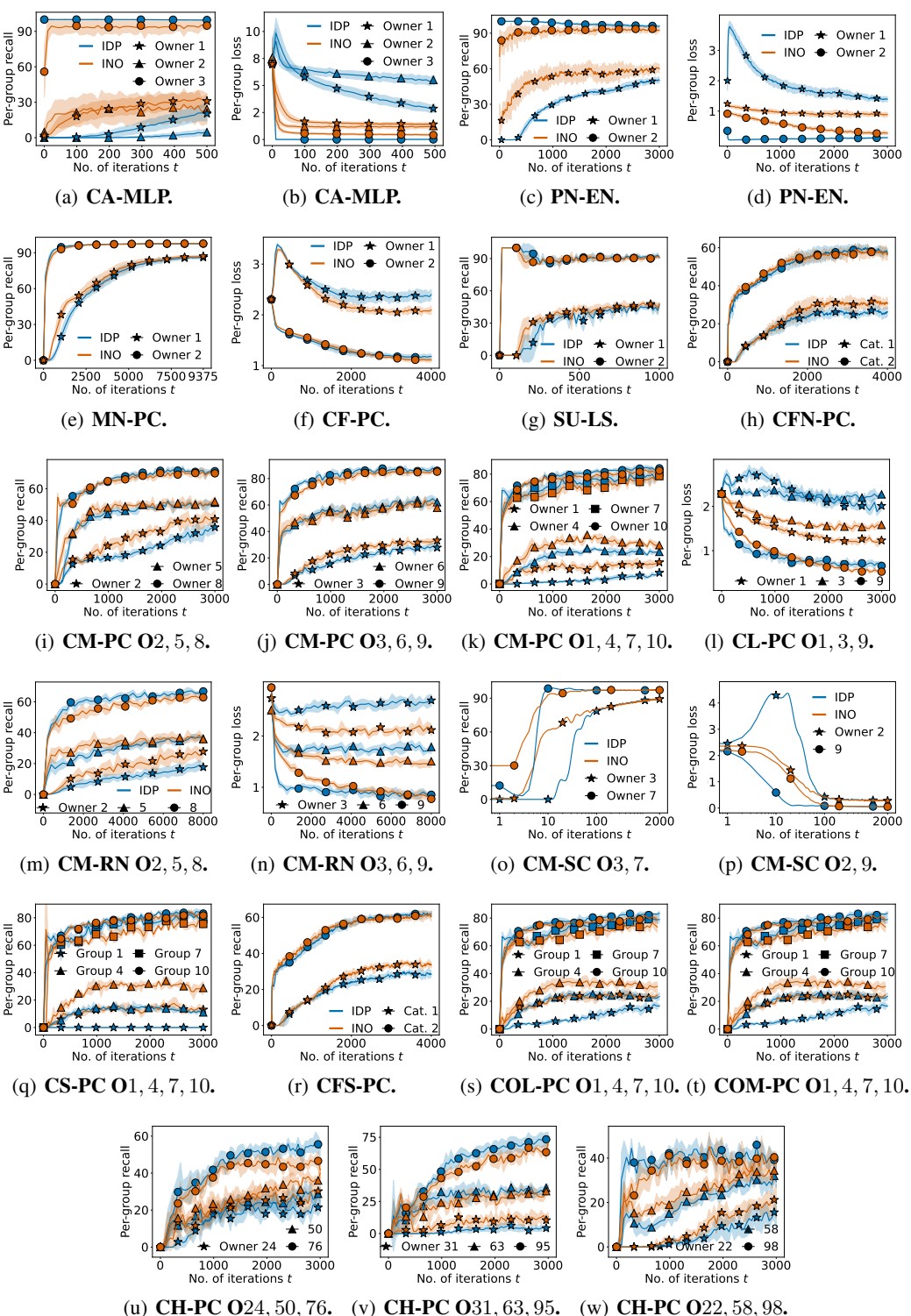

Figure 11: **Comparison of per-owner model utility between IDP-SGD and INO-SGD.** Here **O**2, 5, 8 refers to Owners 2, 5 and 8 and likewise for the others. INO-SGD consistently improves the model performance for the more private data owners while sometimes conservatively lowering the performance for the less private data owners. In general, it results in a more balanced learning dynamics.

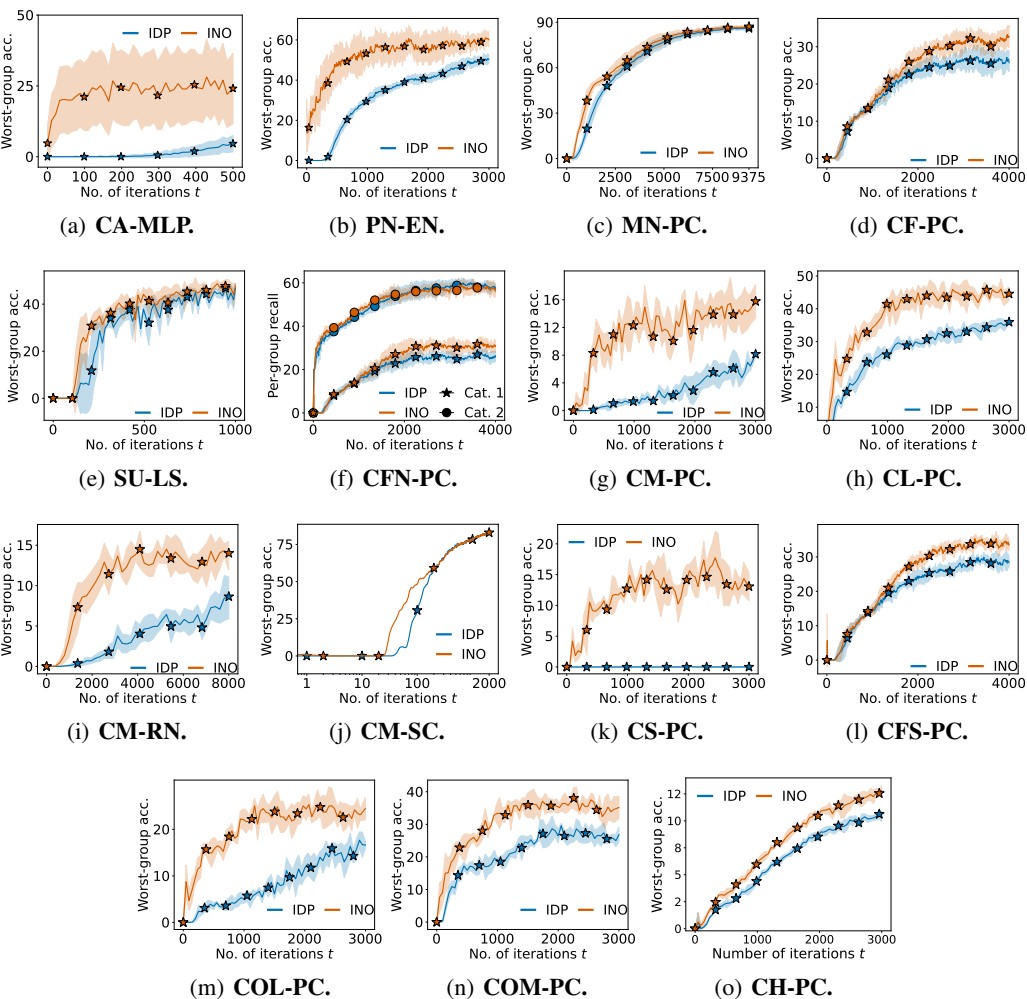

Figure 12: **Comparison of worst-owner accuracy between IDP-SGD and INO-SGD.** INO-SGD consistently improves the model performance for the most private data owners and thus address utility imbalance.

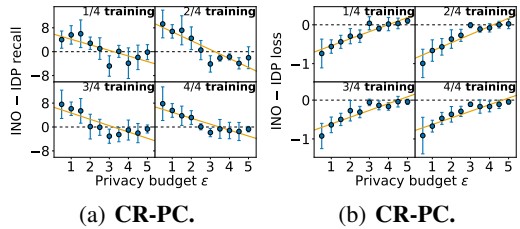

(a) **CR-PC.**     (b) **CR-PC.**

Figure 13: **INO-SGD's per-group utility minus IDP-SGD's.** Model utilities for the more private owners show higher increase.

Another way to assess utility imbalance is through the *worst-group accuracy* (i.e., the accuracy of the worst group(s) at different stages of training). In Fig. 12, we demonstrate that INO-SGD consistently improves the worst-group accuracy over IDP-SGD. Specifically, for **CH-PC** (Fig. 12(d)), we show the average of the worst 50% owners due to a large number of small privacy budgets and difficult classes.

In Fig. 13, we repeat the experiments in Fig. 6 (graphs of changes in model performance from IDP-SGD to INO-SGD against different privacy budgets) on the **CR-PC** setting, where each of the 10 data owners has a fixed privacy budget but receive a randomly sampled class in each random run. This could eliminate the impact of inherent different in class complexities. We can see that INO-SGD still effectively improve the utility on the more private owners.

## D.3    Preserving Overall Model Utility

In the main paper, Fig. 7(a), (b) and (c) are under the **MN-PC**, **CL-PC** and **CF-PC** setting respectively.

In Fig. 14, we present more experiments that illustrate how INO-SGD improves/preserves overall model utility as compared with IDP-SGD. In particular, we use balanced accuracy for the Surnames dataset (**SU-LS**) and Cardiotocography dataset (**CA-MLP**) since their validation sets are highly imbalanced. It can be seen that INO-SGD demonstrates the same trends under the various settings noted in App. D.2.

## D.4    Other Empirical Benefits and Discussions

In this section, we demonstrate and discuss several other benefits of INO-SGD.

### D.4.1    Better Within-Owner Learning Dynamics

We notice that the learning curve within a single data owner is not smooth and demonstrates certain fluctuations when it holds several different groups of data. In real life, this represents groups with data of different distribution but similar privacy preferences (e.g., two types of one stigmatized disease). To investigate it, we analyze the per-class recall for different classes within the same data owner in **MN**, **SU** and **CF** and show it in Fig. 15. Although classes/groups belonging to the same data owner share the same privacy budget, they may have different learning complexities and hence different learning dynamics and duration of the MID effect. For example, in Fig. 15(c), Class 0 is much easier to learn than Class 3 (possibly because 0 has a more distinguishable shape). Thus, the MID effect of Class 0 (around iteration 0 to 100) is much shorter than that of Class 3 (around iteration 0 to 3000).

This experiment also demonstrates another benefit of INO-SGD: it also improves the learning dynamics of different groups within the same data owner. This is because an easy-to-learn group within a private data owner can also be learnt fast and its loss will decrease quickly. Samples from such group will then be down-weighted by the INO-SGD algorithm so that we can focus more on reducing the loss for the more difficult samples from the private data owner. For example, in Fig. 15(c) again, the MID effect with regards to class 3 is significantly improved under INO-SGD (shortened to around iteration 0 to 1500). This shows that INO-SGD can not only address IDP-induced utility

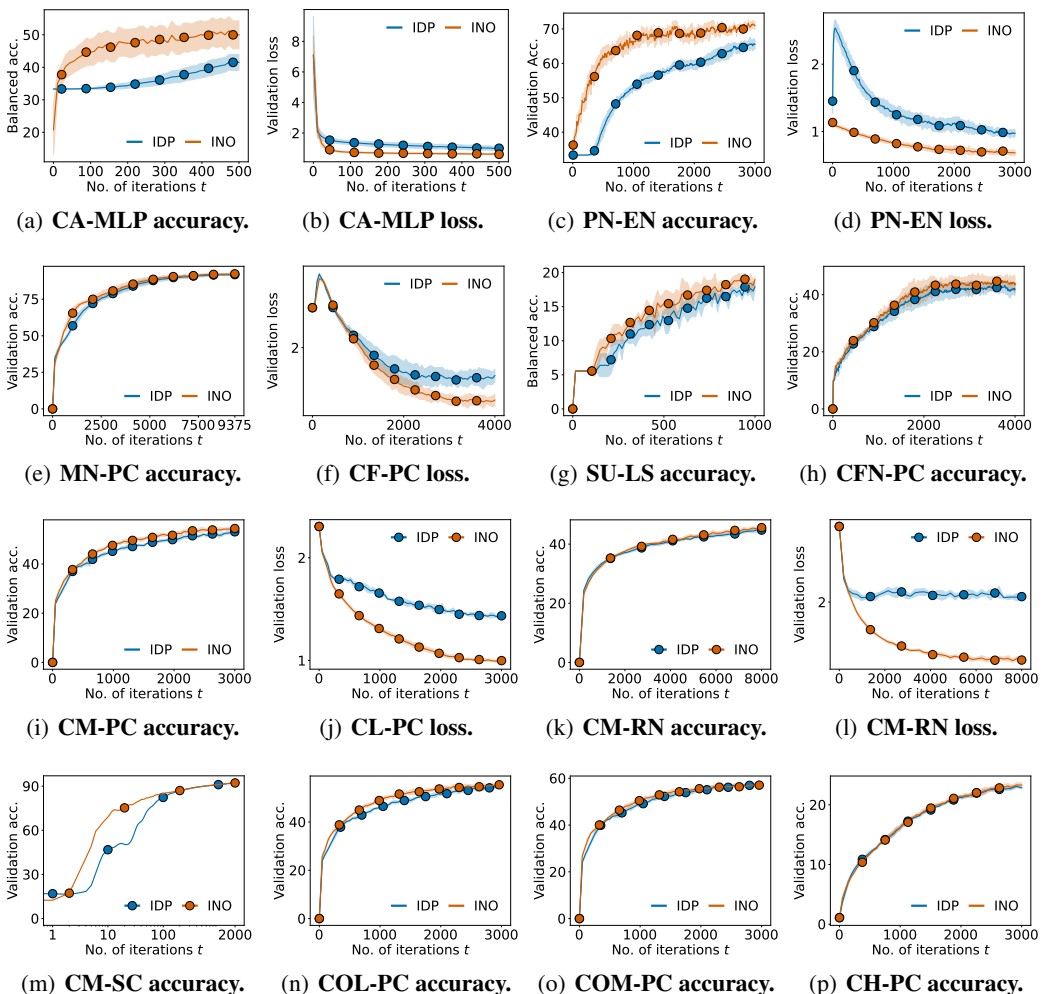

Figure 14: **Comparison of overall model performance between IDP-SGD and INO-SGD.** INO-SGD consistently improves the overall model performance.

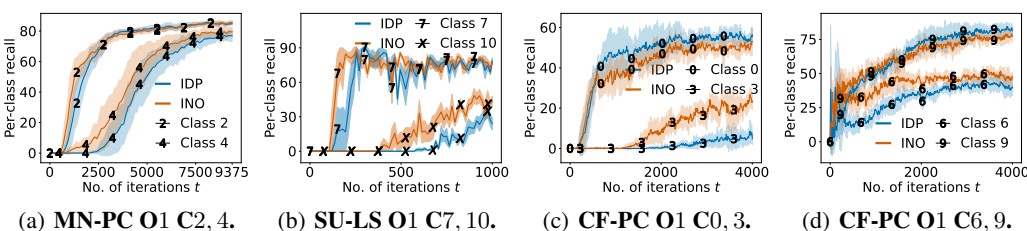

Figure 15: **Per-class learning dynamics within a data owner.** Here **O**1 **C**2, 4 refers to Owner 1 Classes 2 and 4, and likewise for the others. Different classes within a data owner are learnt with different dynamics. INO-SGD improves such within-owner learning dynamics too.

imbalance, but also **mitigate the inherent imbalance caused by different learning complexities among groups**.

### D.4.2    COMPONENTS OF INO-SGD

The INO-SGD algorithm has two main components: the TIF and the ordering of data used. In this section, we conduct ablation studies by removing/varying each component to assess its impact. In **(I)** to **(III)**, we demonstrate that **TIF is important**. Moreover, **INO-SGD is robust to the choice of hyperparameters** so that it is easy for practitioners to apply it in real life. As mentioned in Sec. 3.2, the hyperparameters of TIF $f_{\text{tail}}$ include the tail length $\gamma$ and the form of $f_{\text{tail}}$. Therefore, we run the INO-SGD algorithm using different values of $\gamma$, parameters $\alpha$ and $\beta$ of Beta distribution and also try different function forms such as the step function mentioned in App. C.2.5. In **(IV)**, we verify the benefit of using a descending order of loss by comparing it with a random and an ascending order of loss.

**(I) Tail length, $\gamma$.**    In Fig. 16, we show how the learning dynamics change as we change the tail length $\gamma$. It can be seen that in general, INO-SGD shows an improved learning dynamics when we set $\gamma$ to be less than half of the expected sum of clipping thresholds calculated by $\sum_{n \in [N]} |D_n| q_n C_n$ (which equals around 256 for CIFAR-100-FV (**CF-PC**) and 1987 for CIFAR-10 (**CM-PC**)). If we set $\gamma$ to be even larger, there will be too little good information left in the batch and the summed gradient will get more noisy due to the added Gaussian noise. It will start trading overall utility for better balance.

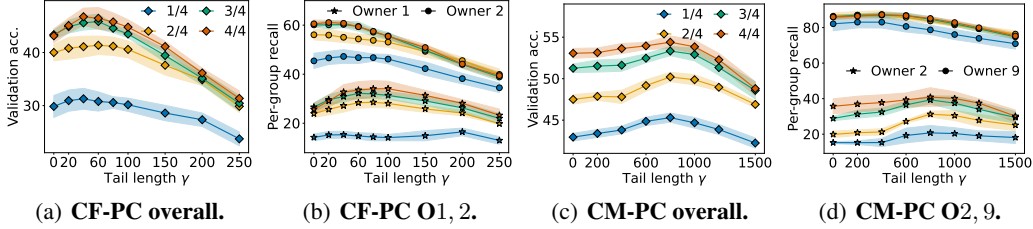

(a) **CF-PC overall.**    (b) **CF-PC O$1, 2$.**    (c) **CM-PC overall.**    (d) **CM-PC O$2, 9$.**

Figure 16: **INO-SGD is robust to choice of $\gamma$ for a limited range.** After that, it starts to trade overall utility for better balance. In particular, we plot the model performances at $1/4, 2/4, 3/4$, and $4/4$ of the entire training stage (in terms of iterations). For the CIFAR-10 dataset, we only show 2 data owners for better visualization.

**(II) $\alpha$ and $\beta$ in Beta distribution.**    In Fig. 17, we show how the learning dynamics change as we change the value of $\alpha$ and $\beta$ in Beta distribution when we set the tail importance function $f_{\text{tail}}$. All choices of $\alpha$ and $\beta$ enable INO-SGD to produce better learning dynamics than IDP-SGD. Specifically, when $\alpha$ is smaller and $\beta$ is larger, INO-SGD will become more and more similar to IDP-SGD because fewer gradients are down-weighted.

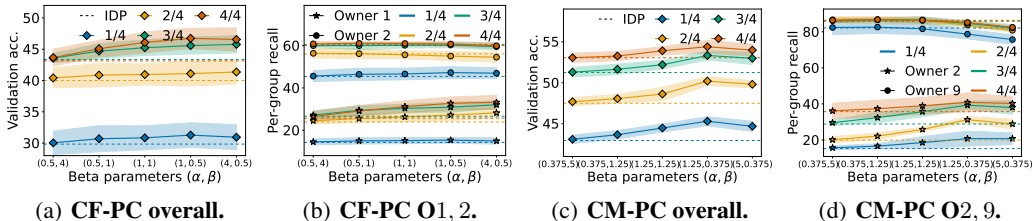

(a) **CF-PC overall.**    (b) **CF-PC O$1, 2$.**    (c) **CM-PC overall.**    (d) **CM-PC O$2, 9$.**

Figure 17: **INO-SGD is robust to the choice of $\alpha$ and $\beta$ if the length of the tail is chosen adequately.** The dashed line represents our baseline IDP-SGD. In particular, a small $\alpha$ and large $\beta$ will cause INO-SGD's performance to be similar to IDP-SGD and vice versa.

**(III) Alternative TIF forms.**    In this section, we consider the alternative function form of tail importance function $f_{\text{tail}}$, a step function, as described in App. C.2.5. Specifically, we use 4 steps with a constant step length (i.e., instead of setting $\gamma$, $\alpha$ and $\beta$, we only need to set the constant step length now). We vary the value of the constant step length to see whether our observed trends are still relevant when we use this alternative function form. The results are shown in Fig. 18, and in

particular, we recover IDP-SGD when we set the step length as $0$. It can be seen that as long as we choose a reasonable step length (one that is not too long, or more precisely the sum of four steps should not exceed half of $\sum_{n\in[N]}|D_n|q_n C_n$ (same as how we choose $\gamma$ previously)), we can get a better learning dynamics than IDP-SGD. Therefore, we can conclude that our INO-SGD is robust to the choice of tail importance functions as long as we choose a reasonable tail length $\gamma$. We still recommend using the TIF based on Beta distribution because it provides a flexible range of TIF shapes using only two parameters.

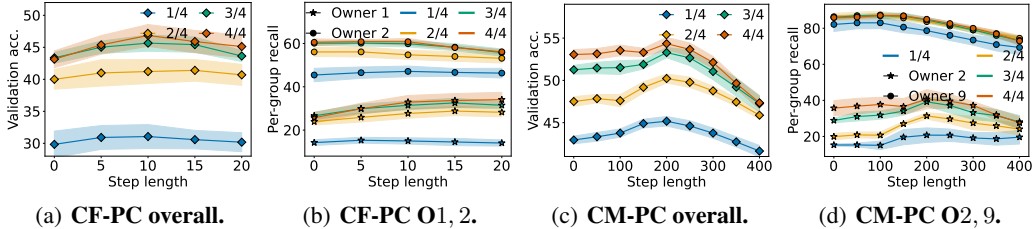

(a) **CF-PC overall.**      (b) **CF-PC O**$1, 2$**.**      (c) **CM-PC overall.**      (d) **CM-PC O**$2, 9$**.**

Figure 18: **INO-SGD is robust to forms of the tail importance function $f_{\text{tail}}$.** In this set of experiments, we use the step function with constant step length as described in App. C.2.5: $1/2$ for the first step length, $1/4$ for the second step length, $1/8$ for the third step length, and $0$ for the last step length. Similar trends to our main experiments can be observed.

**(IV) Impact of order.** In Fig. 19, we show the benefit of using a descending order of loss by replacing it with (1) a random order (INO (random)) and (2) a reverse (ascending) order (INO (reverse)). The overall results in Fig. 19(a) and the per-owner results in Fig. 19(b) and Fig. 19(c) all show that a descending order is crucial, which justifies our argument that the loss values indicate the importance of data. In particular, using a descending order of loss not only benefits the performance of the more private data owners (e.g., Owners $1$ and $2$), but also benefits the performance of the less private data owners (e.g., Owners $7$ and $8$) because the more difficult data within them are assigned higher importance too.

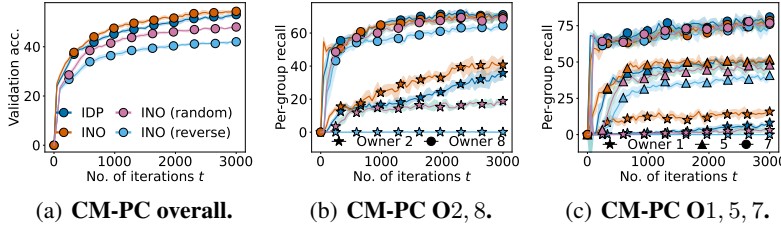

(a) **CM-PC overall.**      (b) **CM-PC O**$2, 8$**.**      (c) **CM-PC O**$1, 5, 7$**.**

Figure 19: **Benefit of using a descending order of loss.**

### D.4.3 PARETO SUPERIORITY

Beyond the region where the model owner can simultaneously correct utility imbalance and improve/preserve overall model utility (which only INO-SGD can achieve), the model owner can also choose to set the TIF more aggressively (e.g., downweighting more of the less important gradients by setting a larger tail length $\gamma$) in order to trade overall utility to further correct for utility imbalance. In this use case, method A is said to *Pareto-dominate* method B if at any level of overall utility, method A corrects for utility imbalance better than method B; at any level of utility imbalance, method A achieves a higher overall utility than method B. In Fig. 20, we demonstrate that **INO-SGD Pareto-dominates naïve methods that satisfy IDP.** In particular, we control the INO-SGD's tradeoff by using different tail length $\gamma$, and consider the baseline where every data owner decreases their privacy budgets proportionately (e.g. by $10\%, 20\%, 30\%, 40\%$) to match the most private owner, and we use the standard deviation across per-group recalls to measure utility imbalance. Just as the privacy-utility tradeoff (Makhdoumi & Fawaz, 2013) describes, decreasing privacy budgets would incur lower overall utility, so such methods cannot achieve a dual improvement in both utility balance and overall utility as INO-SGD achieves. Even if the model owner is willing to trade some overall utility to further correct for utility imbalance, INO-SGD is a better solution because of its Pareto superiority.

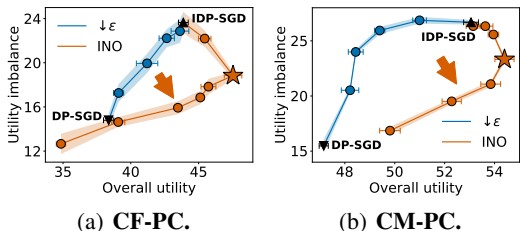

(a) **CF-PC.**        (b) **CM-PC.**

Figure 20: **Pareto superiority of INO-SGD.** ▲ denotes IDP-SGD and ▼ denotes DP-SGD using the strongest privacy. ★ shows our reported results where the model owner attains the largest **dual improvement** in both utility balance and overall utility. The red arrow indicates that INO-SGD Pareto-dominates any method whose tradeoff curve lies to the upper left of INO-SGD's, including the simple baseline. Therefore, INO-SGD offers the model owner a superior way to control the tradeoff between utility imbalance and overall model utility.

*Remark.* The reverse C-shaped region at the right of INO-SGD's tradeoff curve indicates the robustness of INO-SGD discussed in App. D.4.2: in this region, the model owner can correct for utility imbalance without sacrificing any overall utility.

### D.4.4 COMPATIBILITY WITH ADAPTIVE CLIPPING

In App. C.1.2, we note that adaptive clipping methods (Bu et al., 2023; Xia et al., 2023) would further lengthen IDP-induced MID because they make Cond. 2 more difficult to be satisfied, thus worsening IDP-induced utility imbalance. Fig. 21 verifies this and demonstrates that INO-SGD works effectively to address utility imbalance when using adaptive clipping, since INO-SGD can effectively mitigate MID.

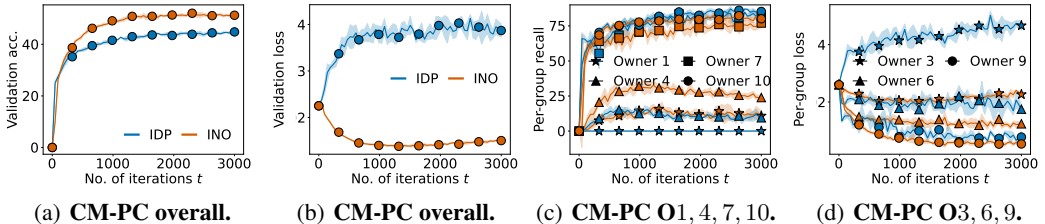

(a) **CM-PC overall.**    (b) **CM-PC overall.**    (c) **CM-PC O**$1, 4, 7, 10$**.**    (d) **CM-PC O**$3, 6, 9$**.**

Figure 21: **INO-SGD is highly compatible with adaptive clipping.** (d) shows that for adaptive clipping Owner 3 suffers from MID till the end, which verifies our theoretical analysis that adaptive clipping could lengthen IDP-induced MID. Since INO-SGD addresses MID and utility imbalance, it significantly improves the performance of such methods.

### D.4.5 PRIVACY ASSESSMENT VIA MEMBERSHIP INFERENCE ATTACK

In Sec. 3.3.1 and App. C.3.1, we theoretically present and prove the privacy guarantees of INO-SGD. In this section, we conduct the *LiRA membership inference attack* (Carlini et al., 2022) to empirically verify the privacy protection of INO-SGD.

We first give a brief introduction to the LiRA method. The goal of LiRA is to infer whether a given datum $d$ is in the training set $D$ (i.e., whether $d$ has been used for training the target model $\mathcal{A}(D)$). Let $\mathcal{M}$ denote the larger pool of data which may or may not be used for training. LiRA firstly trains many *shadow models* via $\mathcal{A}$, each using a subset of $\mathcal{M}$, $M \subseteq \mathcal{M}$. For each trained shadow model $\mathcal{A}(M)$ and the target model $\mathcal{A}(D)$, LiRA then computes a confidence score for each datum $d \in \mathcal{M}$ from its corresponding output logits. Let $z_M(d)$ denote $d$'s confidence score from the model trained on $M$. During the attack, for each datum $d \in \mathcal{M}$, we fit two Gaussian distributions $\mathcal{N}_d^{\text{in}}$ and $\mathcal{N}_d^{\text{out}}$ using the confidence scores $z_M(d)$ from shadow models trained with $d$ (i.e., $d \in M$) and shadow models trained without $d$ (i.e., $d \notin M$) respectively. The LiRA score of $d$ is then calculated using

$$\text{LiRA}(d) = \frac{\Pr_{z \sim \mathcal{N}_d^{\text{in}}}[z = z_D(d)]}{\Pr_{z \sim \mathcal{N}_d^{\text{out}}}[z = z_D(d)]}.$$

Intuitively, the LiRA score represents how different $\mathcal{N}_d^{\text{in}}$ is from $\mathcal{N}_d^{\text{out}}$. If a datum $d$ is really protected by (I)DP, then the LiRA scores between data used for training ($d \in D$) and not used for training ($d \notin D$) should be indistinguishable. Empirically, the true positive rate (TPR) and false positive rate (FPR) should be approximately equal to each other for every chosen threshold.

In this experiment, we follow Boenisch et al. (2024) and train $512$ shadow models via INO-SGD, each using half of the data from each group. In Fig. 22, we show the *receiver operating characteristic* (ROC) curve of IDP-SGD and INO-SGD. We observe that the ROC curves are very close to the line $\text{TPR} = \text{FPR}$, which shows that INO-SGD indeed protects the data well. We also see the effect of IDP: for example, in Fig. 22(b), the ROC curve for the more private group with $\epsilon = 1$ is closer to the line $\text{TPR} = \text{FPR}$ compared with that for the less private group with $\epsilon = 5$, which shows that the more private group indeed receives better privacy protection.

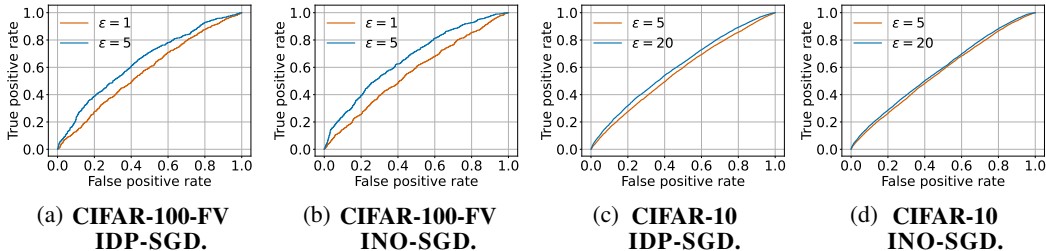

(a) **CIFAR-100-FV IDP-SGD.**  (b) **CIFAR-100-FV INO-SGD.**  (c) **CIFAR-10 IDP-SGD.**  (d) **CIFAR-10 INO-SGD.**

Figure 22: **The privacy of INO-SGD is validated by LiRA membership inference attack.** The AUROC for all owners are close to $0.5$ and the ROC curves for IDP-SGD and INO-SGD are similar. The AUROCs for the more private owner (smaller $\epsilon$) and less private owner (larger $\epsilon$) are respectively (a) $0.571, 0.645$, (b) $0.563, 0.663$, (c) $0.569, 0.623$, and (d) $0.563, 0.580$ for the subfigures.

# E  ADDITIONAL DISCUSSIONS

## E.1  LIMITATIONS

In this section, we explain some possible limitations of the INO-SGD algorithm and discuss their causes and possible future directions.

Firstly, increasing the utility of more private owners might unavoidably reduce that of the less private owners. This is because we have to weaken the dominance of gradients from the less private owners in order to achieve Eq. 5, mitigate MID and BOO, and address IDP-induced utility imbalance. The need for tradeoff is also considered in the Pigou-Dalton principle (Pigou, 1912): if we want to improve the model utility for the more private data owners and hence make the utilities for all owners less imbalanced, we must transfer some resources from the less private owners to the more private ones. To mitigate this limitation and control the tradeoff in the INO-SGD algorithm, the model owner can choose the TIF function $f_{\text{tail}}$ according to their desired utility for each owner.

Secondly, the INO-SGD algorithm can only reduce utility imbalance across data owners to a certain extent without harming the overall model utility. As shown in Fig. 16 and 20, when a more potent TIF function $f_{\text{tail}}$ is chosen (i.e., TIF with longer tail $\gamma$), although the utility imbalance across all owners may be further reduced, the model utility for all data owners starts to drop and so does the overall utility. This can be explained by the IDP-balance-utility tradeoff elaborated in App. E.2: for the same IDP requirements, there exists a tradeoff between overall utility and utility balance. The INO-SGD algorithm is closer to the Pareto frontier of the tradeoff than IDP-SGD, because within a certain range INO-SGD simultaneously improves both overall utility and utility balance. To mitigate this limitation, the model owner can again choose the TIF function $f_{\text{tail}}$ carefully.

Thirdly, in this paper, we do not provide theoretical guarantees on how much model utility can be improved for each data owner without harming the overall utility. This requires a theoretical analysis of the IDP-balance-utility tradeoff which is highly non-trivial and beyond the scope of this paper. Future work can theoretically characterize the Pareto frontier of the IDP-balance-utility tradeoff and find Pareto-optimal solutions. On the other hand, the theoretical scope of this paper is to analyze the cause and impact of IDP-induced utility imbalance we identify (Sec. 3.1 and App. C.1) and provide

the privacy and objective analysis of the INO-SGD algorithm we propose (Sec. 3.3). Nevertheless, our experiments in Sec. 4 and App. D demonstrate the empirical benefits of our INO-SGD algorithm over IDP-SGD.

### E.2 THE IDP-BALANCE-UTILITY TRADEOFF

Our experiment has shown that INO-SGD achieves a better learning dynamics than IDP-SGD, but it seems that the improvement is somewhat limited (i.e., it does not result in equal performance for every data owner), which seemingly makes our contribution limited. However, in this section, we will briefly discuss the IDP-balance-utility tradeoff and justify why it is not possible to achieve equal performance for every data owner without loss of overall model utility.

The disparate impact of DP (Bagdasaryan et al., 2019) has been a well-known drawback that limits the performance of DP-ML models. It refers to the phenomenon that DP-ML models give much lower accuracy for underrepresented groups in the training set, and it is known to be unavoidable due to the nature of DP: since we need to protect the privacy of every datum, we cannot specially make the underrepresented groups identifiable. The only workaround is to undersample the over-represented groups and waste some of their privacy budgets, so as to achieve "similar utility" by giving the overrepresented groups much worse performance. This situation is known as the privacy-disparity-utility tradeoff. In IDP, the situation becomes even worse. In Sec. 3.1, we explain that even if the original training set is balanced and there is no underrepresented group, IDP would forcefully create underrepresented groups because of the individualized privacy preferences. Therefore, we are expected to see a worse privacy-disparity-utility tradeoff in IDP than vanilla DP. Fig. 23 gives an intuitive explanation of such *IDP-balance-utility tradeoff*.

**Our work successfully gets closer to the boundary of the IDP-balance-utility tradeoff**, because we are able to achieve better balance without losing the overall model utility. Nonetheless, we are still facing the IDP-balance-utility tradeoff. In fact, one can refer to Fig. 16(a) and (b) to visualize our contribution to the IDP-balance-utility tradeoff. When the length of tail $\gamma < 100$, we are achieving a better balance without sacrificing overall model performance. This means that the original IDP-SGD does not hit the IDP-balance-utility tradeoff bound while our INO-SGD does better. However, when the length of tail $\gamma \geq 100$, although the utility imbalance is still decreasing, the overall model performance shows a rapid drop too. This is where we reach our Pareto frontier and experience the IDP-balance-utility tradeoff.

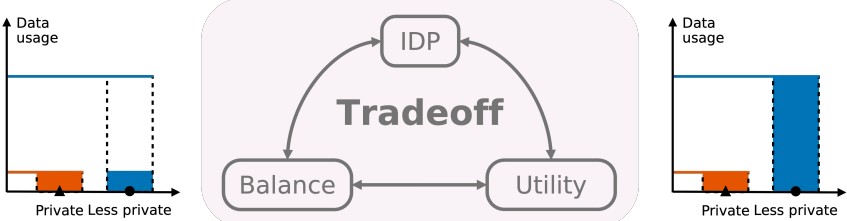

Figure 23: **Intuitive illustration of the IDP-balance-utility tradeoff.** The privacy budgets of two owners constrain how much the model may use their data, so the trained model will either under-utilize data from the less private owners and under-perform (left), or utilize data from both owners unevenly which causes utility imbalance (right).

## F OTHER QUESTIONS

1. **Why do we consider the individualized privacy setting? Why would each owner not choose the maximum privacy level?**

   Research (Berendt et al., 2005; Jensen et al., 2005) has shown **privacy preferences can vary across different data owners**. Moreover, data owners with more sensitive attributes (e.g., stigmatized diseases, race) would prefer stronger privacy as data leakage would expose them to discrimination or denial or services (Best & Arseniev-Koehler, 2023; Lai & Tanner, 2022). We have provided several real life applications in App. A.3.

However, choosing the maximum privacy level is undesirable: The *privacy-utility tradeoff* (Makhdoumi & Fawaz, 2013) is well-studied and shows that using a stronger DP guarantees undesirably reduces the performance (utility) of the trained ML model. Thus, a model owner would want to use each data owner's data at a weaker privacy to train a model with the highest utility. In some cases such as collaborative ML (Sim et al., 2020; Tian et al., 2024), data owners are also the users of trained ML models and hence they will choose the highest privacy budgets they find acceptable (e.g., when data owners are hospitals and the model owner is a medical institute who trains an ML model to predict the risk of heart disease). In other case, model owners can incentivize data owners to choose the highest privacy budgets they find acceptable by rewarding each data owner based on their data values or privacy budgets (i.e., selecting a higher privacy budget and weaker privacy will lead to more monetary compensation).

2. **Why is the problem of the utility imbalance (and improving the utility on more private owners or difficult data) important?**

As motivated in Sec. 1, model owners need to collect data from multiple data owners in order to gather sufficient data to train a good ML model. Moreover, data owners with unique yet sensitive data would prefer a stronger privacy, which is not desired by model owners as this would lead to low model utility on these unique data. For example, in healthcare ML applications, data owners with some rare disease are likely to demand strong privacy requirements. However, the goal of training such an ML model is precisely to predict these rare diseases well instead of predicting well for control patients without the diseases. Thus, the model owner should consider using INO-SGD to improve the utility on these more difficult examples and more private data owners.

3. **What is the difference between this and other works that achieve group fairness?**

Our goal differs significantly from group fairness works as explained in App. B.3.

4. **Do we assume that MID and BOO always occur as a result of IDP-SGD?**

No, we do not assume that they always occur. In Thm. 3.1, we theoretically analyze when MID and BOO will not occur and how to correct for it when they occur. Based on that, we devise the INO-SGD algorithm.
However, we highlight that even for balanced datasets, MID and BOO can occur as a result of IDP requirements across different owners and theoretically prove it in App. C.1.2.

5. **In Sec. 4, why does the model achieve relatively low accuracy? Is this weaker than existing works?**

We would like to highlight that our empirical results should be compared against IDP-SGD with same privacy budgets or DP-SGD that maintains the smallest privacy budget for all owners to attain IDP. For example, in the CIFAR-10 experiment where IDP-SGD achieves $\sim 50\%$ overall accuracy in Fig. 14(i), the most private owner has a small privacy budget $\epsilon = 0.5$. When privacy budget is larger (minimum $\epsilon = 6$), our accuracy also increases and reaches $\sim 67\%$ in 7(b).
Similar accuracy to ours (e.g. $< 60\%$ for small $\epsilon$) in DP-SGD is common for CIFAR-10 training from scratch (Ganesh et al., 2023; Muthu Selva Annamalai & De Cristofaro, 2024; Wei et al., 2022). (Bu et al., 2022; Ganesh et al., 2023; Lin et al., 2023) use pretrained models to achieve high accuracy around $90\%$ which we also show in Fig. 14(m). Notably, our work considers IDP where no follow-up works apply.
Besides privacy budget, the impact of data imbalance on model utility is important too. DP can potentially reduce accuracy on underrepresented groups (Bagdasaryan et al., 2019) and our work shows that IDP can cause it too. INO-SGD would address this data imbalance.

## USE OF LLMS

We use LLMs to aid or polish writing such as checking grammar. The methodological contributions do not involve LLMs.

