# OpenReview forum: "INO-SGD: Addressing Utility Imbalance under Individualized Differential Privacy"
_ICLR.cc/2026/Conference — ICLR 2026 Poster_

### Official Review · Reviewer_mPUX · 2025-10-26

**Soundness:** 3
**Presentation:** 4
**Contribution:** 3
**Rating:** 8
**Confidence:** 3

**Summary:**

This paper introduces the novel problem of **utility imbalance** caused by individualized differential privacy (IDP). In IDP, each data owner specifies their own privacy level, which offers flexibility but leads to uneven model performance across owners with different privacy budgets. The authors propose **Individualized Noisy Ordered SGD (INO-SGD)**, an algorithm that **adjusts the weighting of data within each batch** to improve outcomes for owners with stricter privacy requirements while maintaining the same privacy guarantee as standard IDP-SGD. Through extensive experiments, the authors demonstrate that INO-SGD enhances the utility of more private groups, effectively mitigating utility imbalance, while maintaining or even improving overall model performance.

**Strengths:**

**1. Novel and Important Problem Identification:**
The paper introduces the new problem of utility imbalance under individualized differential privacy. It shows that while IDP gives users control over their privacy, existing algorithms unintentionally penalize those with stronger privacy. Its focus on how varying privacy budgets create uneven model performance highlights an important direction for future research in differential privacy.

**2. Strong Theoretical Justification and Clear Presentation:**
The paper provides a well-structured and convincing theoretical motivation for the identified problem. By connecting the imbalance to sampling rates and clipping thresholds through concepts like Minority Initial Drop and Biased Optimization Objective, it offers clear mathematical insight into the cause of the issue. Theorem 3.1 and its supporting proofs further reinforce the soundness of the argument.

**3. Comprehensive Empirical Validation:**
The experiments are extensive and well-executed, covering a diverse range of datasets, models, and privacy settings. The results consistently show that INO-SGD improves the performance of more private groups, effectively mitigating utility imbalance, while maintaining or even enhancing overall model utility.

**4. Novel and Well-Designed Mechanism:**
The idea behind INO-SGD is to down-weight less important data within each batch to improve performance on more private data throughout training. This approach is practical and broadly applicable, with potential use in learning problems that involve fairness or group balance.

**Weaknesses:**

**1.** In some parts, the paper’s wording seems to overstate the performance gains of INO-SGD. For example, lines 444–446 claim that *Across training stages, INO-SGD significantly increases the recall on validation data groups associated with more private owners (left) at the expense of a **small decrease** in the recall on less private ones (right).* However, from Figure 6, it appears that for larger $\varepsilon$ values (less private owners), IDP-SGD clearly outperforms INO-SGD, and the performance drop for these groups is roughly equal to the gain for more private ones. Additionally, Figures 5 and 7 use validation loss as the utility metric for the MNIST dataset, while recall/accuracy are used for CIFAR-10 and CIFAR-100-FV. Is there a specific reason for this inconsistency? If not, I would suggest using a consistent evaluation metric across datasets for clarity.

**2.** The paper states that INO-SGD guarantees the same privacy level as IDP-SGD (Theorems 2.2 and 3.3). While the paper mentions that INO-SGD consumes privacy budgets at the same rate as IDP-SGD, and therefore maintains the same number of training iterations, it remains unclear why the two methods have exactly the same privacy guarantee at **each iteration**. For instance, INO-SGD involves computing the Batch Importance Function (BIF) from $f_{\text{tail}}$ and $B_t$, as well as sorting losses—operations specific to INO-SGD. Could the authors provide more intuition or justification for why these additional computations do not affect the per-iteration privacy guarantee?

**Questions:**

See the Weaknesses section, please.

---

> ### Author Response · Authors · 2025-11-21
>
> Thank you for reviewing our work and appreciating our clarity, novelty, and theoretical and empirical contribution. Here are our responses to your concerns:
>
> > In some parts, the paper’s wording seems to overstate the performance gains of INO-SGD. For example, lines 444–446 claim that Across training stages, INO-SGD significantly increases the recall on validation data groups associated with more private owners (left) at the expense of a small decrease in the recall on less private ones (right). However, from Figure 6, it appears that for larger  values (less private owners), IDP-SGD clearly outperforms INO-SGD, and the performance drop for these groups is roughly equal to the gain for more private ones.
>
> Thank you for bringing this up! We use the wording *small decrease* because a reduction in imbalance/an improvement in the worse groups’ utility typically comes at the expense of a reduction in overall utility/much larger decrease in the better groups’ utility in tangential fields (e.g., [1, 2, 3, 4]). However, in our case, such a decrease usually does not surpass the improvement in the worse group’s utility, thus we claimed a small decrease. However, we agree that this may cause confusion, and we have updated our manuscript in OpenReview. For example,
>
> *INO-SGD significantly increases the recall on validation data groups associated with more private owners (left) at the expense of a **moderate** decrease in the recall on less private ones (right).*
>
>
>
> > Additionally, Figures 5 and 7 use validation loss as the utility metric for the MNIST dataset, while recall/accuracy are used for CIFAR-10 and CIFAR-100-FV. Is there a specific reason for this inconsistency? If not, I would suggest using a consistent evaluation metric across datasets for clarity.
>
> Thanks for raising this question. We analyze and solve the problem theoretically through the lens of “loss” and hence we include some results using loss to empirically demonstrate its effectiveness. We also wish to empirically demonstrate recall/accuracy results as it is a common utility metric and used in prior IDP works.
> We include the accuracy results for all experiments in App. D.2 and D.3 and the trend is consistent. We have included the above explanation to avoid confusion in main paper Footnote 6.
>
>
>
>
> > The paper states that INO-SGD guarantees the same privacy level as IDP-SGD (Theorems 2.2 and 3.3). While the paper mentions that INO-SGD consumes privacy budgets at the same rate as IDP-SGD, and therefore maintains the same number of training iterations, it remains unclear why the two methods have exactly the same privacy guarantee at each iteration. For instance, INO-SGD involves computing the Batch Importance Function (BIF) from  and , as well as sorting losses—operations specific to INO-SGD. Could the authors provide more intuition or justification for why these additional computations do not affect the per-iteration privacy guarantee?
>
> We provide a formal proof of Thm. 3.3 in App. C.3.1. To summarize, the only output by INO-SGD at each iteration is the summed gradient $\overline{\mathbf{G}}_{t}$ and it suffices to add noise proportional to its sensitivity (i.e., how a single datum may cause it to change). INO-SGD is specially designed such that the sensitivity is bounded (by ensuring each datum has a bounded change in weight for neighboring datasets), i.e., privacy guarantee is satisfied.
>
> In App. C.3.1, we have also remarked why the additional computations like sorting losses do not incur extra privacy cost. This is because the sorting order at iteration $t$ comes from the ML model obtained at iteration $t-1$, which is already protected by DP in earlier iterations (i.e., similar to the **gradients** we compute at iteration $t$ for each sampled datum). Thus, for a mechanism using the gradients and/or the order of loss, as long as we
> bound the (modular) sensitivity of its output at iteration $t$, it strictly satisfies IDP by adaptive composition.
>
> We hope the above justification helps and have added it to Sec. 3.3.1 in our updated manuscript in OpenReview.
>
>
> [1] Is fairness only metric deep? Evaluating and addressing subgroup gaps in deep metric learning.
>
> [2] Inherent tradeoffs in learning fair representations.
>
> [3] Pareto invariant risk minimization: Towards mitigating the optimization dilemma in out-of-distribution generalization.
>
> [4] Group-robust sample reweighting for subpopulation shifts via influence functions.

---

> > ### Comment · Reviewer_mPUX · 2025-11-21
> >
> > Thanks for the clarification and the responses. I have no further questions.

---

### Official Review · Reviewer_WnuE · 2025-10-31

**Soundness:** 2
**Presentation:** 3
**Contribution:** 3
**Rating:** 6
**Confidence:** 4

**Summary:**

The paper studies utility imbalance under individualized differential privacy (IDP). It shows that even on balanced datasets, IDP-SGD (SAMPLE/SCALE) can disadvantage owners (or groups) with stricter privacy (smaller ε) by creating MID (minority initial drop) during training and BOO (biased optimization objective) at convergence. To mitigate this, the authors propose INO-SGD, which (i) sorts per-batch examples by loss, (ii) uses a Tail Importance Function (TIF) to define a Batch Importance Function (BIF), and (iii) rescales clipped gradients by per-rank importance scores before adding Gaussian noise—while keeping each datum’s modular sensitivity bounded by its individualized clipping threshold, thereby preserving IDP.

**Strengths:**

The problem framing is clear. IDP can itself create imbalanced learning dynamics (MID/BOO) even when class counts are balanced. This problem actually step away practical DP deployment.

The intuition makes sense. To my understanding, it aims to increase the sensitive users' utility by trading off some non-sensitive users' utility. Meanwhile, the loss-ordered importance scheme is designed to not increase per-datum sensitivity (keeps ΔA^d ≤ Co(d)) while re-weighting gradient contributions.

**Weaknesses:**

The narrative argues standard class-imbalance fixes don’t satisfy IDP or get neutralized by clipping (oversampling/weighting/rescaling), but empirical comparisons with IDP-compliant alternatives are missing.

Claims rest on formal sensitivity bounds and RDP accounting; there’s no auditing checking if per-owner attack success (e.g., per-group LiRA, TPR/FPR rates) still meet TPR/FPR<=exp(ε). Though formal analysis makes sense, using auditing to check if there is any bug in implementation is also important.

The main figures emphasize per-group recall and validation loss/accuracy. To assess “imbalance”, measures like worst-group accuracy should be reported directly.

**Questions:**

Please check the weaknesses.

---

> ### Author Response · Authors · 2025-11-21
>
> Thank you for reviewing our work and appreciating our motivation and theoretical contribution! Here are our responses to your concerns:
>
> > The narrative argues standard class-imbalance fixes don’t satisfy IDP or get neutralized by clipping (oversampling/weighting/rescaling), but empirical comparisons with IDP-compliant alternatives are missing.
>
> Indeed, the narrative in Sec. 3.1 and App. B.3 argues that, **there is no prior work that are IDP-compliant and addresses the IDP-induced utility imbalance problem we propose**. Moreover, we provide some **empirical comparisons with straightforward IDP-compliant alternatives** (the only ones to the best of our knowledge) in the appendix. These alternatives, such as random undersampling, waste privacy budgets. In App. D.1.3 and App. D.4.3, we empirically validate our claim that methods that waste privacy budgets would lead to a severe decrease in overall model utility and thus a much worse Pareto curve between utility imbalance and overall utility when compared with INO-SGD. We are willing to discuss more if you have other alternatives in mind!
>
> > Claims rest on formal sensitivity bounds and RDP accounting; there’s no auditing checking if per-owner attack success (e.g., per-group LiRA, TPR/FPR rates) still meet TPR/FPR<=exp(ε). Though formal analysis makes sense, using auditing to check if there is any bug in implementation is also important.
>
> Thank you for the good suggestion! We have added a section **Privacy Assessment via Membership Inference Attack** (App. D.4.5) in the updated manuscript in OpenReview which includes a brief introduction to the LiRA method and empirical results/figures demonstrating how INO-SGD’s output is robust to such attacks. To summarize, we verify that the AUROC for each owner is close to 0.5 and closer for the more private owners, similar to IDP-SGD. This empirically supports our formal analysis.
>
> > The main figures emphasize per-group recall and validation loss/accuracy. To assess “imbalance”, measures like worst-group accuracy should be reported directly.
>
> Thank you for your suggestion! We agree that worst-group accuracy is a good metric to assess “imbalance” and we have added the worst-group accuracy results for our experiments in App. D.2 Fig. 11 in our latest manuscript (please refer to our updated manuscript in OpenReview). The results show that INO-SGD consistently improves the worst-group accuracy.
>
> Thank you for your valuable suggestions and we hope our response can improve your opinion of our work!

---

### Official Review · Reviewer_Tg2g · 2025-11-01

**Soundness:** 3
**Presentation:** 2
**Contribution:** 3
**Rating:** 6
**Confidence:** 3

**Summary:**

The paper identifies a utility imbalance that arises when training with individualized DP (IDP): data owned by users with stricter privacy budgets (e.g., sensitive subpopulations) receives less effective learning, degrading their group utility even when dataset sizes are balanced. The authors analyze why IDP-SGD induces MID/BOO-style dynamics and propose INO-SGD, which (i) sorts per-batch examples by loss, (ii) constructs a tail importance function (TIF) and corresponding batch importance function (BIF) to assign importance weights, and (iii) scales clipped per-example gradients by these data-dependent weights without exceeding each owner’s clipping threshold, thereby preserving IDP guarantees. They prove IRDP privacy with the same accounting as IDP-SGD and show that INO-SGD can obtain gains on MNIST, CIFAR-10/100, improving the more private groups’ recall while preserving or improving overall accuracy.

**Strengths:**

+ The paper clearly identifies and analyzes a practical failure mode of IDP training, showing how stricter privacy budgets systematically depress group utility even without data imbalance.

+ The proposed method preserves individualized DP guarantees without additional privacy cost, because it reweights only after clipping and uses the same accounting as IDP-SGD.

+ The experimental results consistently improve performance for highly private users while keeping or improving global accuracy, demonstrating that the approach meaningfully reduces utility imbalance in practice.

**Weaknesses:**

- The empirical evaluation lacks breadth against strong IDP-compliant alternatives, as it focuses mainly on vision benchmarks and does not compare to a wider set of within-IDP reweighting or sampling schemes.

- The method provides limited guidance for selecting the importance-function hyperparameters, leaving open how to tune them or manage trade-offs across privacy groups in real deployments.

**Questions:**

Refer to the weaknesses.

---

> ### Author Response · Authors · 2025-11-21
>
> Thank you for reviewing our work and appreciating the practicality of our identified problem and the effectiveness of our proposed solution. Here are our responses to your concerns:
>
> > The empirical evaluation lacks breadth against strong IDP-compliant alternatives, as it focuses mainly on vision benchmarks and does not compare to a wider set of within-IDP reweighting or sampling schemes.
>
> We provide **a wider breadth of evaluation against IDP-compliant alternatives** and justification on why the total breadth may be limited due to **a lack of prior works** in the appendix.
>
> In Sec. 3.1 and App. B.3, we explain that **there is no prior work that is both IDP-compliant and addresses the IDP-induced utility imbalance problem we propose**. To the best of our knowledge, we consider straightforward reweighting or sampling schemes in App. C.2.1 and argue that **they either violate IDP or under-utilize privacy budgets**. For example,
> - methods that upscale the gradients from the more private groups would be exactly the same as vanilla IDP-SGD after applying the per-sample gradient clipping;
> - randomly underscaling (e.g. down-weighting or under-sampling) owners with higher privacy budgets under-utilizes privacy budgets.
> In App. D.1.3 and App. D.4.3, we empirically validate our claim that methods that under-utilize privacy budgets, such as random undersampling, would lead to a severe decrease in overall model utility and thus a much worse Pareto curve between utility imbalance and overall utility when compared with INO-SGD.
>
> We hope the above justification helps and are willing to discuss more if you have other alternatives in mind!
>
>
>
>
> > The method provides limited guidance for selecting the importance-function hyperparameters, leaving open how to tune them or manage trade-offs across privacy groups in real deployments.
>
> Thank you for raising this concern. We provide the following **guidelines** on how to tune the hyperparameters efficiently (for the Beta distribution based TIF):
> - the TIF with some tail length $\gamma_1$ and some $\beta_1$ has a similar shape to another TIF with a larger tail length $\gamma_2 >\gamma_1 $  and larger beta $\beta_2 > \beta_1$, so it suffices to fix a moderately large tail length $\gamma$ (e.g., 50% of the total clipping threshold) to only tune $\alpha$ and $\beta$; OR
> - fix $\alpha$ and $\beta$ (e.g., $1$ and $1$) and set the tail length to (or only tune the tail length around) a certain percentage of the expected sum of clipping thresholds for each batch.
>
> We demonstrate that **INO-SGD is robust to hyperparameter choice in Sec. 4.3 and App. D.4.2**. Thus, tuning according to the above guidelines is practical in real deployments.
> When the model owner is willing to trade more overall utility for less imbalance, they can manage the tradeoff by increasing the tail length $\gamma$ (the easiest way), or setting a larger $\alpha$ and smaller $\beta$.
> We hope the above guideline helps and have added it to App. C.2.5 in our updated manuscript in OpenReview.
>
> Thank you for your valuable suggestions and we hope our response can improve your opinion of our work!

---

### Author Response · Authors · 2025-12-03
**Final Responses and Summary (1/2)**

Dear AC and reviewers,

To conclude the discussion phase, we summarize the key discussion points below. Overall, all reviewers give our paper a positive rating ($ \geq 6$). As of November 27, one reviewer has replied to our rebuttal: Reviewer mPUX was [satisfied with our explanations and had no further questions](#:~:text=Thanks%20for%20the%20clarification%20and%20the%20responses.%20I%20have%20no%20further%20questions.).

The reviewers highlight the following strengths of our paper:
- All reviewers agree that **our problem framing is clear and practical** ([Tg2g](#:~:text=The%20paper%20clearly%20identifies%20and%20analyzes%20a%20practical%20failure%20mode%20of%20IDP%20training), [WnuE](#:~:text=The%20problem%20framing%20is%20clear.), [mPUX](#:~:text=Novel%20and%20Important%20Problem%20Identification)). It also provides **an important research direction** ([mPUX](#:~:text=Its%20focus%20on%20how%20varying%20privacy%20budgets%20create%20uneven%20model%20performance%20highlights%20an%20important%20direction%20for%20future%20research%20in%20differential%20privacy.)). All reviewers have all rated the contribution as 3: good.
- Our paper has a **strong and sound theoretical justification** ([mPUX](#:~:text=Strong%20Theoretical%20Justification,of%20the%20argument.)).
- Reviewers Tg2g and mPUX agree that **our proposed method is meaningful and effective** ([Tg2g](#:~:text=the%20approach%20meaningfully%20reduces%20utility%20imbalance%20in%20practice), [mPUX](#:~:text=Novel%20and%20Well,or%20group%20balance.)). Our proposed method is also **intuitive and theoretically sound** ([WnuE](#:~:text=The%20intuition%20makes,weighting%20gradient%20contributions.)).
- The empirical evaluation is **comprehensive and well-executed** ([mPUX](#:~:text=Comprehensive%20Empirical%20Validation,overall%20model%20utility.)). The results **consistently show the effectiveness of our method** ([Tg2g](#:~:text=The%20experimental%20results%20consistently%20improve%20performance%20for%20highly%20private%20users%20while%20keeping%20or%20improving%20global%20accuracy), [mPUX](#:~:text=The%20results%20consistently%20show%20that%20INO%2DSGD%20improves%20the%20performance%20of%20more%20private%20groups%2C%20effectively%20mitigating%20utility%20imbalance%2C%20while%20maintaining%20or%20even%20enhancing%20overall%20model%20utility.)).

---

**(Continued below.)**

---

> ### Author Response · Authors · 2025-12-03
> **Final Responses and Summary (2/2)**
>
> We summarize the reviewers’ concerns and our responses below. We have also updated our manuscript in OpenReview according to the reviewers' suggestions.
> - *Comparison with “IDP-compliant alternatives”* ([Tg2g](#:~:text=Weaknesses%3A-,The%20empirical%20evaluation%20lacks%20breadth%20against%20strong%20IDP%2Dcompliant%20alternatives%2C%20as%20it%20focuses%20mainly%20on%20vision%20benchmarks%20and%20does%20not%20compare%20to%20a%20wider%20set%20of%20within%2DIDP%20reweighting%20or%20sampling%20schemes.,-The%20method%20provides), [WnuE](#:~:text=(oversampling/weighting/rescaling)%2C-,but%20empirical%20comparisons%20with%20IDP%2Dcompliant%20alternatives%20are%20missing.,-Claims%20rest%20on)): We have explained that since our proposed problem is novel, there is a lack of prior work addressing it as discussed in App C.2.1. Additionally, we highlighted to the reviewers that [we have empirically shown that INO-SGD outperformed other IDP-compliant alternatives in App D.1.3 and App D.4.3](#:~:text=utilizes%20privacy%20budgets.-,In%20App.%20D.1.3%20and%20App.%20D.4.3%2C%20we%20empirically%20validate,between%20utility%20imbalance%20and%20overall%20utility%20when%20compared%20with%20INO%2DSGD.,-We%20hope%20the).
> - *Guideline for hyperparameter selection* ([Tg2g](#:~:text=or%20sampling%20schemes.-,The%20method%20provides%20limited%20guidance%20for%20selecting%20the%20importance%2Dfunction%20hyperparameters%2C%20leaving%20open%20how%20to%20tune%20them%20or%20manage%20trade%2Doffs%20across%20privacy%20groups%20in%20real%20deployments.,-Questions%3A)): We have provided a practical [guideline in our rebuttal](#:~:text=Thank%20you%20for%20raising,updated%20manuscript%20in%20OpenReview.) and added it to App. C.2.5 of our manuscript.
> - *Privacy auditing using attacks* ([WnuE](#:~:text=and%20RDP%20accounting%3B-,there%E2%80%99s%20no%20auditing%20checking%20if%20per%2Downer%20attack%20success%20(e.g.%2C%20per%2Dgroup%20LiRA%2C%20TPR/FPR%20rates)%20still%20meet%20TPR/FPR%3C%3Dexp(%CE%B5).%20Though%20formal%20analysis%20makes%20sense%2C%20using%20auditing%20to%20check%20if%20there%20is%20any%20bug%20in%20implementation%20is%20also%20important.,-The%20main%20figures)): We agree that such auditing would validate our formal theorem on privacy guarantee, and have added the LiRA attack experiments in App. D.4.5. The results support our privacy guarantee.
> - *Adding reports on worst-group accuracy to assess imbalance* ([WnuE](#:~:text=is%20also%20important.-,The%20main%20figures%20emphasize%20per%2Dgroup%20recall%20and%20validation%20loss/accuracy.%20To%20assess%20%E2%80%9Cimbalance%E2%80%9D%2C%20measures%20like%20worst%2Dgroup%20accuracy%20should%20be%20reported%20directly.,-Questions%3A)): We have added this for all our experiments in App. D.2 Fig. 11. The results consistently show that INO-SGD effectively addresses utility imbalance.
> - *Rationale for using both loss and recall/accuracy* ([mPUX](#:~:text=private%20ones.%20Additionally%2C-,Figures%205%20and%207%20use%20validation%20loss%20as%20the%20utility%20metric%20for%20the%20MNIST%20dataset%2C%20while%20recall/accuracy%20are%20used%20for%20CIFAR%2D10%20and%20CIFAR%2D100%2DFV.%20Is%20there%20a%20specific%20reason%20for%20this%20inconsistency%3F%20If%20not%2C%20I%20would%20suggest%20using%20a%20consistent%20evaluation%20metric%20across%20datasets%20for%20clarity.,-2.%20The)): We have clarified that [we report both because we analyze and solve the problem theoretically through the lens of “loss” while recall/accuracy is a common utility metric and used in prior IDP works](#:~:text=Thanks%20for%20raising,paper%20Footnote%206.). We have added the clarification to the main paper in Footnote 6.
> - *Intuition on why INO-SGD does not incur extra privacy cost* ([mPUX](#:~:text=of%20training%20iterations%2C-,it%20remains%20unclear%20why%20the%20two%20methods%20have%20exactly%20the%20same,these%20additional%20computations%20do%20not%20affect%20the%20per%2Diteration%20privacy%20guarantee%3F,-Questions%3A)): We have provided a more intuitive explanation (which Reviewer mPUX is satisfied with) and added it to Sec. 3.3.1.
>
> We greatly appreciate all reviewers' and the area chair's time, efforts, and insightful feedback.
>
> Regards,
>
> Authors of Submission 23358

---

### Meta-Review · Area_Chair_W7G3 · 2026-01-07

**Summary:**

This paper proposes a new privacy preserving algorithm that's designed to satisfy individualized differential privacy. Reviewers seem positive about this work given the empirical performance. The algorithm itself also looks intuitive and elegant. They down sample gradients that have lower loss in a privacy aware manner to avoid the model from overfitting to those samples.

**Reviewer Concerns:**

Two major concerns raised by reviewers are not 1) validating the results using auditing and 2) not having proper guid for selecting the hyper parameters (specifically the function that determines the tail of the batch). Authors address this by adding proper auditing experiments and also a section on how to choose the hyper-parameters.

**Reviewer Scores:**

I believe reviewers Tg2g and mPUX will not change their score but reviewer WnuE may raise the score given the new auditing experiments provided by authors.

---

### Decision · Program_Chairs · 2026-01-26

Accept (Poster)